# Unifying Causal Representation Learning with the Invariance Principle

**Dingling Yao**, **Dario Rancati**, **Riccardo Cadei**, **Marco Fumero**, and **Francesco Locatello**

Institute of Science and Technology Austria

## Abstract

Causal representation learning (CRL) aims at recovering latent causal variables from high-dimensional observations to solve causal downstream tasks, such as predicting the effect of new interventions or more robust classification. A plethora of methods have been developed, each tackling carefully crafted problem settings that lead to different types of identifiability. These different settings are widely assumed to be important because they are often linked to different rungs of Pearl's causal hierarchy, even though this correspondence is not always exact. This work shows that instead of strictly conforming to this hierarchical mapping, many causal representation learning approaches methodologically align their representations with inherent data symmetries. Identification of causal variables is guided by invariance principles that are not necessarily causal. This result allows us to unify many existing approaches in a single method that can mix and match different assumptions, including non-causal ones, based on the invariance relevant to the problem at hand. It also significantly benefits applicability, which we demonstrate by improving treatment effect estimation on real-world high-dimensional ecological data. Overall, this paper clarifies the role of causal assumptions in the discovery of causal variables and shifts the focus to preserving data symmetries.

## 1 Introduction

Causal representation learning (Schölkopf et al., 2021) posits that many high-dimensional perceptual data can be described through a simplified latent structure specified by a few low-dimensional causally-related variables. Discovering hidden causal structures from data has been a long-standing goal across many scientific disciplines, spanning neuroscience (Vigário et al., 1997; Brown et al., 2001), communication theory (Ristaniemi, 1999; Donoho, 2006), economics (Angrist & Pischke, 2009) and social science (Antonakis & Lalive, 2011). From the machine learning perspective, algorithms and models integrated with causal structure are often proven to be more robust at distribution shift (Ahuja et al., 2022a; Bareinboim & Pearl, 2016; Rojas-Carulla et al., 2018), providing better out-of-distribution generalization results and reliable agent planning (Fumero et al., 2024; Seitzer et al., 2021; Urpí et al., 2024). The general goal of CRL approaches is formulated as to provably identify ground-truth latent causal variables and their causal relations (up to certain ambiguities). Many existing approaches in causal representation learning carefully formulate their problem settings to guarantee identifiability and justify the assumptions within the framework of Pearl's causal hierarchy, such as "observational, interventional, or counterfactual CRL" (von Kügelgen et al., 2024; Ahuja et al., 2023; Brehmer et al., 2022; Buchholz et al., 2024; Zhang et al., 2024a; Varici et al., 2024a). Yet, several emerging lines of work reveal that not all approaches adhere strictly to this framework; for instance, the problem setting of temporal CRL works (Lachapelle et al., 2022; Lippe et al., 2022a;b; 2023) does not always align straightforwardly with existing categories. They often assume that an individual trajectory is "intervened" upon, but this is not an intervention in the traditional sense, as noise variables are not resampled. It is also not a counterfactual, as the value of non-intervened variables can change due to default dynamics. Similarly, domain generalization (Sagawa et al., 2019; Krueger et al., 2021; Ahuja et al., 2022a) and certain multi-task learning approaches (Lachapelle et al., 2023; Fumero et al., 2024) are sometimes framed as informally related to CRL. However, the precise relation to causality is not always clearly articulated. Consequently, a wide range of methods and empirical findings has emerged, although some of these approaches rely on assumptions that might be too narrowly tailored for practical, real-world applications. For example, Cadei et al. (2024) collected a dataset for estimating treatment effects from high-dimensional

observations in real-world ecology experiments. Despite the clear causal focus of the benchmark, they note that even when multiple views and interventions are accessible, neither existing multiview nor interventional CRL methods are directly applicable due to mismatching assumptions.

This paper contributes a unified framework of many existing CRL works through the lens of invariance (Peters et al., 2014; Heinze-Deml et al., 2018; Arjovsky et al., 2020). We observe that *many existing CRL approaches share methodological similarities, particularly in aligning the representation with known data symmetries*, while differing primarily in how the invariance principle is invoked. Typically, the invariance principle is formulated implicitly within the assumed data-generating process. By making it explicit, we demonstrate that latent causal variable identification originates from various data subsets characterized by inherent equivalence relations, which are often known a priori with a few exceptions. This explicit formulation not only unifies disparate CRL methods but also offers clear practical advantages. First, it helps clarify the alignment between seemingly different categories of CRL methods, contributing to a more coherent and accessible framework for understanding CRL. This perspective may also allow for the integration of multiple invariance relations in latent variable identification, which could improve the flexibility of these methods in certain practical settings. Additionally, our theory highlights a critical discrepancy between the assumptions required for causal graph learning and those necessary for identifying causal variables: While graph learning often requires explicit causal assumptions such as interventions, the invariance principles for variable identification do not have to be causal. Last but not least, this formulation of invariance relation links CRL to many existing representation learning areas outside of causality, including invariant training (Arjovsky et al., 2020; Ahuja et al., 2022a), domain adaptation (Sagawa et al., 2019; Krueger et al., 2021), and geometric deep learning (Cohen & Welling, 2016; Bronstein et al., 2017; 2021).

We highlight our contributions as follows:

- We propose a unified framework for existing CRL approaches that leverages the invariance principles and proving latent variable identifiability in this general setting (§ 3). We show that 30 existing identification results can be seen as special cases directly implied by our framework (Tab. 4). This approach also enables us to derive new results, including latent variable identifiability from one imperfect intervention per node in the nonparametric setting (Cor. D.1).

- In addition to employing different methods, many CRL works use varying definitions of "identifiability." We formalize these definitions at different levels of granularity and highlight their interconnections (Proposition C.1). Moreover, we show that existing CRL algorithms can achieve different levels of identifiability based on additional (e.g., parametric) assumptions, obviating the need for separate proofs in each case (Proposition C.2).

- Upon the identifiability of the latent variables, we discuss the necessary causal assumptions for graph identification and the possibility of partial graph identification using the language of causal consistency. With this, we draw a distinction between the causal assumptions necessary for graph discovery (such as interventions or graphical assumptions) and those required for variable discovery (App. C.2). This distinction relaxes the stringent requirements typically imposed for latent variable identifiability, thereby enhancing the framework's applicability in real-world settings.

- Our framework is broadly applicable across a range of settings. We observe improved results on real-world experimental ecology data using a high-dimensional causal inference benchmark by Cadei et al. (2024) (§ 5.1). Additionally, we present a synthetic ablation to demonstrate that existing methods, which assume access to interventions, actually only require a form of distributional invariance to identify variables. This invariance does not necessarily need to correspond to a valid causal intervention (§ 5.2).

## 2 PROBLEM SETTING

**Notation.** $[N]$ is used as a shorthand for $\{1, \ldots, N\}$. We use bold lower-case $\mathbf{z}$ for random vectors and normal lower-case $z$ for their realizations. A vector $\mathbf{z}$ can be indexed either by a single index $i \in [\dim(\mathbf{z})]$ via $\mathbf{z}_i$ or a index subset $A \subseteq [\dim(\mathbf{z})]$ with $\mathbf{z}_A := \{\mathbf{z}_i : i \in A\}$. $P_{\mathbf{z}}$ denotes the probability distribution of the random vector $\mathbf{z}$ and $p_{\mathbf{z}}(z)$ denotes the associated probability density function (We omit the subscription and write $p(z)$ when the context is clear). By default, a "measurable" function is *measurable* w.r.t. the Borel sigma algebras and defined w.r.t. the Lebesgue measure. A more comprehensive summary of notations is provided in App. A.

Table 1: Concrete examples of CRL categories and domain generalization are unified by our framework, their invariance, and a non-exhaustive list of corresponding references. The invariant partition $A$ is highlighted with a smoke blue box (*: $\mathcal{I}_{\mathbf{z}_A^1}$ represents the interventional target for $\mathbf{z}^1$ which is $\{1\}$ in this example). For further technical details and an in-depth discussion on how various approaches fit within our unified framework, see Appendix App. D.

| Category | Example | Invariance | Related work |
|---|---|---|---|
| Multiview CRL |  | Sample level invariance $\mathbf{z}_A^1 = \mathbf{z}_A^2$ | Locatello et al. (2020); von Kügelgen et al. (2021); Gresele et al. (2020) |
| Interventional CRL (two interventions per node) |  | *Same interventional target $\mathcal{I}_{\mathbf{z}_A^1} = \mathcal{I}_{\mathbf{z}_A^2}$ | von Kügelgen et al. (2024); Varici et al. (2024a) |
| Interventional CRL (one intervention per node) |  | Marginal invariance $p_{\mathbf{z}_A^1} = p_{\mathbf{z}_A^2}$ | Zhang et al. (2024a); Squires et al. (2023); Buchholz et al. (2024) |
| Interventional CRL (one intervention per node) |  | Score invariance $S_{\mathbf{z}_A^1} = S_{\mathbf{z}_A^2}$ | Varici et al. (2023; 2024a) |
| Temporal CRL |  | Transition invariance $p_{\mathbf{z}_A\mid\mathbf{z}^{t-1}} = p_{\tilde{\mathbf{z}}_A\mid\mathbf{z}^{t-1}}$ | Lippe et al. (2022b;a; 2023) |
| Multi-task CRL |  | Overlapping task support $\mathbf{z}_A^{T_1} = \mathbf{z}_A^{T_2}$ | Lachapelle et al. (2023); Fumero et al. (2024) |
| Domain generalization |  | Risk invariance on optimal weights $\mathcal{R}_1^*(\mathbf{w}^*\mathbf{z}_A^1, \mathbf{y}^1) = \mathcal{R}_2^*(\mathbf{w}^*\mathbf{z}_A^2, \mathbf{y}^2)$ | Arjovsky et al. (2020); Krueger et al. (2021); Ahuja et al. (2022a) |

This section defines our problem setting using standard CRL concepts and assumptions (Formal definitions are deferred to App. B). While prior works in CRL typically categorize their settings using established causal language (such as "counterfactual," "interventional," or "observational"), our approach introduces a more general invariance principle that aims to unify diverse problem settings. We introduce the following concepts as mathematical tools to describe our data generating process.

**Definition 2.1** (Invariance property). Let $A \subseteq [N]$ be an index subset of the Euclidean space $\mathbb{R}^N$ and let $\sim_\iota$ be an equivalence relationship on $\mathbb{R}^{|A|}$, with $A$ of known dimension. Let $\mathcal{M} := \mathbb{R}^{|A|}/_{\sim_\iota}$ be the quotient of $\mathbb{R}^{|A|}$ under this equivalence relationship; $\mathcal{M}$ is a topological space equipped with the quotient topology. Let $\iota : \mathbb{R}^{|A|} \to \mathcal{M}$ be the projection onto the quotient induced by the equivalence relationship $\sim_\iota$. This projection $\iota$ is termed the *invariance property* of this equivalence relation. Two vectors $\mathbf{a}, \mathbf{b} \in \mathbb{R}^{|A|}$ are invariant under $\iota$ if and only if they belong to the same $\sim_\iota$ equivalence class, i.e.:

$$\iota(\mathbf{a}) = \iota(\mathbf{b}) \Leftrightarrow \mathbf{a} \sim_\iota \mathbf{b}.$$

Extending this definition to the entire latent space $\mathbb{R}^N$, we say that two latent vectors $\mathbf{z}, \tilde{\mathbf{z}} \in \mathbb{R}^N$ are ***non-trivially*** invariant on a subset $A \subseteq [N]$ if

(i) the invariance property $\iota$ holds on the indices $A \subseteq [N]$ in the sense that $\iota(\mathbf{z}_A) = \iota(\tilde{\mathbf{z}}_A)$;

(ii) for any smooth functions $h_1, h_2 : \mathbb{R}^N \to \mathbb{R}^{|A|}$, the invariance property between $\mathbf{z}, \tilde{\mathbf{z}}$ *breaks* under the $h_1, h_2$ transformations if either function depends on some other component $\mathbf{z}_q$ with $q \in [N] \setminus A$. More formally, considering $h_1$ and $\mathbf{z}$ as an example,

$$\exists q \in [N] \setminus A, \mathbf{z}^* \in \mathbb{R}^N, \quad s.t. \ \frac{\partial h_1}{\partial \mathbf{z}_q}(\mathbf{z}^*) \text{ exists and is non zero} \quad \Rightarrow \quad \iota(h_1(\mathbf{z})) \neq \iota(h_2(\tilde{\mathbf{z}})).$$

This indicates the output of function $h_1$ is influenced by a latent variable $\mathbf{z}_q$ outside the invariant set $A$, thereby violating the invariance property.

**Intuition**: The invariance property $\iota$ maps the invariant latent subset $\mathbf{z}_A$ to the quotient space $\mathcal{M}$. Both $\iota$ and $\mathcal{M}$ can take various concrete forms depending on the problem settings: In the multi-view literature (von Kügelgen et al., 2021; Brehmer et al., 2022; Yao et al., 2023), $\iota$ is the *identity map* because the pre-and post action views share the *exact value* of the invariant latents. For both interventional and temporal CRL (Varici et al., 2023; von Kügelgen et al., 2024; Lachapelle et al., 2022; Lippe et al., 2022a), $\iota$ acts as an operator that maps the invariant latent subset $\mathbf{z}_A$ to its associated density function – yielding the marginal density $p_{\mathbf{z}_A}$ in the interventional setting and conditional density $p_{\mathbf{z}_A^t|\mathbf{z}^{t-1}}$ in the temporal setting. In this context, the codomain $\mathcal{M}$ is the set of valid density functions. In multi-task CRL (Lachapelle et al., 2023; Fumero et al., 2024) and domain generalization (Arjovsky et al., 2020; Krueger et al., 2021), $\iota$ maps the overlapping task support and the risk implied by ground truth latent-target dependency, respectively. Concrete examples and detailed formulations of the corresponding invariance properties for each case are provided in Tab. 1.

**Remark**: Defn. 2.1 (ii) is crucial for ensuring latent variable identification on the invariant partition $A$. Its necessity is further justified in App. E.1, where we demonstrate that violating (ii) leads to non-identifiability. Intuitively, condition (ii) enforces a clear separation between the invariant and variant components of the ground truth generating process. This idea parallels key identifiability assumptions in CRL – albeit under different names – such as *sufficient variability* (von Kügelgen et al., 2024; Lippe et al., 2022b), *interventional regularity* (Varici et al., 2023; 2024b), and *interventional discrepancy* (Wendong et al., 2024; Varici et al., 2024a). At a high level, these assumptions ensure that the mechanism under intervention deviates sufficiently from the default causal mechanism, thereby allowing one to distinguish between intervened and non-intervened latent variablesa function that is directly served by Defn. 2.1 (ii). We elaborate on this link further in App. E.1.

We denote by $\mathcal{S}_{\mathbf{z}} := \{\mathbf{z}^1, \ldots, \mathbf{z}^K\}$ the set of latent random vectors with $\mathbf{z}^k \in \mathbb{R}^N$ and write its joint distribution as $P_{\mathcal{S}_{\mathbf{z}}}$. The joint distribution $P_{\mathcal{S}_{\mathbf{z}}}$ has a probability density $p_{\mathcal{S}_{\mathbf{z}}}(z^1, \ldots, z^K)$. Each individual random vector $\mathbf{z}^k \in \mathcal{S}_{\mathbf{z}}$ follows the marginal density $p_{\mathbf{z}^k}$ with the non-degenerate support $\mathcal{Z}^k \subseteq \mathbb{R}^N$, whose interior is a non-empty open set of $\mathbb{R}^N$.

**Definition 2.2** (Observable of a set of latent random vectors)**.** Consider a set of random vectors $\mathcal{S}_{\mathbf{z}} := \{\mathbf{z}^1, \ldots, \mathbf{z}^K\}$ with $\mathbf{z}^k \in \mathbb{R}^N$, the corresponding set of observables $\mathcal{S}_{\mathbf{x}} := \{\mathbf{x}^1, \ldots, \mathbf{x}^K\}$ is generated by $\mathcal{S}_{\mathbf{x}} = F(\mathcal{S}_{\mathbf{z}})$, where the map $F$ defines a push-forward measure $F_\#(P_{\mathcal{S}_{\mathbf{z}}})$ on the image of $F$ as:

$$F_\#(P_{\mathcal{S}_{\mathbf{z}}})(\mathbf{x}^1, \ldots, \mathbf{x}^K) = P_{\mathcal{S}_{\mathbf{z}}}(f_1^{-1}(\mathbf{x}^1), \ldots, f_K^{-1}(\mathbf{x}^K)) \tag{2.1}$$

with the support $\mathcal{X} := \mathrm{Im}(F) \subseteq \mathbb{R}^{K \times D}$. Note that $F$ satisfies the diffeomorphism assumption (Asm. B.1) as each $f_k$ is a diffeomorphism onto its image according to Asm. B.1.

**Intuition.** Defn. 2.2 formalizes how the set of observables $\mathcal{S}_{\mathbf{x}}$ is generated from a set of latent random vectors $\mathcal{S}_{\mathbf{z}}$ via a joint pushforward. For example, in the multiview scenario (von Kügelgen et al., 2021; Daunhawer et al., 2023; Yao et al., 2023), $P_{\mathcal{S}_{\mathbf{x}}}$ represents the distribution of the concurrently observed views $\{\mathbf{x}^1, \ldots, \mathbf{x}^k\}$. In interventional CRL, each observable $\mathbf{x}_k$ corresponds to data collected under different environments, so that $P_{\mathcal{S}_{\mathbf{x}}}$ factorizes into environment-specific distributions. In temporal CRL, the observable distribution $P_{\mathcal{S}_{\mathbf{x}}}$ explicitly conditions on the previous time step, reflecting the sequential dependencies inherent in the process. In the supervised setting, such as multi-task CRL and domain generalization, the observables $\mathbf{x}^k$ are augmented with task labels $\mathbf{y}^k$, capturing additional structure necessary for the task. Overall, Defn. 2.2 provides a unified formulation of the data-generating process across diverse settings, serving as the foundation for the unified latent variable identification techniques developed in later sections.

In the following, we denote by $\mathfrak{I} := \{\iota_i : \mathbb{R}^{|A_i|} \to \mathcal{M}_i\}$ a finite set of invariance properties with their respective invariant subsets $A_i \subseteq [N]$ and their equivalence relationships $\sim_{\iota_i}$, each inducing a projection onto its quotient and invariance property $\iota_i$ (Defn. 2.1). For a set of observables $\mathcal{S}_\mathbf{x} := \{\mathbf{x}^1, \dots, \mathbf{x}^K\} \in \mathcal{X}$ generated from the data generating process described in § 2, we assume:

**Assumption 2.1.** For each $\iota_i \in \mathfrak{I}$, there exists a unique, known index subset $V_i \subseteq [K]$ with at least two elements (i.e., $|V_i| \geq 2$) such that the corresponding observables $\mathbf{x}^{V_i} := \{\mathbf{x}^k : k \in V_i\}$ are generated from an equivalence class of latent vectors:

$$[\mathbf{z}]_{\sim_{\iota_i}} := \{\tilde{\mathbf{z}} \in \mathbb{R}^N : \mathbf{z}_{A_i} \sim_{\iota_i} \tilde{\mathbf{z}}_{A_i}\}.$$

Following Defn. 2.2, the generating process is formally written as:

$$\mathbf{x}^{V_i} = \{f^k(\mathbf{z}^k) : k \in V_i, \mathbf{z}^k \in [\mathbf{z}]_{\sim_{\iota_i}}\} = F([\mathbf{z}]_{\sim_{\iota_i}}).$$

**Remark:** Intuitively, Asm. 2.1 ensures that for each invariance property $\iota_i \in \mathfrak{I}$, there are at least two observables generated from latents that share $\iota_i$; otherwise the invariance partition $A_i$ becomes undefined and no identification results can be derived. While $\mathfrak{I}$ does not need to be fully described with explicit forms, the assignment of observables to equivalence classes is known (denoted as $V_i \subseteq [K]$ for the invariance property $\iota_i \in \mathfrak{I}$). This is a standard assumption and is equivalent to knowing, e.g., two views are generated from partially overlapped latents (Yao et al., 2023).

> **Problem setting.** Given a set of observables $\mathcal{S}_\mathbf{x} \in \mathcal{X}$ satisfying Asm. 2.1, we show that we can simultaneously identify multiple invariant latent blocks $A_i$ under a set of weak assumptions. In an ideal scenario, if each individual latent component is represented as a single invariant block through individual invariance property $\iota_i \in \mathfrak{I}$, we can learn a fully disentangled representation and further identify the latent causal graph by additional technical assumptions.

## 3 IDENTIFIABILITY THEORY VIA THE INVARIANCE PRINCIPLE

**High-level overview.** This section presents a general theory for latent variable identification that brings together many identifiability results from existing CRL works, including multiview, interventional, temporal, and multi-task CRL. Our theory of latent variable identifiability, based on the invariance principle, consists of two key components: (1) ensuring the encoder's sufficiency, thereby obtaining an adequate representation of the original input for the desired task; (2) guaranteeing the learned representation to preserve known data symmetries as invariance properties. The sufficiency is often enforced by minimizing the reconstruction loss (Locatello et al., 2020; Ahuja et al., 2022b; Lippe et al., 2022b;a; Lachapelle et al., 2022) in auto-encoder based architecture, maximizing the log likelihood in normalizing flows or maximizing entropy (Zimmermann et al., 2021; von Kügelgen et al., 2021; Daunhawer et al., 2023; Yao et al., 2023) in self-supervised approaches. The invariance property in the learned representations is often enforced by minimizing some equivalence relation-induced regularizer (von Kügelgen et al., 2021; Yao et al., 2023; Lippe et al., 2022b; Zhang et al., 2024a) or by some iterative algorithm that provably ensures the invariance property on the output (Squires et al., 2023; Varici et al., 2024b). As a result, all invariant blocks $A_i, i \in [|\mathfrak{I}|]$ can be identified up to a mixing within the blocks while being disentangled from the rest. This type of identifiability is defined as *block-identifiability* (von Kügelgen et al., 2021) which we restate as follows:

**Definition 3.1** (Block-identifiability (von Kügelgen et al., 2021))**.** A subset $\mathbf{z}_A := \{\mathbf{z}_j\}_{j \in A}$ with $A \subseteq [N]$ of the latent variables is block-identified by an encoder $g : \mathbb{R}^D \to \mathbb{R}^N$ on the invariant subset $A$ if the learned representation $\hat{\mathbf{z}}_{\hat{A}} := [g(\mathbf{x})]_{\hat{A}}$ with $\hat{A} \subseteq [N], |A| = |\hat{A}|$ contains all and only information about the ground truth $\mathbf{z}_A$, i.e. $\hat{\mathbf{z}}_{\hat{A}} = h(\mathbf{z}_A)$ for some diffeomorphism $h : \mathbb{R}^{|A|} \to \mathbb{R}^{|A|}$.

> **Intuition**: Block-identifiability relaxes the standard notion of disentanglement (Locatello et al., 2020; Lachapelle et al., 2023; Fumero et al., 2021) by allowing the learned representation $\mathbf{z}_A$ to be an entangled block that still captures all the information of the true latent block $\mathbf{z}_A$ up to a diffeomorphism. In this framework, classical disentanglement corresponds to the special case where each latent variable is identified as an individual invariant block.

**Definition 3.2** (Encoders)**.** The encoders $G := \{g_k : \mathcal{X}^k \to \mathcal{Z}^k\}_{k \in [K]}$ consist of smooth functions mapping from the observational support $\mathcal{X}^k$ to the corresponding latent support $\mathcal{Z}^k$ (§ 2).

**Intuition**: For the purpose of generality, we design the encoder $g_k$ to be specific to individual observable $\mathbf{x}^k \in \mathcal{S}_{\mathbf{x}}$. However, multiple $g_k$ can share parameters if they work on the same modality. Ideally, the encoders should preserve as many invariances (from $\mathfrak{I}$) as possible. Thus, a clear separation between different encoding blocks is needed. To this end, we introduce selectors.

**Definition 3.3** (Selection (Yao et al., 2023)). A selection $\oslash$ operates between two vectors $a \in \{0,1\}^d, b \in \mathbb{R}^d$ where $a \oslash b := [b_j : a_j = 1, j \in [d]]$.

**Definition 3.4** (Invariant block selectors). The invariant block selectors $\Phi := \{\phi^{(i,k)}\}_{i \in [|\mathfrak{I}|], k \in V_i}$ with $\phi^{(i,k)} \in \{0,1\}^N$ perform selection (Defn. 3.3) on the encoded information $\hat{\mathbf{z}}^k$: for any invariance property $\iota_i \in \mathfrak{I}$, any observable $\mathbf{x}^k, k \in V_i$ we have the selected representation:

$$\phi^{(i,k)} \oslash \hat{\mathbf{z}}^k = \phi^{(i,k)} \oslash g_k(\mathbf{x}^k) = \left[[g_k(\mathbf{x}^k)]_j : \phi_j^{(i,k)} = 1, j \in [N]\right], \qquad (3.1)$$

with $\|\phi^{(i,k)}\|_0 = \|\phi^{(i,k')}\|_0 = |A_i|$ for all $\iota_i \in \mathfrak{I}, k, k' \in V_i$.

**Intuition**: Selectors select the relevant encoding dimensions for each invariance property $\iota_i \in \mathfrak{I}$. Each selector $\phi^{(i,k)}$ implies a index subset $\hat{A}_i^k := \{j : \phi_j^{(i,k)} = 1\} \subseteq [N]$ that is specific to the invariance property $\iota_i$ and the observable $\mathbf{x}^k$. The assumption of known invariance size $|A_i|$ can be lifted in certain scenarios by, e.g., enforcing sharing between the learned latent variables, as shown by Fumero et al. (2024); Yao et al. (2023), or leveraging sparsity constraints (Lachapelle et al., 2022; 2024; Zheng et al., 2022; Xu et al., 2024).

**Constraint 3.1** (Invariance constraint). *For any invariance property $\iota_i \in \mathfrak{I}, i \in [|\mathfrak{I}|]$, the **selected** representations $\phi^{(i,k)} \oslash g_k(\mathbf{x}^k), k \in V_i$ must be $\iota_i$-invariant across the observables from the subset $V_i \subseteq [K]$:*

$$\iota_i(\phi^{(i,k)} \oslash g_k(\mathbf{x}^k)) = \iota_i(\phi^{(i,k')} \oslash g_{k'}(\mathbf{x}^{k'})) \quad \forall i \in [|\mathfrak{I}|] \; \forall k, k' \in V_i \qquad (3.2)$$

**Constraint 3.2** (Sufficiency constraint). *For any $\iota_i \in \mathfrak{I}, i \in [|\mathfrak{I}|]$, the **selected** representation $\phi^{(i,k)} \oslash g_k(\mathbf{x}^k), k \in V_i$ must preserve all information of the invariant partition $\mathbf{z}_{A_i}$ that we aim to identify, i.e., $I(\mathbf{z}_{A_i}, \phi^{(i,k)} \oslash g_k(\mathbf{x}^k)) = H(\mathbf{z}_{A_i}) \forall i \in [|\mathfrak{I}|], k \in V_i$, where $I(\cdot, \cdot)$ denotes the mutual information and $H(\cdot)$ denotes the differential entropy of the ground truth latent distribution $p_{\mathbf{z}_{A_i}}$.*

**Intuition:** The regularizer enforcing this sufficiency constraint can be tailored to suit the specific task of interest. For example, for self-supervised training, it can be implemented as the mutual information between the input data and the encodings, i.e., $I(\mathbf{x}, g(\mathbf{x})) = H(\mathbf{x})$, to preserve the entropy from the observations; for classification, it becomes the mutual information between the task labels and the learned representation $I(\mathbf{y}, g(\mathbf{x}))$. Sometimes, sufficiency does not have to be enforced on the whole representation. For example, in the multiview line of work (von Kügelgen et al., 2021; Daunhawer et al., 2023), when considering a single invariant block $A$, enforcing sufficiency on the shared partition (implemented as entropy on the learned encoding $H(g(\mathbf{x})_{1:|A|})$) is enough to block-identify these shared latent variables $\mathbf{z}_A$.

**Theorem 3.1** (Identifiability of multiple invariant blocks). *Consider a set of observables $\mathcal{S}_{\mathbf{x}} = \{\mathbf{x}^1, \mathbf{x}^2, \ldots, \mathbf{x}^K\} \in \mathcal{X}$ generated from § 2 satisfying Asm. 2.1. Let $G, \Phi$ be the set of smooth encoders (Defn. 3.2) and selectors (Defn. 3.4) that satisfy Constraints 3.1 and 3.2, then the invariant component $\mathbf{z}_{A_i}^k$ is block-identified (Defn. 3.1) by $\phi^{(i,k)} \oslash g_k$ for all $\iota_i \in \mathfrak{I}, k \in [K]$.*

**Discussion:** Thm. 3.1 demonstrates that by enforcing all invariance properties $\iota_i \in \mathfrak{I}$ jointly, our framework can simultaneously learn representations that block-identify all invariant latent blocks. This unified approach accommodates multiple invariance principles, making it well-suited for complex real-world scenarios where diverse invariance relations coexist. In practice, the resulting constrained optimization problem admits various solution strategies. For instance, Lippe et al. (2022b;a) adopt a two-stage process: first addressing the sufficiency constraint and then the invariance constraint, while Lachapelle et al. (2023); Fumero et al. (2024) frame the problem as a bi-level constrained optimization task. Some works (von Kügelgen et al., 2021; Yao et al., 2023; Daunhawer et al., 2023; Zhang et al., 2024a; Ahuja et al., 2024) propose

loss functions that directly enforce these constraints, whereas others (Squires et al., 2023; Varici et al., 2024a;b) develop iterative, step-by-step algorithms. This diversity of solution methods not only underscores the flexibility of our theoretical framework but also highlights its practical relevance across different CRL settings.

**What about the variant latents?** Intuitively, the variant latents are not identifiable, as the invariance constraint (Constraint 3.1) is applied only to the selected invariant encodings, leaving the variant part without any weak supervision (Locatello et al., 2019). This result is formalized as follows:

**Proposition 3.2** (General non-identifiability of variant latent variables). *Consider the setup in Thm. 3.1, let $A := \bigcup_{i \in [|\Im|]} A_i$ denote the union of block-identified latent indices and $A^{\mathrm{c}} := [N] \setminus A$ the complementary set where no $\iota$-invariance $\iota \in \Im$ applies, then the variant latents $\mathbf{z}_{A^{\mathrm{c}}}$ cannot be identified.*

Although variant latent variables are generally non-identifiable, they can be identified under certain conditions. The following demonstrates that variant latent variables can be identified under invertible encoders when the variant and invariant partitions are *mutually independent*.

**Proposition 3.3** (Identifiability of variant latent under independence). *Consider an optimal encoder $g \in G^*$ and optimal selector $\phi \in \Phi^*$ from Thm. 3.1 that jointly identify an invariant block $\mathbf{z}_A$ (we omit subscriptions $k, i$ for simplicity), then $\mathbf{z}_{A^{\mathrm{c}}}(A^{\mathrm{c}} := [N] \setminus A)$ can be identified by the complementary encoding partition $(1 - \phi) \oslash g$ only if*

*(i) $g$ is invertible in the sense that $I(\mathbf{x}, g(\mathbf{x})) = H(\mathbf{x})$;*

*(ii) $\mathbf{z}_{A^{\mathrm{c}}}$ is independent on $\mathbf{z}_A$.*

**Discussion:** The generalization of new interventions in CRL can be viewed through two distinct layers. The first layer involves generalizing to unseen interventional values, where the model encounters novel combinations of intervention settings; this has been demonstrated in several existing works (Zhang et al., 2024a; von Kügelgen et al., 2024). The second layer concerns generalization to unseen nodes, which, as Proposition 3.2 shows, is fundamentally challenging without additional conditions. However, under weak assumptions such as the independence of variant and invariant latent partitions and sufficient latent representation for reconstruction, non-intervened nodes in the training phase can be identified during inference (Proposition 3.3). This observation aligns with the identifiability algebra described in (Yao et al., 2023) and is supported by findings in disentanglement literature (Locatello et al., 2020; Fumero et al., 2024) as well as in temporal CRL (Lippe et al., 2022b; Lachapelle et al., 2024). Overall, this result emphasizes the need for carefully considering the conditions under which variant latents can be identified, and it delineates the practical limitations of CRL methods when generalizing to unseen nodes.

## 4 RELATED WORKS AS SPECIAL CASES OF OUR THEORY

This section provides an overview of the literature on CRL, including multiview, interventional, temporal, and multi-task settings, as well as domain generalization. We explain the underlying invariance principles and show how they naturally fit into our framework as special cases. Tab. 1 lists concrete examples and the explicit forms of their underlying invariance, and further mathematical details are deferred to Appendix App. D.

**Multiview CRL.** Multiview CRL (also considered as "counterfactual" CRL) considers a setting where each view (observable $\mathbf{x}^k$) is generated from a subset of latent causal variables (Locatello et al., 2020; Ahuja et al., 2022b; von Kügelgen et al., 2021; Daunhawer et al., 2023; Yao et al., 2023). Given any set of jointly observed views, the view-specific generating latents could overlap, giving rise to **sample level invariance** on all realizations of these shared latents. The common theoretical contribution in this line of work in terms of identifiability is that the invariant partition of latents (shared ones) can be block-identified by enforcing aligned and sufficient representation, which is a special case of Thm. 3.1 with specified sample invariance.

**Interventional CRL.** Interventional (also termed multi-environment) CRL (Ahuja et al., 2023; Squires et al., 2023; Zhang et al., 2024a; Buchholz et al., 2024; Varici et al., 2023; 2024a; von Kügelgen et al., 2021; Wendong et al., 2024) collects data from multiple environments that follow different data distributions, often originated from interventions ($\mathbf{x}^k \sim P^k$). Current interventional

CRL literature has provided fruitful identifiability results based on various types of interventions: either atomic or paired interventions per node or different parametric assumptions on the mixing function or the latent causal model. Interventions give rise to many types of invariance: When performing an atomic intervention on an arbitrary node, the ***marginal*** of its non-descendants remain invariant; the ***score*** of all other nodes than its parents and itself also remain invariant. By utilizing these two types of invariance, we can not only explain various prior identification theories as special cases of Thm. 3.1, but also directly develop new element-wise identification results on the latent variables, given *imperfect* atomic interventions per node (Cor. D.1). Some other works (von Kügelgen et al., 2021; Varici et al., 2024a) consider paired interventions per node, with an ***invariant interventional target*** between these paired interventional environments. This invariance imposes a certain score structure in the latent space, which can be used as the invariant constraint (Constraint 3.1). More details in this regard are provided in App. D.2. More recently, Ahuja et al. (2024) explains previous interventional identifiability results from a general weak distributional invariance perspective. Ahuja et al. (2024) proves block-affine identification (Defn. C.1) by additionally assuming the mixing function to be finite degree polynomial, which can be explained by Proposition C.2 together with our block-identifiability results under the general nonparametric setting. They consider *one* single invariance set, which is a special case of Thm. 3.1 with one joint $\iota$-property. Another line of interventional CRL work (Zhang et al., 2024a) employs an orthogonal proof technique, originating from nonlinear ICA with auxiliary variables (Hyvarinen et al., 2019). We remark that our framework does not directly include this line of identifiability theory.

**Temporal CRL**. Extending CRL into time-series setting, temporal CRL often assumes an "intervenable" trajectory in the latent space (Lippe et al., 2022a;b; 2023; Lachapelle et al., 2022; Yao et al., 2022b;a; Li et al., 2024b). At each time step, an intervention/action modifies the dynamics of a subset of latent variables, with the remaining invariant partition following the default dynamics conditioning on the previous time step. Existing works have shown that the intervened part can be disentangled from the invariant part when there is no causal link between the latent causal variables at the same time step (Lachapelle et al., 2022; Lippe et al., 2022a;b). Comparing the "counterfactual" latent with the actual partially intervened latents on the same time step, one observes the ***transitional distribution*** (current latents conditioning on previous latents) remain invariant for the non-intervened partition (see Tab. 1 for concrete examples). This formulates an explicit $\iota$-property (Defn. 2.1) for each time step with potentially different invariant partitions, explaining many existing temporal CRL identifiability theories by incorporating Thm. 3.1.

**Multi-task CRL** In supervised CRL, latent variables (Lachapelle et al., 2023; Fumero et al., 2024) are shown to be identifiable under multi-task setting, meaning there are multiple task labels available for each observable ($\mathbf{x}^k := (\mathbf{x}, \mathbf{y}^k)$). The key criterion for achieving identifiability is ***overlapping task support***, i.e., a set of tasks depends on a shared set of latents. This shared set of latents constitutes the invariant partition $\mathbf{z}_A$, as illustrated in Tab. 1. Incorporating this invariance principle into Thm. 3.1 explains the identification results of (Lachapelle et al., 2023; Fumero et al., 2024), showing the overlapping task support can be identified.

**Domain Generalization.** The field of domain generalization focuses on the out-of-distribution performance of the learned representation instead of the theoretical identifiability guarantee (Rojas-Carulla et al., 2018; Arjovsky et al., 2020; Ahuja et al., 2022a; Krueger et al., 2021; Sagawa et al., 2019). The goal is to learn representations that perform equally well across domains originating from distributional shifts, such as covariates shift or concept shift. Domain generalization typically assumes the same downstream prediction task, and this task depends on the same subset of latent factors $A$ across all domains. Given the same ground truth task-latent dependency, the ***domain risk*** w.r.t. ground truth inverting process remains invariant across all domains. This invariance property together with Thm. 3.1 could provide theoretical insights for domain generalization works such as (Krueger et al., 2021; Sagawa et al., 2019) (formal mathematical derivation provided in (f)).

## 5 EXPERIMENTS

This section illustrates the expanded applicability of CRL algorithms under the invariance principle. § 5.1 shows improved treatment effect estimation on the high-dimensional causal inference benchmark (Cadei et al., 2024) by enforcing the invariance principle through existing domain generalization techniques (Krueger et al., 2021). This result underscores the practical utility of our unified approach. Additionally, § 5.2 provides ablation studies on existing interventional CRL meth-

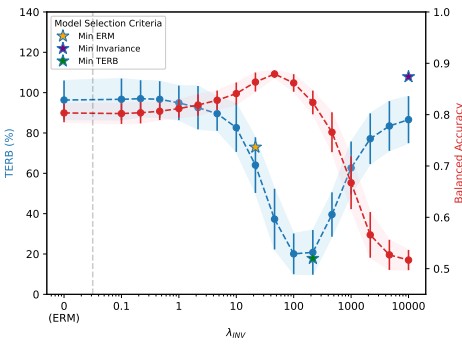 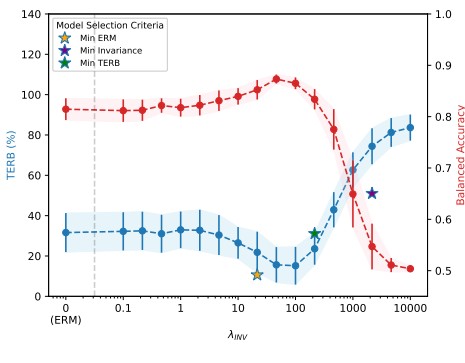

(a) Experiment Sampling    (b) Position Sampling

Figure 1: TERB and Balanced Accuracy with standard deviation over 20 different seeds varying the invariance weight $\lambda_{\text{INV}}$ of V-REx (Krueger et al., 2021) on ISTAnt dataset (Cadei et al., 2024). Stars represent the selected best models based on a small but heterogeneous validation set.

ods (Ahuja et al., 2023; Zhang et al., 2024a), showcasing that the non-trivial distributional invariance required for latent variable identification can arise from non-causal assumptions.

## 5.1 CASE STUDY: ISTANT

This experiment focuses on ISTAnt (Cadei et al., 2024), a recent real-world ecological benchmark designed for treatment effect estimation. ISTAnt consists of video recordings of ants triplets with occasional grooming behavior. The goal is to extract a per-frame representation for supervised behavior classification (grooming or not) to estimate the Average Treatment Effect of an intervention (exposure to a chemical substance). Further details about the problem setting are provided in App. F.1.

**Experiment settings.** Different videos in ISTAnt are considered different *experiments* as the experiment settings and treatments vary. We consider hard annotation sampling criteria (more non-annotated than annotated) for both experiments (videos) and positions, as described by Cadei et al. (2024). For the training, we adopt a domain generalization objective that utilizes the invariance principle (Krueger et al., 2021), which is restated as follows:

$$\mathcal{R}_{\text{V-REx}}(\mathbf{w} \circ g) = \underbrace{\lambda_{\text{INV}} \operatorname{Var}(\{\mathcal{R}_1(\mathbf{w} \circ g), \ldots, \mathcal{R}_K(\mathbf{w} \circ g)\})}_{\text{invariance}} + \underbrace{\sum_{k \in [K]} \mathcal{R}_k(\mathbf{w} \circ g)}_{\text{sufficiency}}, \qquad (5.1)$$

we provide a detailed derivation in (f) showing the invariance term above is indeed enforcing risk invariance. We vary the strength of the invariant component in eq. (5.1) by setting the regularization multiplier $\lambda_{\text{INV}}$ from 0 (ERM) to 10 000. We repeat 20 independent runs for each $\lambda_{\text{INV}}$ to estimate the statistical error. Further implementational details are deferred to App. F.1. We evaluate the performance with both *balanced accuracy* and *Treatment Effect Relative Bias* (TERB). TERB is defined by Cadei et al. (2024) as the ratio between the bias in the predictions across treatment groups and the true average treatment effect estimated with ground-truth annotations over the whole trial.

**Results.** Fig. 1 depicts the model performance regarding varying invariance regularization strength $\lambda_{\text{INV}}$. Consistent with our expectation, the balanced accuracy initially increases with the $\lambda_{\text{INV}}$, as adequate invariance enforces identifying task-related latents, thus benefiting the prediction problem. At a later point, the performance decreases because the sufficiency component is not correctly balanced with the invariance. Similarly, the TERB improves positively, weighting the invariance component until a certain threshold. On average, with $\lambda_{\text{INV}} = 100$ the TERB decreases to 20% (from 100% using ERM) with experiment subsampling. In agreement with (Cadei et al., 2024), a naive estimate of the TERB on a small validation set is a reasonable (albeit not perfect) model selection criterion. Although it performs slightly worse than model selection based on *Empirical Risk Minimization*(ERM) loss in the position sampling case, it shows more reliability overall. This experiment underscores the advantages of flexibly enforcing known invariances in the data, corroborating our identifiability theory (§ 3).

## 5.2 SYNTHETIC ABLATION WITH "NINTERVENTIONS"

This subsection presents identifiability results under non-causal conditions using simulated data. We consider a simple graph of three causal variables as $\mathbf{z}_1 \rightarrow \mathbf{z}_2 \rightarrow \mathbf{z}_3$. The corresponding joint density

has the form of

$$p_{\mathbf{z}}(z_1, z_2, z_3) = p(z_3 \mid z_2)p(z_2 \mid z_1)p(z_1).$$

This experiment aims at demonstrating that existing methods of interventional CRL rely primarily on distributional invariance, regardless of whether this invariance arises from a well-defined intervention or some other arbitrary transformation. To illustrate this, we introduce the concept of a "nintervention," which has a similar distributional effect to a regular intervention, maintaining certain conditionals invariant while altering others, but without a causal interpretation.

**Definition 5.1** (Nintervention)**.** We define a "*nintervention*" on a causal conditional as the process of changing its distribution but cutting all *incoming and outgoing* edges. Child nodes condition on the old, pre-intervention, random variable. Formally, we consider the latent SCM as defined in Defn. B.1, an *nintervention* on a node $j \in [N]$ gives rise to the following conditional factorization

$$\tilde{p}_{\mathbf{z}}(z) = \tilde{p}(z_j) \prod_{i \in [N] \setminus \{j\}} p(z_i \mid z^{\text{old}}_{\text{pa}(i)})$$

Note that the marginal distribution of all non-nintervened nodes $P_{\mathbf{z}_{[N] \setminus j}}$ remain invariant after nintervention. In previous example, we perform a nintervention by replacing the conditional density $p(z_2 \mid z_1)$ using a sufficiently different marginal distribution $\tilde{p}(z_2)$ that satisfies Defn. 2.1 (ii), which gives rise to the following new factorization $\tilde{p}_{\mathbf{z}}(z_1, z_2, z_3) = p(z_3 \mid z^{\text{old}}_2)\tilde{p}(z_2)p(z_1)$. Note that $\mathbf{z}_3$ conditions on the random variable $\mathbf{z}_2$ before nintervention, whose realization is denoted as $z^{\text{old}}_2$. Differing from a causal *intervention*, we cut both the incoming and outgoing links of $\mathbf{z}_2$ and keep the marginal distribution of $\mathbf{z}_3$ the same. Clearly, this is a non-sensical intervention from the causal perspective because we eliminate the causal effect from $\mathbf{z}_2$ to its descendants.

**Experiment settings.** As a proof of concept, we choose a linear Gaussian additive noise model and a nonlinear mixing function implemented as a 3-layer invertible MLP. We average the results over three independently sampled *ninterventional* densities $\tilde{p}(z_2)$ while guaranteeing all *ninterventional* distributions satisfy Defn. 2.1 (ii). As the marginal distribution of both $\mathbf{z}_1, \mathbf{z}_3$ remains the same after a *nintervention*, we expect $\mathbf{z}_1, \mathbf{z}_3$ to be block-identified (Defn. 3.1) according to Thm. 3.1. In practice, we enforce the marginal invariance constraint (Constraint 3.1) by minimizing the MMD loss, as implemented by the interventional CRL works (Zhang et al., 2024a; Ahuja et al., 2024) and train an auto-encoder for a sufficient representation (Constraint 3.2). Further details are included in App. F.2.

**Results.** To validate block-identifiability, we perform Kernel-Ridge Regression between the estimated block $[\hat{\mathbf{z}}_1, \hat{\mathbf{z}}_3]$ and the ground truth latents $\mathbf{z}_1, \mathbf{z}_2, \mathbf{z}_3$. Both $\mathbf{z}_1$ and $\mathbf{z}_3$ are block-identified with high $R^2$ scores of $0.863 \pm 0.031$ and $0.872 \pm 0.035$. In contrast, $\mathbf{z}_2$ is not identified, with a low $R^2$ of $0.065 \pm 0.017$, indicating identification is driven by the underlying distributional invariance.

## 6 CONCLUSION

In this paper, we examined a broad range of CRL methods and found that many of them share common strategies for aligning representations with known data symmetries. We identified two key components in achieving identifiability: preserving data information and enforcing a set of known invariances (see § 3). Our work clarifies the role of causal assumptions in latent variable identification, shifting the focus from specific, often impractical, assumptions to a general recipe that enables practitioners to specify and leverage known invariances in their problems. Following this recipe, we exemplified the practical impact of our approach on ecological data (§ 5.1). This paper leaves out settings involving discrete variables and finite sample guarantees, which might be interesting for future work.

### ETHICS STATEMENT

This work unifies many existing theoretical results in CRL, thus vastly broadening its real-world applicability. As the paper is predominantly theoretical, we believe it poses no immediate ethical risks.

### REPRODUCIBILITY STATEMENT

All proofs in this paper are deferred to App. E. The ISTAnt dataset in § 5.1 is published by (Cadei et al., 2024). Results provided in § 5 can be reproduced following the details given in App. F. Since our primary focus is on the theoretical unification of latent variable identification algorithms, the practical implementation of the invariance and sufficiency constraints may take various forms (as illustrated in Tab. 4) and should be tailored to the specific problem at hand. For a brief example of the ecology experiment (§ 5.1), please visit: https://github.com/CausalLearningAI/ISTAnt/blob/main/experiments/invariance.ipynb.

ACKNOWLEDGEMENTS

We thank Jiaqi Zhang, Francesco Montagna, David Lopez-Paz, Kartik Ahuja, Thomas Kipf, Sara Magliacane, Julius von Kügelgen, Kun Zhang, and Bernhard Schölkopf for extremely helpful discussion. Riccardo Cadei was supported by a Google Research Scholar Award to Francesco Locatello. We acknowledge the Third Bellairs Workshop on Causal Representation Learning held at the Bellairs Research Institute, February 9/16, 2024, and a debate on the difference between interventions and counterfactuals in disentanglement and CRL that took place during Dhanya Sridhar's lecture, which motivated us to significantly broaden the scope of the paper. We thank Dhanya and all participants of the workshop.

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

# Appendix

## Table of Contents

## A  NOTATION AND TERMINOLOGY

This section provides a glossary of symbols and notations used throughout the paper.

| | |
|---|---|
| $f$ | Mixing function |
| $g$ | Smooth encoder |
| $\mathbf{x}$ | Entangled observables |
| $\mathbf{z}$ | Ground truth latent variables |
| $D$ | Dimensionality of observable $\mathbf{x}$ |
| $N$ | Dimensionality of latents $\mathbf{z}$ |
| $\mathcal{S}_{\mathbf{x}}$ | A set of observables |
| $\mathcal{S}_{\mathbf{z}}$ | A set of latent vectors |
| $A$ | Subset of latent indices with invariance properties ($A \subseteq [N]$) |
| $\iota$ | Invariance property |

$\sim_\iota$     The latent equivalence relation

$\mathfrak{I}$     A set of invariance properties

$\mathcal{X}$     Support of a set of observables $\mathcal{S}_{\mathbf{x}}$

$\mathcal{Z}$     Support of a set of latent vectors $\mathcal{S}_{\mathbf{z}}$

$G$     A set of smooth encoders

$\Phi$     A set of selectors

$\mathcal{G}$     Ground truth causal graph

TC     Transitive closure

## B  PRELIMINARIES

In this subsection, we revisit the common definitions and assumptions in identifiability works from CRL that are needed for subsequent theoretical analysis. We begin with the definition of a latent structural causal model:

**Definition B.1** (Latent SCM (von Kügelgen et al., 2024)). Let $\mathbf{z} = \{\mathbf{z}_1, \ldots, \mathbf{z}_N\}$ denote a set of causal "endogenous" variables with each $\mathbf{z}_i$ taking values in $\mathbb{R}$, and let $\mathbf{u} = \{\mathbf{u}_1, \ldots, \mathbf{u}_N\}$ denotes a set of mutually independent "exogenous" random variables. The latent SCM consists of a set of structural equations

$$\{\mathbf{z}_i := m_i(\mathbf{z}_{\mathrm{pa}(i)}), \mathbf{u}_i\}_{i=1}^N, \tag{B.1}$$

where $\mathbf{z}_{\mathrm{pa}(i)}$ are the causal parents of $\mathbf{z}_i$ and $m_i$ are the deterministic functions that are termed "causal mechanisms". We indicate with $P_{\mathbf{u}}$ the joint distribution of the exogenous random variables, which, due to the independence hypothesis, is the product of the probability measures of the individual variables. The associated causal diagram $\mathcal{G}$ is a directed graph with vertices $\mathbf{z}$ and edges $\mathbf{z}_i \to \mathbf{z}_j$ iff. $\mathbf{z}_i \in \mathbf{z}_{\mathrm{pa}(j)}$; we assume the graph $\mathcal{G}$ to be acyclic.

The latent SCM induces a unique distribution $P_{\mathbf{z}}$ over the endogenous variables $\mathbf{z}$ as a pushforward of $P_{\mathbf{u}}$ via eq. (B.1). Its density $p_{\mathbf{z}}$ follows the causal Markov factorization:

$$p_{\mathbf{z}}(z) = \prod_{i=1}^N p_i(z_i \mid z_{\mathrm{pa}(i)}). \tag{B.2}$$

Instead of directly observing the endogenous and exogenous variables $\mathbf{z}$ and $\mathbf{u}$, we only have access to some "entangled" measurements $\mathbf{x}$ of $\mathbf{z}$ generated through a nonlinear mixing function:

**Definition B.2** (Mixing function). A deterministic smooth function $f : \mathbb{R}^N \to \mathbb{R}^D$ mapping the latent vector $\mathbf{z} \in \mathbb{R}^N$ to its observable $\mathbf{x} \in \mathbb{R}^D$, where $D \geq N$ denotes the dimensionality of the observational space.

**Assumption B.1** (Diffeomorphism). The mixing function $f$ is diffeomorphic onto its image, i.e. $f$ is $C^\infty$, $f$ is injective and $f^{-1}|_{\mathrm{Im}(f)} : \mathrm{Im}(f) \to \mathbb{R}^D$ is also $C^\infty$.

**Remark:** Settings with noisy observations ($\mathbf{x} = f(\mathbf{z}) + \epsilon$, $\mathbf{z} \perp \epsilon$) can be easily reduced to our denoised version by applying a standard deconvolution argument as a pre-processing step, as indicated by Lachapelle et al. (2022); Buchholz et al. (2024).

## C  IDENTIFIABILITY THEORY

In addition to the general results for latent variable identification presented in § 3, we compare in App. C.1 different granularity of latent variable identification and show their transitions through certain assumptions on the causal model or mixing function. Afterward, App. C.2 discusses the identification level of a causal graph depending on the granularity of latent variable identification under certain structural assumptions. Proofs are deferred to App. E.

Figure 2: Relations between different identification classes (Defns. 3.1 and C.1 to C.3). Some CRL works proposed a more fine-grained classification of identifiability concepts with slightly different terminology, which we omit here for readability.

## C.1 ON THE GRANULARITY OF LATENT VARIABLE IDENTIFICATION

Different levels of identification can be achieved depending on the degree of underlying invariance and data symmetry. Below, we present three standard identifiability definitions from the CRL literature, each providing a stronger identification result than block-identifiability (Defn. 3.1).

**Definition C.1** (Block affine-identifiability). Let $\hat{\mathbf{z}}$ be the learned representation, for a subset $A \subseteq [N]$ it satisfies that:

$$\hat{\mathbf{z}}_{\pi(A)} = D \cdot \mathbf{z}_A + \mathbf{b}, \tag{C.1}$$

where $D \in \mathbb{R}^{|A| \times |A|}$ is an invertible matrix, $\pi(A)$ denotes the index permutation of $A$, then $\mathbf{z}_A$ is block affine-identified by $\hat{\mathbf{z}}_{\pi(A)}$.

**Definition C.2** (Element-identifiability). The learned representation $\hat{\mathbf{z}} \in \mathbb{R}^N$ satisfies that:

$$\hat{\mathbf{z}} = \mathbf{P}_\pi \cdot h(\mathbf{z}), \tag{C.2}$$

where $\mathbf{P}_\pi \in \mathbb{R}^{N \times N}$ is a permutation matrix, $h(\mathbf{z}) := (h_1(\mathbf{z}_1), \dots h_N(\mathbf{z}_N)) \in \mathbb{R}^N$ is an element-wise diffeomorphism.

**Definition C.3** (Affine-identifiability). The learned representation $\hat{\mathbf{z}} \in \mathbb{R}^N$ satisfies that:

$$\hat{\mathbf{z}} = \Lambda \cdot \mathbf{P}_\pi \cdot \mathbf{z} + \mathbf{b}, \tag{C.3}$$

where $\mathbf{P}_\pi \in \mathbb{R}^{N \times N}$ is a permutation matrix, $\Lambda \in \mathbb{R}^{N \times N}$ is a diagonal matrix with nonzero diagonal entries.

> **Remark**: Block affine-identifiability (Defn. C.1) is defined by Ahuja et al. (2023), stating that a subset of the learned representation $\hat{\mathbf{z}}_{\pi(A)}$ is related to the ground truth partition $\mathbf{z}_A$ through some affine transformation. Defn. C.2 indicates element-wise identification of latent variables up to individual diffeomorphisms. Element-identifiability for the latent variable identification together with the graph identifiability (Defn. C.4) is defined as $\sim_{\text{CRL}}$-identifiability (von Kügelgen et al., 2024, Defn. 2.6), perfect identifiability (Varici et al., 2024a, Defn. 3). Affine identifiability (Defn. C.3) describes when the ground truth latent variables are identified up to permutation, shift, and linear scaling. In many CRL works, affine identifiability (Defn. C.3) is also termed as follows: perfect identifiability under linear transformation (Varici et al., 2024b, Defn. 1), CD-equivalence (Zhang et al., 2024a, Defn. 1), disentanglement (Lachapelle et al., 2022, Defn. 3).

**Proposition C.1** (Granularity of identification). *Affine-identifiability (Defn. C.3) implies element-identifiability (Defn. C.2) and block affine-identifiability (Defn. C.1) while element-identifiability and block affine-identifiability implies block-identifiability (Defn. 3.1).*

**Proposition C.2** (Transition between identification levels). *The transition between different levels of latent variable identification (Fig. 2) can be summarized as follows:*

  *(i) Element- identifiability (Defns. C.2 and C.3) can be obtained from block-wise identifiability (Defns. 3.1 and C.1) when each individual latent constitutes an invariant block;*

  *(ii) Identifiability up to an affine transformation (Defns. C.1 and C.3) can be obtained from general identifiability on arbitrary diffeomorphism (Defns. 3.1 and C.2) by additionally assuming that both the ground truth mixing function and decoder are finite degree polynomials of the same degree.*

**Discussion.** We note that the granularity of identifiability results is primarily determined by the strength of invariance and parametric assumptions (such as those on mixing functions or causal models) rather than by the specific algorithmic choice. For example, for settings that can achieve element-identifiability (von Kügelgen et al., 2024), affine-identifiability results can be obtained by additionally assuming *finite degree polynomial* mixing function (proof see App. E.4). Similarly, element-identifiability can be achieved from block-identifiability by enforcing invariance properties on each latent component (Yao et al., 2023, Thm. 3.8) instead of having only *one* multivariate invariant block (von Kügelgen et al., 2021). In summary, existing CRL algorithms are capable of achieving different identifiability definitions depending on the additional (e.g., parametric) assumptions without requiring separate proofs for each case. Tab. 4 provides an overview of recent identifiability results along with their corresponding invariance and parametric assumptions, illustrating the direct relationship between these assumptions and the level of identifiability they achieve.

### C.2 IDENTIFYING THE CAUSAL GRAPH

In addition to latent variable identification, another goal of CRL is to infer the underlying latent dependency, namely the causal graph structure. Revisiting the literature on causal graph identification highlights a key distinction: While graph discovery often depends on causal assumptions like interventions or graphical constraints, identifying causal variables can proceed by leveraging only the invariance relations without requiring these additional assumptions, e.g., distributional invariance that does not necessarily arise from valid interventions. We begin with restating the standard definition of graph identifiability in CRL.

**Definition C.4** (Graph-identfiability)**.** The estimated graph $\hat{\mathcal{G}}$ is isomorphic to the ground truth graph $\mathcal{G}$ through a bijection $h : V(\mathcal{G}) \to V(\hat{\mathcal{G}})$ in the sense that two vertices $\mathbf{z}_i, \mathbf{z}_j \in V(\mathcal{G})$ are adjacent in $\mathcal{G}$ if and only if $h(\mathbf{z}_i), h(\mathbf{z}_j) \in V(\hat{\mathcal{G}})$ are adjacent in $\hat{\mathcal{G}}$.

We remark that the "faithfulness" assumption (Pearl, 2009, Defn. 2.4.1) is a standard assumption in the CRL literature, commonly required for graph discovery. We restate it as follows:

**Assumption C.1** (Faithfulness (or Stability))**.** $P_{\mathbf{z}}$ is a faithful distribution induced by the latent SCM (Defn. B.1) in the sense that $P_{\mathbf{z}}$ contains no extraneous conditional independence; in other words, the only conditional independence relations satisfied by $P_{\mathbf{z}}$ are those given by $\{\mathbf{z}_i \perp \mathbf{z}_{\mathrm{nd}(i)} \mid \mathbf{z}_{\mathrm{pa}(i)}\}$ where $\mathbf{z}_{\mathrm{nd}(i)}$ denotes the non-descends of $\mathbf{z}_i$.

As indicated by Defn. C.4, the preliminary condition of identifying the causal graph is to have an element-wise correspondence between the vertices in the ground truth graph $\mathcal{G}$ (i.e., the ground truth latents) and the vertices of the estimated graph. Therefore, the following assumes that the learned encoders $G$ (Defn. 3.2) achieve element-identifiability (Defn. C.2), that is, for each $\mathbf{z}_i \in \mathbf{z}$, we have a diffeomorphism $h_i : \mathbb{R} \to \mathbb{R}$ such that $\hat{\mathbf{z}}_i = h_i(\mathbf{z}_i)$. However, additional assumptions are needed to identify the graph structure: either on the source of invariance or on the parametric form of the latent causal model.

**Graph identification via interventions.** Under the element-identifiability (Defn. C.2) of the latent variables $\mathbf{z}$, the causal graph structure $\mathcal{G}$ can be identified up to its isomorphism (Defn. C.4), given multi-environment data from *paired perfect* interventions per-node (von Kügelgen et al., 2024; Varici et al., 2024a). Using data generated from *imperfect* interventions is generally insufficient to identify the direct edges in the causal graph. It can only identify the ancestral relations, i.e., up to the transitive closure of $\mathcal{G}$ (Brehmer et al., 2022; Zhang et al., 2024a). Unfortunately, even imposing the linear assumption on the latent SCM does not provide a solution (Squires et al., 2023). Nevertheless, by adding sparsity assumptions on the causal graph $\mathcal{G}$ and polynomial assumption on the mixing function $f$, Zhang et al. (2024a) has shown isomorphic graph identifiability (Defn. C.4) under *imperfect* intervention per node. In general, access to the interventions is necessary for graph identification if alternative parametric assumptions are not imposed. Conveniently, in this setting, the graph identifiability is linked with that of the variables since the latter leverages the invariance induced by the intervention.

**Graph identification via parametric assumptions.** In this work, we focus exclusively on the post-nonlinear additive noise model (Zhang & Hyvärinen, 2010, Sec. 2) because it provides a sufficiently general framework that subsumes other parametric instances (such as the standard additive noise

and the location-scale models) while offering greater flexibility in modeling complex causal mechanisms.

**Definition C.5** (Post-nonlinear acyclic causal model)**.** The following causal mechanism describes a post-nonlinear acyclic causal model:

$$\mathbf{z}_i = \gamma_i(m_i(\mathbf{z}_{\text{pa}(i)}) + \mathbf{u}_i), \tag{C.4}$$

where $\gamma_i : \mathbb{R} \to \mathbb{R}$ is a diffeomorphism and $m_i$ is a non-constant causal mechanism, and $\mathbf{u}_i$ is an exogenous noise term.

Note that this model reduces to the standard additive noise model (Hoyer et al., 2008) with $\gamma_i = \text{id}$ and to the location-scale form when $\gamma_i$ is affine (i.e., $\gamma_i(x) = \Lambda_i x + \beta_i$ with an invertible matrix $\Lambda_i$ and bias $\beta_i$).

We assume that the ground truth latent SCM (Defn. B.1) is a post-nonlinear acyclic causal model as specified in Defn. C.5. Given that each latent variable $\mathbf{z}_i$ is element-wise identified via a diffeomorphism $h_i : \mathbb{R} \to \mathbb{R}$ for all $i \in [N]$, we define the estimated causal parents as

$$\hat{\mathbf{z}}_{\text{pa}(i)} := \{\hat{\mathbf{z}}_j : \mathbf{z}_j \in \mathbf{z}_{\text{pa}(i)}\}.$$

It then follows that the learned representations $\hat{\mathbf{z}}_i$ also obey a postnonlinear acyclic model:

$$
\begin{aligned}
\hat{\mathbf{z}}_i = h_i(\mathbf{z}_i) &= h_i\Big(\gamma_i\big(m_i(\mathbf{z}_{\text{pa}(i)}) + \mathbf{u}_i\big)\Big) \\
&= h_i\Big(\gamma_i\big(m_i(\{h_j^{-1}(\hat{\mathbf{z}}_j) : \mathbf{z}_j \in \mathbf{z}_{\text{pa}(i)}\}) + \mathbf{u}_i\big)\Big) \\
&= h_i\Big(\gamma_i\big(\tilde{m}_i(\hat{\mathbf{z}}_{\text{pa}(i)}) + \mathbf{u}_i\big)\Big) \\
&= \big(h_i \circ \gamma_i\big)\Big(\tilde{m}_i(\hat{\mathbf{z}}_{\text{pa}(i)}) + \mathbf{u}_i\Big),
\end{aligned}
\tag{C.5}
$$

where we define

$$\tilde{m}_i(\hat{\mathbf{z}}_{\text{pa}(i)}) := m_i\Big(\{h_j^{-1}(\hat{\mathbf{z}}_j) : \mathbf{z}_j \in \mathbf{z}_{\text{pa}(i)}\Big).$$

Since the composition $h_i \circ \gamma_i$ is a diffeomorphism, the conditional dependencies among the latents are preserved. Thus, following the approach in Zhang & Hyvärinen (2009, Sec. 4), the underlying causal graph $\mathcal{G}$ can be identified up to an isomorphism as defined in Defn. C.4.

**What happens if variables are identified in blocks?** Consider the case where the latent variables cannot be identified up to element-wise diffeomorphism; instead, one can only obtain a coarse-grained version of the variables (e.g., as a mixing of a block of variables (Defn. 3.1)). Nevertheless, certain causal links between these coarse-grained block variables are of interest. These block variables and their causal relations in between form a "macro" level of the original latent SCM, which is shown to be causally consistent under mild structural assumptions (Rubenstein et al., 2017, Thm. 11). In particular, the macro-level model can be obtained from the micro-level model through an *exact transformation* (Beckers & Halpern, 2019, Defn. 3.4) and thus produces the same causal effect as the original micro-level model under the same type of interventions, providing useful knowledge for downstream causal analysis. More formal connections are beyond the scope of this paper. Still, we see this concept of coarse-grained identification on both causal variables and graphs as an interesting avenue for future research.

# D RELATED WORKS

This section reviews related CRL and domain generalization works and frames them as specific instances of our theory (§ 3). These CRL works were initially categorized into various types (multiview, interventional, multi-task, and temporal CRL) based on the level of invariance in the data-generating process, leading to varying degrees of identifiability results (App. C.1). While the implementation of individual works may vary, the *methodological principle of aligning representation with known data symmetries* remains consistent, as shown in § 3. We begin with revisiting the data-generating process of each category and explain how they can be viewed as specific cases of the proposed invariance framework (§ 2). We then present individual identification algorithms from existing literature as particular applications of our theorems based on the implementation choices needed to satisfy the invariance and sufficiency constraints (Constraints 3.1 and 3.2). A more detailed overview of the individual works is provided in Tab. 4.

### D.1 MULTIVIEW CRL

**High-level overview.** The multiview setting in CRL (Daunhawer et al., 2023; Yao et al., 2023) considers multiple observables that are *concurrently* generated by an overlapping subset of latent variables. Multiview scenarios are often found in a partially observable setup. For example, multiple devices on a robot measure different modalities, jointly monitoring the environment through these real-time measurements. While each device measures a distinct subset of latent variables, these subsets probably still overlap as they are measuring the same system at the same time. In addition to partial observability, another way to obtain multiple views is to perform an "intervention/perturbation" (Locatello et al., 2020; von Kügelgen et al., 2021; Ahuja et al., 2022b; Brehmer et al., 2022) and collect both pre-action and post-action views on the same sample. This setting is often improperly termed "counterfactual"[1] in the CRL literature, and this type of data is termed "paired data". From another perspective, the paired setting can be cast in the partial observability scenario by considering the same latent before and after an action (mathematically modeled as an intervention) as two separate latent nodes in the causal graph, as shown by von Kügelgen et al. (2021, Fig. 1). Thus, both pre-action and post-action views are partial because neither of them can observe pre-action and post-action latents simultaneously. These works assume the latents that are not affected by the action remain constant, an assumption that is relaxed in temporal CRL works. See App. D.3 for more discussion on temporal CRL.

**Data generating process.** In the following, we introduce the data-generating process of a multiview setting in the flavor of the invariance principle as introduced in § 2. We consider a set of views $\{\mathbf{x}^k\}_{k \in [K]}$ with each view $\mathbf{x}^k \in \mathcal{X}^k$ generated from some latents $\mathbf{z}^k \in \mathcal{Z}^k$. Let $S_k \subseteq [N]$ be the index set of generating factors for the view $\mathbf{x}^k$, we define $\mathbf{z}_j^k = 0$ for all $j \in [N] \setminus S_k$ to represent the uninvolved partition of latents. Each entangled view $\mathbf{x}^k$ is generated by a view-specific mixing function $f_k : \mathcal{Z}^k \to \mathcal{X}^k$:

$$\mathbf{x}^k = f_k(\mathbf{z}^k) \quad \forall k \in [K] \tag{D.1}$$

Define the joint overlapping index set $A := \bigcap_{k \in [K]} S_k$, and assume $A \subseteq [N]$ is a non-empty subset of $[N]$. Then the value of the sharing partition $\mathbf{z}_A$ remain invariant for all observables $\{\mathbf{x}^k\}_{k \in [K]}$ on a *sample level*. By considering the joint intersection $A$, we have *one single* invariance property $\iota : \mathbb{R}^{|A|} \to \mathbb{R}^{|A|}$ in the invariance set $\mathfrak{I}$; and this invariance property $\iota$ emerges as the identity map id on $\mathbb{R}^{|A|}$ in the sense that $\mathrm{id}(\mathbf{z}_A^k) = \mathrm{id}(\mathbf{z}_A^{k'})$ and thus $\mathbf{z}_A^k \sim_\iota \mathbf{z}_A^{k'}$ for all $k, k' \in [K]$. Note that Defn. 2.1 (ii) is satisfied because any transformation $h_k$ that involves other components $\mathbf{z}_q$ with $q \notin A$ violates the equality introduced by the identity map. For a subset of observations $V_i \subseteq [K]$ with at least two elements $|V_i| > 1$, we define the latent intersection as $A_i := \bigcap_{k \in V_i} S_k \subseteq [N]$, then for each non-empty intersection $A_i$, there is a corresponding invariance property $\iota_i : \mathbb{R}^{|A_i|} \to \mathbb{R}^{|A_i|}$ which is the identity map specified on the subspace $\mathbb{R}^{|A_i|}$. By considering all these subsets $\mathcal{V} := \{V_i \subseteq [K] : |V_i| > 1, |A_i| > 0\}$, we obtain a set of invariance properties $\mathfrak{I} := \{\iota_i : \mathbb{R}^{|A_i|} \to \mathbb{R}^{|A_i|}\}$ that satisfy Asm. 2.1.

**Identification algorithms.** Many multiview works (von Kügelgen et al., 2021; Daunhawer et al., 2023; Yao et al., 2023) employ the $L_2$ loss as a regularizer to enforce **sample-level** invariance on the invariant partition, cooperated with some sufficiency regularizer to preserve sufficient information about the observables (Constraint 3.2). Aligned with our theory (Thm. 3.1), these works have shown block-identifiability on the invariant partition of the latents across different views. Following the same principle, there are certain variations in the implementations to enforce the invariance principle, e.g. Locatello et al. (2020) directly average the learned representations from paired data $g(\mathbf{x}^1), g(\mathbf{x}^2)$ on the shared coordinates before forwarding them to the decoder; Ahuja et al. (2022b) enforces $L_2$ alignment up to a learnable sparse perturbation $\delta$. As each latent component constitutes a single invariant block in the training data, these two works *element-identifies* (Defn. C.2) the latent variables, as explained by Proposition C.2.

---

[1]Traditionally, counterfactual in causality refers to non-observable outcomes that are "counter to the fact" (Rubin, 2005). The works we refer to here represent pre- and post-actions that affect some latent variables but not all. This can be mathematically expressed as a counterfactual in an SCM but is conceptually different as both pre- and post-action outcomes are realized (Liu et al., 2023). The "counterfactual" terminology silently implies that this is a strong assumption, but nuance is needed and it can in fact be much weaker than an intervention.

### D.2 INTERVENTIONAL CRL

**High-level overview.** Interventional/Multi-environment CRL considers data generated from multiple environments with different data distributions. In the scope of CRL, multi-environment data is often instantiated through interventions on the latent structured causal model (von Kügelgen et al., 2021; Zhang et al., 2024a; Buchholz et al., 2024; Squires et al., 2023; Varici et al., 2023; 2024b;a). Recently, Ahuja et al. (2024) provides a more general identifiability statement where multi-environment data is not necessarily originated from interventions; instead, they can be individual data distributions that preserve certain symmetries such as marginal invariance or support invariance.

**Data generating process** The following presents the data generating process described in most interventional CRL works. Formally, we consider a set of observables $\{P_{\mathbf{x}^k}\}_{k \in [K]}$ that are collected from multiple environments (indexed by $k \in [K]$) with a shared latent SCM (Defn. B.1) and a shared mixing function $f : \mathbf{x}^k = f(\mathbf{z}^k)$ (Defn. B.2) satisfying Asm. B.1. Let $k = 0$ denote the non-intervened environment and $\mathcal{I}_k \subseteq [N]$ denotes the set of intervened nodes in $k$-th environment, the latent distribution $P_{\mathbf{z}^k}$ is associated with the density

$$p_{\mathbf{z}^k}(z^k) = \prod_{j \in \mathcal{I}_k} \tilde{p}(z_j^k \mid z_{\mathrm{pa}(j)}^k) \prod_{j \in [N] \setminus \mathcal{I}_k} p(z_j^k \mid z_{\mathrm{pa}(j)}^k), \tag{D.2}$$

where we denote by $p$ the original density and by $\tilde{p}$ the intervened density. Interventions naturally introduce various distributional invariances that can be utilized for latent variable identification: Under the intervention $\mathcal{I}_k$ in the $k$-th environment, we observe that both (1) the marginal distribution of $\mathbf{z}_A$ with $A := [N] \setminus \mathrm{TC}(\mathcal{I}_k)$, with TC denoting the transitive closure and (2) the score $[S(\mathbf{z}^k)]_{A'} := \nabla_{\mathbf{z}_{A'}^k} \log p_{\mathbf{z}^k}$ on the subset of latent components $A' := [N] \setminus \overline{\mathrm{pa}}(\mathcal{I}_k)$ with $\overline{\mathrm{pa}}(\mathcal{I}_k) := \{j : j \in \mathcal{I}_k \cup \mathrm{pa}(\mathcal{I}_k)\}$ remain invariant across the observational and the $k$-th interventional environment. Formally, under intervention $\mathcal{I}_k$, we have

- *Marginal invariance*:

$$p_{\mathbf{z}^0}(z_A^0) = p_{\mathbf{z}^k}(z_A^k) \qquad A := [N] \setminus \mathrm{TC}(\mathcal{I}_k); \tag{D.3}$$

- *Score invariance*:

$$[S(\mathbf{z}^0)]_{A'} = [S(\mathbf{z}^k)]_{A'} \qquad A' := [N] \setminus \overline{\mathrm{pa}}(\mathcal{I}_k). \tag{D.4}$$

According to our theory Thm. 3.1, we can block-identify both $\mathbf{z}_A, \mathbf{z}_{A'}$ using these invariance principles (eqs. (D.3) and (D.4)). Since most interventional CRL works assume at least one intervention per node (Squires et al., 2023; Zhang et al., 2024a; von Kügelgen et al., 2024; Varici et al., 2024a; 2023; Buchholz et al., 2024; Ahuja et al., 2023), more fine-grained variable identification results, such as element-wise identification (Defn. C.2) or affine-identification (Defn. C.3), can be achieved by combining multiple invariances from these per-node interventions, as we elaborate below.

**Identifiability with one intervention per node.** By invoking Thm. 3.1, we establish that, in non-parametric settings, latent causal variables $\mathbf{z}$ can be identified up to an element-wise diffeomorphism (see Defn. C.2) using single-node *imperfect* interventions for each node. This result addresses an open conjecture posed by von Kügelgen et al. (2024). We assume:

**Assumption D.1** (Topologically ordered interventional targets). Specifying Asm. 2.1 in the interventional setting, we assume there are exactly $N$ environments $\{k_1, \ldots, k_N\} \subseteq [K]$ where each node $j \in [N]$ undergoes one imperfect intervention in the environment $k_j \in [K]$. The interventional targets $1 \preceq \cdots \preceq N$ preserve the topological order, meaning that $i \preceq j$ only if there is a directed path from node $i$ to node $j$ in the underlying causal graph $\mathcal{G}$.

**Remark:** Asm. D.1 is directly implied by Asm. 2.1 as we need to know which environments fall into the same equivalence class. We believe that identifying the topological order is another subproblem orthogonal to identifying the latent variables, which is often termed "uncoupled/non-aligned problem" (Varici et al., 2024a; von Kügelgen et al., 2024). As described by Zhang et al. (2024a), the topological order of unknown interventional targets can be recovered from single-node imperfect intervention by iteratively identifying the interventions that target the source nodes. This iterative identification process may require additional assumptions on the mixing functions (Zhang et al., 2024a; Ahuja et al., 2023; Varici et al., 2023; 2024b; Squires et al., 2023) and the latent structured causal model (Buchholz et al., 2024; Squires et al., 2023), or on the interventions, such *paired perfect* interventions per node (von Kügelgen et al., 2024; Varici et al., 2024a).

**Corollary D.1** (Identifiability from single node imperfect intervention per node). *Given $N$ environments $\{k_1, \ldots, k_N\} \subseteq [K]$ satisfying Asm. D.1, the ground truth latent variables $\mathbf{z}$ can be identified up to element-wise diffeomorphism (Defn. C.2) by combining both marginal and score invariances (eqs. (D.3) and (D.4)) under our framework (Thm. 3.1).*

The proof for Cor. D.1 is included in App. E.5. Upon element-wise identification from single-node intervention per node, existing works often provide more fine-grained identifiability results by incorporating other parametric assumptions on the mixing functions (Varici et al., 2023; Ahuja et al., 2023; Zhang et al., 2024a; Squires et al., 2023). This perspective is elaborated in Proposition C.2, as element-wise identification can be refined to affine-identification (Defn. C.3) given additional parametric assumptions on the mixing functions. However, note that under the milder setting of *imperfect* intervention per node, the full graph is not identifiable without further assumptions. See (Zhang et al., 2024a) for more details.

**Identifiability with two interventions per node** Current literature in interventional CRL targeting the general nonparametric setting (Varici et al., 2024a; von Kügelgen et al., 2024) typically assumed a pair of *sufficiently different* perfect interventions per node. Thus, any latent variable $\mathbf{z}_j, j \in [N]$, as an interventional target, is uniquely shared by a pair of interventional environment $k, k' \in [K]$, forming an invariant partition $A_i = \{j\}$ constituting of individual latent node $j \in [N]$. Formally, we write

$$\mathcal{I}_k = \mathcal{I}_{k'} = A_i = \{j\} \tag{D.5}$$

where $\mathcal{I}_k$ represent the interventional target for the $k$-th environment. Note that this invariance property implies the following distributional property:

$$[S(\mathbf{z}^k) - S(\mathbf{z}^{k'})]_j \neq 0 \qquad \text{only if} \qquad \mathcal{I}_k = \mathcal{I}_{k'} = \{j\}. \tag{D.6}$$

According to Thm. 3.1, each latent variable can thus be identified separately, giving rise to element-wise identification, as shown by (Varici et al., 2024a; von Kügelgen et al., 2024).

**Identifiability under multiple distributions.** More recently, Ahuja et al. (2024) explains previous interventional identifiability results from a general weak distributional invariance perspective. In a nutshell, a set of variables $\mathbf{z}_A$ can be block-identified if certain invariant distributional properties hold: The invariant partition $\mathbf{z}_A$ can be block-identified (Defn. 3.1) from the rest by utilizing the *marginal distributional invariance* or *invariance on the support, mean or variance*. Ahuja et al. (2024) additionally assume the mixing function to be finite degree polynomial, which leads to block-affine identification (Defn. C.1), whereas we can also consider a general nonparametric setting; they consider *one* single invariance set, which is a special case of Thm. 3.1 with one joint $\iota$-property.

**Identification algorithms.** Instead of iteratively enforcing the invariance constraint across the majority of environments as described in Cor. D.1, most single-node interventional works develop equivalent constraints between pairs of environments to optimize. For example, the marginal invariance (eq. (D.3)) implies the marginal of the source node is changed *only if* it is intervened upon, which is utilized by Zhang et al. (2024a) to identify latent variables and the ancestral relations simultaneously. In practice, Zhang et al. (2024a) propose a regularized loss that includes Maximum Mean Discrepancy(MMD) between the reconstructed "counterfactual" data distribution and the interventional distribution, enforcing the distributional discrepancy that reveals graphical structure (e.g., detecting the source node). Similarly, by enforcing sparsity on the score change matrix, Varici et al. (2023) restricts only score changes from the intervened node and its parents. In the nonparametric case, von Kügelgen et al. (2024) optimize for the invariant (aligned) interventional targets through model selection, whereas Varici et al. (2024a) directly solve the constrained optimization problem formulated using score differences. Considering a more general setup, Ahuja et al. (2024) provides various invariance-based regularizers as plug-and-play components for any losses that enforce a sufficient representation (Constraint 3.2).

## D.3 TEMPORAL CRL

**High-level overview.** Temporal CRL (Lippe et al., 2022a; 2023; 2022b; Yao et al., 2022a;b; Lachapelle et al., 2022; 2024; Li et al., 2024a;b) focuses on retrieving latent causal structures from time series data, where the latent causal structure is typically modeled as a Dynamic Bayesian Network (DBN) (Dean & Kanazawa, 1989; Murphy, 2002). Existing temporal CRL literature has developed identifiability results under varying sets of assumptions. A common overarching assumption is to require the Dynamic Bayesian Network to be first-order Markovian, allowing only causal links

from $t-1$ to $t$, eliminating longer dependencies (Lippe et al., 2022b; 2023; 2022a; Yao et al., 2022b). While many works assume that there is no instantaneous effect, restricting the latent components of $\mathbf{z}^t$ to be mutually dependent (Lippe et al., 2022b; Yao et al., 2022b; Lippe et al., 2023), some approaches have lifted this assumption and prove identifiability allowing for instantaneous links among the latent components at the same timestep (Lippe et al. (2022a)).

**Data generating process.** We present the data generating process followed by most temporal causal representation works and explain the underlying latent invariance and data symmetries. Let $\mathbf{z}^t \in \mathbb{R}^N$ denotes the latent vector at time $t$ and $\mathbf{x}^t = f(\mathbf{z}^t) \in \mathbb{R}^D$ the corresponding entangled observable with $f : \mathbb{R}^N \to \mathbb{R}^D$ the shared mixing function (Defn. B.2) satisfying Asm. B.1. The actions $\mathbf{a}^t$ with cardinality $|\mathbf{a}^t| = N$ mostly only target a subset of latent variables while keeping the rest untouched, following its default dynamics (Lippe et al., 2022b; 2023; Lachapelle et al., 2022; 2024). Intuitively, these actions $\mathbf{a}^t$ can be interpreted as a component-wise indicator for each latent variable $\mathbf{z}^t_j, j \in [N]$ stating whether $\mathbf{z}^t_j$ follows the default dynamics $p(\mathbf{z}^t_j \mid \mathbf{z}^{t-1})$ or the modified dynamics induced by the action $\mathbf{a}^t_j$. From this perspective, the non-intervened causal variables at time $t$ can be considered the invariant partition under our formulation, denoted by $\mathbf{z}^t_{A_t}$ with the index set $A_t$ defined as $A_t := \{j : \mathbf{a}_j = 0\}$. Note that this invariance can be considered as a generalization of the multiview case because the realizations $z^t_j, z^{t-1}_j$ are not exactly identical (as in the multiview case) but are related via a default transition mechanism $p(\mathbf{z}^t_j \mid \mathbf{z}^{t-1})$. To formalize this intuition, we define $\tilde{\mathbf{z}}^t := \mathbf{z}^t \mid \mathbf{a}^t$ as the conditional random vector conditioning on the action $\mathbf{a}^t$ at time $t$. For the non-intervened partition $A_t \subseteq [N]$ that follows the default dynamics, the transition model should be invariant:

$$p(\mathbf{z}^t_{A_t} \mid \mathbf{z}^{t-1}) = p(\tilde{\mathbf{z}}^t_{A_t} \mid \mathbf{z}^{t-1}), \tag{D.7}$$

which gives rise to a non-trivial distributional invariance property (Defn. 2.1). Note that the invariance partition $A_t$ could vary across different time steps, providing a set of invariance properties $\mathfrak{I} := \{\iota_t : \mathbb{R}^{|A_t|} \to \mathcal{M}_t\}_{t=1}^T$, indexed by time $t$. Given by Thm. 3.1, all invariant partitions $\mathbf{z}^t_{A_t}$ can be block-identified; furthermore, as shown in Proposition 3.3, the complementary variant partition can also be identified under an invertible encoder and mutual independence within $\mathbf{z}^t$ (here conditioning on the previous time step $\mathbf{z}^{t-1}$). This result aligns with the identification results without instantaneous effect, i.e. there is no causal link between variables at the same time step (Lippe et al., 2022b; Yao et al., 2022b; Lachapelle et al., 2022; 2024). On the other hand, temporal causal variables with instantaneous effects are shown to be identifiable *only if* "instantaneous parents" (i.e., nodes affecting other nodes instantaneously) are cut by actions (Lippe et al., 2022a), reducing to the setting without instantaneous effect where the latent components at $t$ are mutually independent. Upon invariance, more fine-grained latent variable identification results, such as element-wise identifiability, can be obtained by incorporating additional technical assumptions, such as the sparse mechanism shift (Lachapelle et al., 2022; 2024; Li et al., 2024b) and parametric latent causal model (Yao et al., 2022b; Klindt et al., 2021; Khemakhem et al., 2020).

**Identification algorithms.** From a high level, the distributional invariance (eq. (D.7)) indicates full explainability and predictability of $\mathbf{z}^t_{A_t}$ from its previous time step $\mathbf{z}^{t-1}$, regardless of the action $\mathbf{a}^t$. In principle, this invariance principle can be enforced by directly maximizing the information content of the proposed default transition density between the learned representation $p(\hat{\mathbf{z}}^t_{A_t} \mid \hat{\mathbf{z}}^{t-1})$ (Lippe et al., 2022a;b). In practice, the invariance regularization is often incorporated together with the predictability of the variant partition conditioning on actions, implemented as a KL divergence between the observational posterior $q(\hat{\mathbf{z}}^t \mid \mathbf{x}^t)$ and the transitional prior $p(\hat{\mathbf{z}}^t \mid \hat{\mathbf{z}}^{t-1}, \mathbf{a}^t)$ (Lachapelle et al., 2022; 2024; Klindt et al., 2021; Yao et al., 2022a;b; Lippe et al., 2023), estimated using variational Bayes (Kingma & Welling, 2013) or normalizing flow (Rezende & Mohamed, 2015).

### D.4 MULTI-TASK CRL

**High-level overview.** Multi-task CRL aims to identify latent causal variables via external supervision, in this case, the label information of the same instance for various tasks. Previously, multi-task learning (Caruana, 1997; Zhang & Yang, 2018) has been mostly studied outside the scope of identifiability, mainly focusing on domain adaptation and out-of-distribution generalization. One of the popular ideas that was extensively used in the context of multi-task learning is to leverage interactions between different tasks to construct a generalist model that is capable of solving all classification tasks and potentially better generalizes to unseen tasks (Zhu et al., 2022; Bai et al.,

2022). Recently, Lachapelle et al. (2023); Fumero et al. (2024) systematically studied under which conditions the latent variables can be identified in the multi-task scenario and correspondingly provided identification algorithms.

**Data generating process.** Multi-task CRL considers a *supervised* setup: Given a latent SCM as defined in Defn. B.1, we generate the observable $\mathbf{x} \in \mathbb{R}^D$ through some mixing function $f : \mathbb{R}^N \to \mathbb{R}^D$ satisfying Asm. B.1. Consider a set of task $\mathcal{T} = \{T_1, \ldots, T_k\}$ with corresponding task labels $\mathbf{y}^k \in \mathcal{Y}_k$, we assume each task only depends on a subset of latent variables $S_k \subseteq [N]$. In other words, the label $\mathbf{y}^k$ can be expressed as a function that contains all and only information about the latent variable $\mathbf{z}_{S_k}$:

$$\mathbf{y}^k = r_k(\mathbf{z}_{S_k}), \tag{D.8}$$

where $r : \mathbb{R}^{|S_k|} \to \mathcal{Y}_k$ is some deterministic function which maps the latent subspace $\mathbb{R}^{|S_k|}$ to the task-specific label space $\mathcal{Y}_k$, which is often assumed to be linear and implemented using a linear readout in practice (Lachapelle et al., 2023; Fumero et al., 2024). For each task $t_k, k \in [K]$, we observe the associated data distribution $P_{\mathbf{x}, \mathbf{y}^k}$. Consider two different tasks $T_k, T_{k'}$ with $k, k' \in [K]$, the corresponding data $\mathbf{x}, \mathbf{y}^k$ and $\mathbf{x}, \mathbf{y}^{k'}$ are invariant in the intersection of task-related features $\mathbf{z}_A$ with $A = S_k \cap S_{k'}$. To ease the notation, let $\mathbf{z}^{T_k} := \mathbf{z}_{S_k}$ represent the task-related latents for task $T_k$. Formally, it holds that

$$\mathbf{z}_A^{T_k} = \mathbf{z}_A^{T_{k'}}, \tag{D.9}$$

showing alignment on the shared partition of the task-related latents. In the ideal case, each latent component $j \in [N]$ is *uniquely shared* by a subset of tasks, all factors of variation can be fully disentangled, which aligns with the theoretical claims by Lachapelle et al. (2023); Fumero et al. (2024).

**Identification algorithms.** We remark that the *sharing* mechanism in the context of multi-task learning fundamentally differs from that of multiview setup, thus resulting in different learning algorithms. Regarding learning, the shared partition of task-related latents is enforced to align up to the linear equivalence class (given a linear readout) instead of sample level $L_2$ alignment. Intuitively, this invariance principle can be interpreted as a soft version of the that in the multiview case. In practice, under the constraint of perfect classification, one employs (1) a sparsity constraint on the linear readout weights to enforce the encoder to allocate the correct task-specific latents and (2) an information-sharing term to encourage reusing latents across various tasks. Equilibrium can be obtained between these two terms only when the shared task-specific latent is element-wise identified (Defn. C.2). Thus, this soft invariance principle is jointly implemented by the sparsity constraint and information sharing regularization (Fumero et al., 2024, Sec. 2.1).

## D.5 DOMAIN GENERALIZATION

**High-level overview.** Domain generalization aims at *out-of-distribution* performance. That is, learning an optimal encoder and predictor that performs well at some unseen test domain that preserves the same data symmetries as in the training data. At a high level, domain generalization (Sagawa et al., 2019; Zhang et al., 2017; Ganin et al., 2016; Arjovsky et al., 2020; Krueger et al., 2021) considers a similar framework as introduced for interventional CRL, i.e., having access to multiple environment with different data distributions, but additionally incorporated with external supervision and focusing more on model robustness perspective. While interventional CRL aims to identify the true latent factors of variations (up to some transformation), domain generalization learning focuses directly on *out-of-distribution* prediction, relying on some invariance properties preserved under the distributional shifts. Due to the non-causal objective, new methodologies are motivated and tested on real-world benchmarks (e.g., VLCS (Fang et al., 2013), PACS (Li et al., 2017), Office-Home (Venkateswara et al., 2017), Terra Incognita (Beery et al., 2018), DomainNet (Peng et al., 2019)) and could inspire future real-world applicability of CRL approaches.

**Data generating process.** The problem of domain generalizations is an *extension of supervised learning* where training data from multiple environments are available (Blanchard et al., 2011). An environment is a dataset of i.i.d. observations from a joint distribution $P_{\mathbf{x}^k, \mathbf{y}^k}$ of the observables $\mathbf{x}^k \in \mathbb{R}^D$ and the label $\mathbf{y}^k \in \mathbb{R}$. The label $\mathbf{y}^k \in \mathbb{R}^m$ only depends on the invariant latents $\mathbf{z}_A^k \in \mathbb{R}^{|A|}$ through a linear regression structural equation model (Ahuja et al., 2022a, Assmp. 1), described as follows:

$$\begin{aligned} \mathbf{y}^k &= \mathbf{w}^* \mathbf{z}_A^k + \epsilon_k, \ \mathbf{z}_A^k \perp \epsilon_k \\ \mathbf{x}^k &= f(\mathbf{z}^k) \end{aligned} \tag{D.10}$$

where $\mathbf{w}^* \in \mathbb{R}^{D \times m}$ represents the ground truth relationship between the label $\mathbf{y}^k$ and the invariant latents $\mathbf{z}_A^k$. $\epsilon_k$ is some white noise with bounded variance and $f : \mathbb{R}^N \to \mathbb{R}^D$ denotes the shared mixing function for all $k \in [K]$ satisfying Asm. B.1. The set of environment distributions $\{P_{\mathbf{x}^k, \mathbf{y}^k}\}_{k \in [K]}$ generally differ from each other because of interventions or other distributional shifts such as covariates shift and concept shift. However, as the relationship between the invariant latents and the labels $\mathbf{w}^*$ and the mixing mechanism $f$ are shared across different environments, the risk on optimal weights remain invariant:

$$\mathcal{R}_k^*(\mathbf{w}^* \mathbf{z}_A^k, \mathbf{y}^k) = \mathcal{R}_{k'}^*(\mathbf{w}^* \mathbf{z}_A^{k'}, \mathbf{y}^{k'}) \tag{D.11}$$

where $\mathbf{w}^*$ denotes the ground truth relation between the invariant latents $\mathbf{z}_A^k$ and the labels $\mathbf{y}^k$.

**Identification algorithms.** Different distributional invariance are enforced by interpolating and extrapolating across various environments. Among the countless contribution to the literature, *mixup* (Zhang et al., 2017) linearly interpolates observations from different environments as a robust data augmentation procedure, Domain-Adversarial Neural Networks (Ganin et al., 2016) support the main learning task discouraging learning domain-discriminant features, Distributionally Robust Optimization (DRO) (Sagawa et al., 2019) replaces the vanilla Empirical Risk objective minimizing only with respect to the worst modeled environment, Invariant Risk Minimization (Arjovsky et al., 2020) combines the Empirical Risk objective with an invariance constraint on the gradient, and Variance Risk Extrapolation (Krueger et al., 2021, V-REx), similar in spirit combines the empirical risk objective with an invariance constraint using the variance among environments. For a more comprehensive review of domain generalization algorithms, see Zhou et al. (2022).

### D.6 Further Explanations for Tab. 4

**General clarification.** Tab. 4 summarizes special cases of our invariance framework. For each work, we present their technical assumptions, the type of invariance, the implementation for the invariance and the sufficiency regularizers (to satisfy Constraints 3.1 and 3.2), and the type of identifiability they achieve. Note that this table is by no means exhaustive. Also, we omit some additional results and technical assumptions of individual papers for readability. A list of paragraphs is provided below for further clarification, as referenced in Tab. 4.

**(a) Single-node intervention and parametric assumptions.** Many existing CRL works that consider single node intervention per node require additional parametric assumptions, either on the mixing function (Varici et al., 2023; Zhang et al., 2024a) or the latent causal model (Buchholz et al., 2024) or both (Squires et al., 2023), thus achieving (at least) element-wise identifiability (Defn. C.2). We conjecture these additional parametric assumptions serve two purposes: (1) to identify valid topological order of the interventional targets, as required by Asm. D.1 for Cor. D.1 (2) to get a more fine-grained identification level of affine transformation, as explained by Proposition C.2.

In the following, we restate the definition of linear latent SCM for reference:

**Definition D.1** (Linear latent SCM (Squires et al., 2023; Buchholz et al., 2024)). The latent variables $\mathbf{z}$ follows a linear SCM with Gaussian noise in the sense that

$$\mathbf{z} = A\mathbf{z} + \Gamma^{1/2}\epsilon, \tag{D.12}$$

where $\Gamma$ is a diagonal matrix with positive entries, $A$ encodes the underlying causal graph $G$ and the $\epsilon$ is the standard Gaussian noise. For the sake of simplicity, we often define $B := \Gamma^{-1/2}(\mathrm{Id} - A)$ such that $\mathbf{z} = B^{-1}\epsilon$ to explicitly map from the exogenous noise $\epsilon$ to the latent variables $\mathbf{z}$. We use $B_k$ to denote this matrix for the domain $k$.

**(b) Multi-node intervention and linear mixing.** Recently, Varici et al. (2024b) extends previous interventional CRL works to unknown multi-node interventions and achieves identifiability under the assumption of a linearly independent intervention signature matrix $M_{\mathrm{int}} \in \{0,1\}^{N \times K}$ with each column $k$ represents the intervened node in this environment $k$. The row-wise linear independence of $M_{\mathrm{int}}$ implies that each latent variable must have been intervened at least once. Let $M \in \{0,1\}^{N \times N}$ represent a submatrix of $M_{\mathrm{int}}$ with *linearly independent* columns. By multiplying $M$ with its adjoint transpose $\mathrm{adj}^\intercal(M)$, one obtains a matrix where each column has only one non-zero component. Applying the same transformation to the score change, this problem is reduced to a similar setting as a single node intervention per node, which can be intuitively explained using the same distributional invariance principle introduced earlier (App. D.2).

**(c) Paired single-node intervention per node under nonparametric assumptions.** In the nonparametric settings, several works (von Kügelgen et al., 2024; Varici et al., 2024a) have shown element-wise latent variable identification under sufficiently different paired perfect intervention per node. By having two sufficiently different interventions per node, one introduces invariance on the interventional target across these paired interventional environments. This invariance property can be enforced using the score differences (Varici et al., 2024a) or algorithmically by performing model selection (von Kügelgen et al., 2024), as elaborated in App. D.2.

**(d) Variant latents identification under independence.** While some papers states main identification results on the variant partition, it can be explained by Thm. 3.1 and Proposition 3.3 stating that the variant block can be identified under independence and invertible encoder. For example, Wendong et al. (2024, Thm. 4.5) shows block-identifiability on the intervened (variant) latents under (Wendong et al., 2024, Assumption 4.4) of block-wise independence between the invariant and variant blocks.

**(e) Invariance regularizers in multitask CRL.** Under the assumption of knowing the number of latent variables, Lachapelle et al. (2023) solves a bi-level optimization problem, enforcing $L_{2,1}$ sparsity on individual task readouts in the inner problem. Coupled with a backbone shared across all tasks, this implicitly encourages discovering the ground truth overlapping partition of task support. Fumero et al. (2024) lifted the constraint of assuming the known number of latents by incorporating an additional information-sharing regularizer, as explained in (Fumero et al., 2024, Sec. 2.1).

**(f) Invariance regularizers in domain generalization.** While Sagawa et al. (2019) directly optimize for the worst-case risk, a link can be drawn between this objective and the risk invariance: Given a pair of linear head $\mathbf{w}$ and encoder $g$ shared across $[K]$ domains, let the order of risks be $\mathcal{R}^{\pi_1} \geq \mathcal{R}^{\pi_2} \ldots \mathcal{R}^{\pi_K}$. Since $\mathcal{R}^{\pi_1}$ is lower bounded by $\mathcal{R}^{\pi_2}$, the minimum of the training objective in Sagawa et al. (2019) ($\max_{k \in [K]} \mathcal{R}^k(w,g)$) is obtained when $\mathcal{R}^{\pi_1} = \mathcal{R}^{\pi_2}$. Then we have $\mathcal{R}^{\pi_1} = \mathcal{R}^{\pi_2} \geq \cdots \geq \mathcal{R}^{\pi_K}$, and the next minimum will be obtained when $\mathcal{R}^{\pi_1} = \mathcal{R}^{\pi_2} = \mathcal{R}^{\pi_3}$, and so on so forth. The optimization procedure stops when the risks are equally minimized across all domains.

(Krueger et al., 2021) minimizes variance between domain risks to enforce the risk invariance. We formally show these two are equivalent in the following. Note that the invariance principle for risk alignment can be formulated as

$$(\mathcal{R}_k - \mathcal{R}_{k'})^2 \tag{D.13}$$

According to Zhang et al. (2012), variance can be equivalently expressed as pair-wise distances between the samples. Hence, we can reformulate the risk variance term in (Sagawa et al., 2019) as follows:

$$\mathrm{Var}\left[\mathcal{R}\right] = \frac{1}{K^2} \sum_{k,k' \in [K]} \frac{1}{2} \left(\mathcal{R}_k - \mathcal{R}_{k'}\right)^2,$$

showing that the variance regularization in (Krueger et al., 2021) enforces risk invariance.

## D.7 NOTABLE CASES NOT DIRECTLY COVERED BY THE THEORY

Some works not listed in Tab. 4 cannot yet be directly explained by our invariance frameworks but are rather loosely connected. One representative line of work (Lachapelle et al., 2022; Zheng et al., 2022; Xu et al., 2024; Lachapelle et al., 2024) relies on the sparsity assumption in the latent dependency to achieve latent variable and graph identification. This assumption is closely related to the *sparse mechanism shift* hypothesis in CRL (Schölkopf et al., 2021), stating small distributional changes should not affect all causal variables but only a small subset of these. Note that the sparsity constraint is often formulated as the estimator (either for the graph (Lachapelle et al., 2023; 2024) or of the latents (Xu et al., 2024)) should be at least sparse as the ground truth one, maximizing the cardinality of the unaffected (invariant) part. Some theoretical results do not rely on multiple data pockets that share certain invariance properties but directly employ specific properties within the observational data, such as independent support (Ahuja et al., 2023), or shared cluster membership (Khemakhem et al., 2020; Kivva et al., 2022). Some works (Zhang et al., 2024b) follow an orthogonal proof technique originating from the *nonlinear ICA with auxiliary variable* line of work (Hyvarinen et al., 2019). Their proofs often rely on linear independence derived from the statistical diversity of various underlying data distributions instead of shared invariance properties. Our framework thus does not trivially include them.

# E   PROOFS

This section includes formal proofs for the theoretical statements of the paper.

## E.1   ASSUMPTION JUSTIFICATION

We justify the Defn. 2.1 (ii) by showing negative results under violation of this assumption, i.e., trivially invariant latent variables are not identifiable.

**Proposition E.1** (General non-identifiability of trivially invariant latent variables). *Consider the setup in Thm. 3.1, w.l.o.g we assume $\mathfrak{I} = \{\iota\}$ and $\iota$ is trivial in the sense that assumption (ii) in Defn. 2.1 is violated. Then, the corresponding invariant partition $\mathbf{z}_A^k$ is not identifiable for any $k \in [K]$.*

*Proof.* We provide a counter example as follows: Define a trivial $\iota$-property as "if the first component is greater than zero on $A = \{1\}$ of some two dimensional latents $\mathbf{z}$". Formally,

$$\iota(\mathbf{z}_1) = \mathbf{1}[\mathbf{z}_1 > 0].$$

Consider a mixing function $f = id$ and an invertible encoder $g(\mathbf{x}) = g(f(\mathbf{z})) = [\mathbf{z}_1 + \mathbf{z}_2, \mathbf{z}_2]$ satisfying the sufficiency constraint (Constraint 3.2). Define $h_1 = h_2 = [g \circ f]_A$. Then for some realizations $z, \tilde{z}$ with $z_1 + z_2 > 0$ and $\tilde{z}_1 + \tilde{z}_2 > 0$ we have $\iota(h(\mathbf{z})) = \iota(h(\tilde{\mathbf{z}}))$. However, $h_1, h_2$ can not disentangle $\mathbf{z}_1$, showing non-identifiability for the invariant partition $\mathbf{z}_A$. $\square$

**Link between Defn. 2.1 (ii) and interventional discrepancy.** In the following, we elaborate how Defn. 2.1 (ii) resembles the most common assumption in interventional CRL, the interventional discrepancy (Wendong et al., 2024; Varici et al., 2024a). Note that this assumption may termed differently as *sufficient variability* (von Kügelgen et al., 2024; Lippe et al., 2022b), *interventional regularity* (Varici et al., 2023; 2024b), but the mathematical formulation remain the same. We begin with restating this assumption:

**Assumption E.1** (Interventional discrepancy (Wendong et al., 2024)). Given $k \in [K]$, let $p_{t_k}$ denote the causal mechanism of the intervened variable $\mathbf{z}_{t_k}$ with $t_k \in [N]$. We say a stochastic intervention $\tilde{p}_k$ satisfies interventional discrepancy if

$$\frac{\partial \log p_{t_k}}{\partial \mathbf{z}_{t_k}}(\mathbf{z}_{t_k} \mid \mathbf{z}_{\mathrm{pa}(t_k)}) \neq \frac{\partial \log \tilde{p}_{t_k}}{\partial \mathbf{z}_{t_k}}(\mathbf{z}_{t_k} \mid \mathbf{z}_{\mathrm{pa}(t_k)}) \qquad \text{almost everywhere } (a.e.).$$

*Proof.* We show that any cases violating the interventional discrepancy assumption also violates Defn. 2.1 (ii) and vice versa. Suppose for a contradiction that there exists $t_k \in [N]$ that is intervened in environment $k \in [K]$, and there is a non-empty interior $U \subset \mathbb{R}$ with non-zero measure where the interventional discrepancy is violated, i.e., for all $z_{t_k} \in U$, it holds

$$\frac{\partial \log p_{t_k}}{\partial z_{t_k}}(\mathbf{z}_{t_k} \mid \mathbf{z}_{\mathrm{pa}(t_k)}) = \frac{\partial \log \tilde{p}_{t_k}}{\partial z_{t_k}}(\mathbf{z}_{t_k} \mid \mathbf{z}_{\mathrm{pa}(t_k)}) \tag{E.1}$$

Under a single node imperfect intervention, the complementary set of the transitive closure of $t_k$, i.e., $A := [N] \setminus \mathrm{TC}(t_k)$ remain marginally invariant:

$$\iota(\mathbf{z}_A) = p_{\mathbf{z}_A} = \tilde{p}_{\mathbf{z}_A}.$$

W.l.o.g, we assume $A = \{1, \ldots, t_k - 1\}$, define a function $h : \mathbb{R}^N \to \mathbb{R}^{|A|}$ with

$$h(\mathbf{z}) = [\mathbf{z}_1, \ldots, \mathbf{z}_{t_k-2}, \mathbf{z}_{t_k}]$$

that omits the $t_k-1$-th component of $\mathbf{z}$ but includes the variant component $t_k$. Note that the marginal of $\mathbf{z}_{t_k}$ after intervention remains invariant within $U$ because

$$
\begin{aligned}
p(\mathbf{z}_{t_k}) &= \int p_{t_k}(\mathbf{z}_{t_k} \mid \mathbf{z}_{\mathrm{pa}(t_k)}) p(\mathbf{z}_{\mathrm{pa}(t_k)}) d\mathbf{z}_{\mathrm{pa}(t_k)} && \mathrm{pa}(t_k) \in A \\
&= \int p_{t_k}(\mathbf{z}_{t_k} \mid \mathbf{z}_{\mathrm{pa}(t_k)}) \tilde{p}(\mathbf{z}_{\mathrm{pa}(t_k)}) d\mathbf{z}_{\mathrm{pa}(t_k)} && eq. \text{ (E.1) and both } p_k, \tilde{p}_k \text{ pdfs} \\
&= \int \tilde{p}_{t_k}(\mathbf{z}_{t_k} \mid \mathbf{z}_{\mathrm{pa}(t_k)}) \tilde{p}(\mathbf{z}_{\mathrm{pa}(t_k)}) d\mathbf{z}_{\mathrm{pa}(t_k)} \\
&= \tilde{p}(\mathbf{z}_{t_k}).
\end{aligned}
$$

Therefore, we have $\iota(h(\mathbf{z})) = \iota(h(\tilde{\mathbf{z}}))$ (with $\tilde{\mathbf{z}}$ noting the latent vectors under intervention) contradicting Defn. 2.1 (ii). The other direction (violating Defn. 2.1 (ii) implies violating Asm. E.1) can be proved using the same example.

$\square$

### E.2 PROOF FOR THM. 3.1

Our proof consists of the following steps:

1. We construct the optimal encoders $G^*$ (Defn. 3.2) and selectors $\Phi^*$ (Defn. 3.4) that solves the constrained optimization problem in Thm. 3.1.

2. We show that, for any invariance property $\iota_i \in \mathfrak{I}$ and any observation $\mathbf{x}^k$ in the corresponding $\iota_i$-equivalent subset $\mathbf{x}_{V_i}$, the selected representation $\phi^{(i,k)} \oslash g_k(\mathbf{x}^k)$ cannot contain any other information than the invariant partition $\mathbf{z}_{A_i}^k$.

3. Lastly, we prove that selected representation $\phi^{(i,k)} \oslash g_k(\mathbf{x}^k)$ relates to the ground truth invariant partition $\mathbf{z}_{A_i}^k$ through a diffeomorphism $h_k^i : \mathbb{R}^{|A_i|} \to \mathbb{R}^{|A_i|}$ for all invariance property $\iota_i \in \mathfrak{I}$ and for any observable $\mathbf{x}^k$ from the $\iota_i$-equivalent subset $\mathbf{x}_{V_i}$; in other words, $\phi^{(i,k)} \oslash g_k(\mathbf{x}^k)$ block-identifies $\mathbf{z}_{A_i}^k$ in the sense of Defn. 3.1.

**Lemma E.1** (Existence of optimal encoders and selectors). *Consider a set of observables $\mathcal{S}_{\mathbf{x}} = \{\mathbf{x}^1, \mathbf{x}^2, \ldots, \mathbf{x}^K\} \in \mathcal{X}$ generated from § 2 satisfying Asm. 2.1, then there exists optimal encoders $G^*$ (Defn. 3.2) and selectors $\Phi^*$ (Defn. 3.4) which satisfy both Constraints 3.1 and 3.2.*

*Proof.* The optimal encoders can be constructed as the set of the inverse of the ground truth mixing functions:

$$G^* = \{f_k^{-1}\}_{k \in [K]}, \tag{E.2}$$

$f_k^{-1}$ is smooth and invertible following Asm. B.1. By definition, for each $k \in [K]$, we have:

$$f_k^{-1}(\mathbf{x}^k) = \mathbf{z}^k \in \mathcal{Z}^k. \tag{E.3}$$

Next, we define the optimal selector $\Phi^* = \{\phi^{(i,k)}\}_{i \in [|\mathfrak{I}|], k \in [K]}$ such that for all $i \in |\mathfrak{I}|, k \in [K]$, it holds

$$\phi^{(i,k)} \oslash \mathbf{z}^k = \mathbf{z}_{A_i}^k. \tag{E.4}$$

Thus, the invariance constraint (Constraint 3.1) is trivially satisfied as given by § 2. The optimal encoder $f_k^{-1}$ is smooth and invertible following Asm. B.1 so the sufficiency constraint (Constraint 3.2) is also satisfied. Hence, we have shown the optimum of the constrained optimization problem in Thm. 3.1 exists. $\square$

**Lemma E.2** (Invariant component isolation). *Consider the same set of observables $\mathcal{S}_{\mathbf{x}}$ as introduced in Lemma E.1, then for any set of smooth encoders $G$ (Defn. 3.2), $\Phi$ (Defn. 3.4) that satisfy the invariance condition (Constraint 3.1), the learned representation $\phi^{(i,k)} \oslash g_k(\mathbf{x}^k)$ can only be dependent on the invariant latent variables $\mathbf{z}_{A_i}^k := \{\mathbf{z}_j^k : j \in A_i\}$, not any non-invariant variables $\mathbf{z}_q^k$ with $q \in A_i^c := [N] \setminus A_i$.*

*Proof.* This proof directly follows Defn. 2.1 (ii). Define

$$h_k^i := \phi^{(i,k)} \oslash g_k \circ f_k \quad k \in [K]. \tag{E.5}$$

By Constraint 3.1, for all $\iota_i \in \mathfrak{I}$, we have

$$\iota_i(h_k^i(\mathbf{z}^k)) = \iota_i(h_{k'}^i(\mathbf{z}^{k'})) \quad a.s. \qquad \forall k \neq k' \in [K]. \tag{E.6}$$

According to Defn. 2.1 (ii), for all $i \in [|\mathfrak{I}|], k \in V_i, h_k^i$ cannot depend on any other latent component $\mathbf{z}_q$ with $q \notin A_i$. Therefore, we have shown that $h_k^i$ is a function of $\mathbf{z}_{A_i}^k$, for all $i \in [|\mathfrak{I}|], k \in V_i$. $\square$

**Theorem 3.1** (Identifiability of multiple invariant blocks). *Consider a set of observables $\mathcal{S}_{\mathbf{x}} = \{\mathbf{x}^1, \mathbf{x}^2, \ldots, \mathbf{x}^K\} \in \mathcal{X}$ generated from § 2 satisfying Asm. 2.1. Let $G, \Phi$ be the set of smooth encoders (Defn. 3.2) and selectors (Defn. 3.4) that satisfy Constraints 3.1 and 3.2, then the invariant component $\mathbf{z}_{A_i}^k$ is block-identified (Defn. 3.1) by $\phi^{(i,k)} \oslash g_k$ for all $\iota_i \in \mathfrak{I}, k \in [K]$.*

*Proof.* Lem. E.1 verifies that there exists such optimum which satisfies both invariance and sufficiency conditions (Constraints 3.1 and 3.2). Following Lem. E.2, the composition $\phi^{(i,k)} \oslash g_k$ can only encode information related to the invariant latent subset $A_i$ specified by the invariance property $\iota_i \in \mathfrak{I}$ for all $k \in V_i$. As given by Constraint 3.2, $\phi^{(i,k)} \oslash g_k$ contain all information the ground truth invariant latents $\mathbf{z}_{A_i}$ for $i$ with $k \in V_i$. Therefore, the selected representation $\phi^{(i,k)} \oslash g_k(\mathbf{x}^k)$ relates to the ground truth invariant partition $\mathbf{z}_{A_i}$ through some diffeomorphism, i.e., $\mathbf{z}_{A_i}$ is blocked-identified by $\phi^{(i,k)} \oslash g_k(\mathbf{x}^k)$ for all invariance property $\iota_i \in \mathfrak{I}$ and observable $k \in V_i$, . $\qquad\square$

### E.3  PROOFS FOR GENERALIZATION OF VARIANT LATENTS

**Proposition 3.2** (General non-identifiability of variant latent variables). *Consider the setup in Thm. 3.1, let $A := \bigcup_{i \in [|\mathfrak{I}|]} A_i$ denote the union of block-identified latent indices and $A^c := [N] \backslash A$ the complementary set where no $\iota$-invariance $\iota \in \mathfrak{I}$ applies, then the variant latents $\mathbf{z}_{A^c}$ cannot be identified.*

*Proof.* We provide a simple counter example with two latent variables $\mathbf{z} = [\mathbf{z}_1, \mathbf{z}_2]$, with the mixing function $f$ being the identity map id. W.l.o.g. we assume the invariant partition to be $A = \{1\}$. According to Thm. 3.1, the invariant latent variable can be identified up to a certain bijection $h : \mathbb{R} \rightarrow \mathbb{R}$. Let $\hat{\mathbf{z}}$ be the estimated representation:

$$\hat{\mathbf{z}} = [h(\mathbf{z}_1), \mathbf{z}_2 - \mathbf{z}_1] \tag{E.7}$$

with the estimated mixing function $\hat{f} : \mathbb{R}^2 \rightarrow \mathbb{R}^2$:

$$\hat{f}(\hat{\mathbf{z}}) = [h^{-1}(\hat{\mathbf{z}}_1), \hat{\mathbf{z}}_2 + h^{-1}(\hat{\mathbf{z}}_1)], \tag{E.8}$$

then we obtain the same observations $\hat{f}(\hat{\mathbf{z}}) = f(\mathbf{z})$ whereas $\hat{\mathbf{z}}_2$ consists of a mixing of $\mathbf{z}_1$ and $\mathbf{z}_2$, showing the variant latent variable $\mathbf{z}_2$ can not be identified. $\qquad\square$

**Proposition 3.3** (Identifiability of variant latent under independence). *Consider an optimal encoder $g \in G^*$ and optimal selector $\phi \in \Phi^*$ from Thm. 3.1 that jointly identify an invariant block $\mathbf{z}_A$ (we omit subscriptions $k, i$ for simplicity), then $\mathbf{z}_{A^c}(A^c := [N] \setminus A)$ can be identified by the complementary encoding partition $(1 - \phi) \oslash g$ only if*

*(i)  $g$ is invertible in the sense that $I(\mathbf{x}, g(\mathbf{x})) = H(\mathbf{x})$;*

*(ii)  $\mathbf{z}_{A^c}$ is independent on $\mathbf{z}_A$.*

*Proof.* ($\Leftarrow$): We start by showing the sufficiency of conditions (i) and (ii). The mutual information between the observation $\mathbf{x} \in \mathcal{S}_\mathbf{x}$ and the optimal encoder $g \in G^*$ from Thm. 3.1 writes:

$$I(\mathbf{x}, g(\mathbf{x})) = H(\mathbf{x}) - H(\mathbf{x} \mid g(\mathbf{x})),$$

following condition (i) in Proposition 3.3, the second term (conditional entropy) must equal zero: $H(\mathbf{x} \mid g(\mathbf{x})) = 0$.

Writing the $\mathbf{x} = f(\mathbf{z}_A, \mathbf{z}_{A^c})$, we have

$$H(\mathbf{x} \mid g(\mathbf{x})) = H(f(\mathbf{z}_A, \mathbf{z}_{A^c}) \mid g(\mathbf{x})) = H(\mathbf{z}_A, \mathbf{z}_{A^c} \mid g(\mathbf{x})),$$

because the mixing function $f$ is deterministic as given by Defn. B.2.

Note that $g(\mathbf{x})$ can be decomposed into two separate partitions: $\phi \oslash g(\mathbf{x}), (1 - \phi) \oslash g(\mathbf{x})$; thus we can write the conditional entropy as

$$\begin{aligned} H(\mathbf{x} \mid g(\mathbf{x})) &= H(\mathbf{z}_A, \mathbf{z}_{A^c} \mid \phi \oslash g(\mathbf{x}), (1 - \phi) \oslash g(\mathbf{x})) \\ &= H(\mathbf{z}_{A^c} \mid \mathbf{z}_A, \phi \oslash g(\mathbf{x}), (1 - \phi) \oslash g(\mathbf{x})) + H(\mathbf{z}_A \mid \phi \oslash g(\mathbf{x}), (1 - \phi) \oslash g(\mathbf{x})) \end{aligned}$$

Given that $\phi \oslash g(\mathbf{x})$ block identifies $\mathbf{z}_A$, $(1 - \phi) \oslash g(\mathbf{x}))$ cannot contain any information about $\mathbf{z}_A$, hence we can simplify the second term as

$$H(\mathbf{z}_A \mid \phi \oslash g(\mathbf{x}))$$

Using the additional mutual independence assumption between $\mathbf{z}_A$ and $\mathbf{z}_{A^c}$ (Proposition 3.3 (ii)), we can rewrite the first term as

$$H(\mathbf{z}_{A^c} \mid (1 - \phi) \oslash g(\mathbf{x})).$$

As a result, the condition entropy $H(\mathbf{x} \mid g(\mathbf{x}))$ can be decomposed as

$$H(\mathbf{x} \mid g(\mathbf{x})) = H(\mathbf{z}_A \mid \phi \oslash g(\mathbf{x})) + H(\mathbf{z}_{A^c} \mid (1 - \phi) \oslash g(\mathbf{x})) = 0.$$

Since $H(\mathbf{z}_A \mid \phi \oslash g(\mathbf{x})) = 0$ following Constraint 3.2, the second term also must be zero, i.e., $H(\mathbf{z}_{A^c} \mid (1 - \phi) \oslash g(\mathbf{x})) = 0$, which is satisfied only if $(1 - \phi) \oslash g(\mathbf{x})$ is a invertible function of $\mathbf{z}_{A^c}$. That is, $(1 - \phi) \oslash g(\mathbf{x})$ block-identifies $\mathbf{z}_{A^c}$.

($\Rightarrow$): We show that block-identifiability of $\mathbf{z}_{A^c}$ by $(1 - \phi) \oslash g$ implies both conditions (i) and (ii).

**For condition (i)** To show $g$ is invertible in the sense that $I(\mathbf{x}, g(\mathbf{x})) = H(\mathbf{x})$, it is equivalent to show that $H(\mathbf{x} \mid g(\mathbf{x})) = 0$ because

$$I(\mathbf{x}, g(\mathbf{x})) = H(\mathbf{x}) - H(\mathbf{x} \mid g(\mathbf{x}))$$

Writing $\mathbf{x} = f(\mathbf{z}_A, \mathbf{z}_{A^c})$, we have

$$H(\mathbf{x} \mid g(\mathbf{x})) = H(f(\mathbf{z}_A, \mathbf{z}_{A^c}) \mid g(\mathbf{x})) = H(\mathbf{z}_A, \mathbf{z}_{A^c} \mid g(\mathbf{x})),$$

because the mixing function $f$ is a diffeomorphism as given by Defn. B.2.

Given that $\phi \oslash g(\mathbf{x}) = h(\mathbf{z}_A)$ (indicated by Thm. 3.1) and $(1 - \phi) \oslash g(\mathbf{x}) = h^c(\mathbf{z}_{A^c})$ for some diffeomorphisms $h : \mathcal{Z}_A \to \mathcal{Z}_A$ and $h^c : \mathcal{Z}_{A^c} \to \mathcal{Z}_{A^c}$, we can further decompose the conditional entropy $H(\mathbf{x} \mid g(\mathbf{x}))$ into two separate terms

$$\begin{aligned} H(\mathbf{x} \mid g(\mathbf{x})) &= H(\mathbf{z}_A, \mathbf{z}_{A^c} \mid \phi \oslash g(\mathbf{x}), (1 - \phi) \oslash g(\mathbf{x})) \\ &= H(\mathbf{z}_{A^c} \mid \mathbf{z}_A, \phi \oslash g(\mathbf{x}), (1 - \phi) \oslash g(\mathbf{x})) + H(\mathbf{z}_A \mid \phi \oslash g(\mathbf{x}), (1 - \phi) \oslash g(\mathbf{x})) \\ &= H(\mathbf{z}_{A^c} \mid \mathbf{z}_A, h(\mathbf{z}_A), h^c(\mathbf{z}_{A^c})) + H(\mathbf{z}_A \mid h(\mathbf{z}_A), h^c(\mathbf{z}_{A^c}))) \end{aligned}$$

Since both $h, h^c$ are diffeomorphisms (meaning $h(\mathbf{z}_A)$, $h^c(\mathbf{z}_{A^c})$ fully determines $\mathbf{z}_A, \mathbf{z}_{A^c}$, respectively), both conditional entropy terms are zero; thus $H(\mathbf{x} \mid g(\mathbf{x})) = 0$. Hence, we have shown that $g$ is invertible in the sense that $I(\mathbf{x}, g(\mathbf{x})) = H(\mathbf{x})$.

**For condition (ii)** Given that $\phi \oslash g(\mathbf{x})$ block identifies $\mathbf{z}_A$, $(1 - \phi) \oslash g(\mathbf{x})$ cannot contain any information about $\mathbf{z}_A$ (Defn. 3.1), i.e.,

$$I(\mathbf{z}_A \mid (1 - \phi) \oslash g(\mathbf{x})) = 0$$

Writing $(1 - \phi) \oslash g(\mathbf{x}) = h^c(\mathbf{z}_{A^c})$, we have

$$I(\mathbf{z}_A \mid h^c(\mathbf{z}_{A^c})) = I(\mathbf{z}_A \mid \mathbf{z}_{A^c}) = 0,$$

as mutual information is invariant under invertible transformations and $h^c$ is invertible. Therefore, we have shown that $\mathbf{z}_{A^c}$ is independent on $\mathbf{z}_A$.

To this end, we have shown both condition (i) and (ii) and necessary and sufficient conditions for the block-identifiability of $\mathbf{z}_{A^c}$ which completes the proof.

$\square$

### E.4 PROOFS FOR GRANULARITY OF LATENT VARIABLE IDENTIFICATION

**Proposition C.1** (Granularity of identification). *Affine-identifiability (Defn. C.3) implies element-identifiability (Defn. C.2) and block affine-identifiability (Defn. C.1) while element-identifiability and block affine-identifiability implies block-identifiability (Defn. 3.1).*

*Proof.* The diagonal matrix $\Lambda$ in eq. (C.3) is invertible and thus also a diffeomorphism $h$ (eq. (C.2)). Hence, affine-identifiability implies element-identifiability. Affine-identifiability provides identification results with block-size one thus implies block affine-identifiability. On the other hand, block affine-identifiability is block-identifiability with affine bijection $h$ and element-identifiability defines a special case of block-identifiability where each latent component $\mathbf{z}_i$ is an individual block. $\square$

**Proposition C.2** (Transition between identification levels)**.** *The transition between different levels of latent variable identification (Fig. 2) can be summarized as follows:*

    *(i) Element- identifiability (Defns. C.2 and C.3) can be obtained from block-wise identifiability (Defns. 3.1 and C.1) when each individual latent constitutes an invariant block;*

    *(ii) Identifiability up to an affine transformation (Defns. C.1 and C.3) can be obtained from general identifiability on arbitrary diffeomorphism (Defns. 3.1 and C.2) by additionally assuming that both the ground truth mixing function and decoder are finite degree polynomials of the same degree.*

*Proof.* The proof for (i) is trivial in the sense that identification of block with size one boils down to the identification on the element level. (ii) directly follows Ahuja et al. (2023, Thm. 4.4) and Zhang et al. (2024a, Lem. 1), stating that when both ground truth mixing function and decoder are finite degree polynomials of the same degree, the *invertible* encoder learns a representation that is affine linear to the ground truth latents, i.e., $\hat{\mathbf{z}} = \mathbf{L} \cdot \mathbf{z} + \mathbf{b}$ with $\mathbf{L} \in \mathbb{R}^{N \times N}$.

$\square$

### E.5   Proof for Cor. D.1

**Corollary D.1** (Identifiability from single node imperfect intervention per node)**.** *Given $N$ environments $\{k_1, \ldots, k_N\} \subseteq [K]$ satisfying Asm. D.1, the ground truth latent variables $\mathbf{z}$ can be identified up to element-wise diffeomorphism (Defn. C.2) by combining both marginal and score invariances (eqs. (D.3) and (D.4)) under our framework (Thm. 3.1).*

*Proof.* We consider a coarse-grained version of the underlying causal graph consisting of a block-node $\mathbf{z}_{[N-1]} := \{\mathbf{z}_1, \ldots, \mathbf{z}_{N-1}\}$ and the leaf node $\mathbf{z}_N$ with $\mathbf{z}_{[N-1]}$ causing $\mathbf{z}_N$ (i.e., $\mathbf{z}_{[N-1]} \to \mathbf{z}_N$). We first select a pair of environments $V = \{0, k_N\}$ consisting of the observational environment and the environment where the leaf node $\mathbf{z}_N$ is intervened upon. According to eq. (D.3), the *marginal invariance* holds for the partition $A = [N-1]$, implying identification on $\mathbf{z}_{[N-1]}$ from Thm. 3.1. At the same time, when considering the set of environments $V' = \{0, k_1, \ldots, k_{N-1}\}$, the leaf node $N$ is the only component that satisfy *score* invariance across all environments $V'$, because $N$ is not the parent of any intervened node (also see (Varici et al., 2023, Lemma 4)). So here we have another invariant partition $A' = \{N\}$, implying identification on $\mathbf{z}_N$ (Thm. 3.1). By jointly enforcing the marginal and score invariance on $A$ and $A'$ under a sufficient encoder (Constraint 3.2), we identify both $\mathbf{z}_{[N-1]}$ as a block and $\mathbf{z}_N$ as a single element. Formally, for the parental block $\mathbf{z}_{[N-1]}$, we have:

$$\hat{\mathbf{z}}_{[N-1]}^k = g_{:N-1}(\mathbf{x}^k) \qquad \forall k \in \{0, k_1, \ldots, k_N\} \tag{E.9}$$

where $g_{:N-1}(\mathbf{x}^k) := [g(\mathbf{x}^k)]_{:N-1}$ relates to the ground truth $\mathbf{z}_{[N-1]}$ through some diffeomorphism $h_{[N-1]} : \mathbb{R}^{N-1} \to \mathbb{R}^{N-1}$ (Defn. 3.1). Now, we can remove the leaf node $N$ as follows: For each environment $k \in \{0, k_1, \ldots, k_{N-1}\}$, we compute the pushforward of $P_{\mathbf{x}^k}$ using the learned encoder $g_{:N-1} : \mathcal{X}^k \to \mathbb{R}^{N-1}$:

$$P_{\hat{\mathbf{z}}_{[N-1]}^k} = g_{:N-1}\#(P_{\mathbf{x}^k})$$

Note that the estimated representations $P_{\hat{\mathbf{z}}_{[N-1]}^k}$ can be seen as a new observed data distribution for each environment $k$ that is generated from the subgraph $\mathcal{G}_{-N}$ without the leaf node $N$. Using an iterative argument, we can identify all latent variables element-wise (Defn. C.2). $\square$

## F   Implementation Details

This section provides further details about the experiment settings of § 5, including a formal introduction to the ISTAnt dataset, highlighted open challenges (App. F.1), and additional training settings for reproducibility (App. F.2).

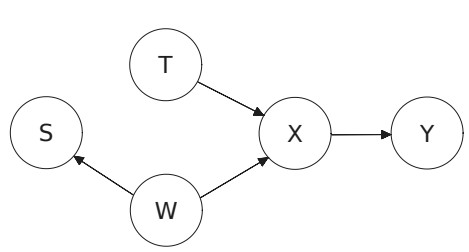

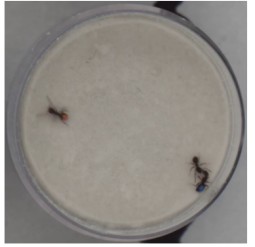

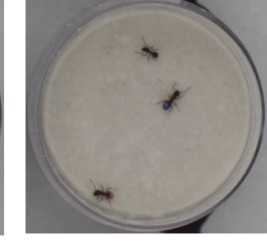

(a) Grooming (blue to focal)    (b) No Action

Figure 3: Causal Model for generic partially annotated scientific experiment: $T$ treatment, $W$ experimental settings, $X$ high-dimensional observation, $Y$ outcome, $S$ annotation flag. Figure and caption adapted from (Cadei et al., 2024, Fig. 1)

Figure 4: Examples of high-dimensional observations $X$ with corresponding annotated social behaviour $Y$ (grooming). Figure and caption adapted from (Cadei et al., 2024, Fig. 2)

### F.1 CASE STUDY: ISTANT

**Problem.** Despite the majority of CRL algorithms being designed to enforce the identifiability of some latent factors and tested on controlled synthetic benchmarks, there are a plethora of real-world applications across scientific disciplines requiring representation learning to answer causal questions (Robins et al., 2000; Samet et al., 2000; Van Nes et al., 2015; Runge, 2023). Recently, Cadei et al. (2024) introduced ISTAnt, the first real-world representation learning benchmark with a real causal downstream task (treatment effect estimation). This benchmark highlights different challenges (sources of biases) that could arise from machine learning pipelines even in the simplest possible setting of a randomized controlled trial. Videos of ants triplets are recorded, and a per-frame representation has to be extracted for supervised behavior classification to estimate the Average Treatment Effect of an intervention (exposure to a chemical substance). Beyond desirable identification result on the latent factors (implying that the causal variables are recovered without bias), no clear algorithm has been proposed yet on minimizing the Treatment Effect Bias (TEB) (Cadei et al., 2024). One of the challenges highlighted by Cadei et al. (2024) is that in practice, there is both covariate and concept shifts due to the effect modification from training on a non-random subset of the RCT because, for example, ecologists do not label individual frames but whole video recordings. Figs. 3 and 4 shows the underlying causal graph and example input.

**Solution.** Relying on our framework, we can explicitly aim for low TEB by leveraging *known data symmetries* from the experimental protocol. In fact, the causal mechanism ($P(Y^e|do(X^e = x))$) stays invariant among the different experiment settings (i.e., individual videos or position of the petri dish). This condition can be easily enforced by existing domain generalization algorithms. For exemplary purposes, we choose Variance Risk Extrapolation (Krueger et al., 2021, V-REx), which directly enforces both the invariance sufficiency constraints (Constraints 3.1 and 3.2) by minimizing the Empirical Risk together with the risk variance inter-environments.

**Implementation details** All training settings follow the best-performing settings from (Cadei et al., 2024), which we restate in Tab. 2 for reference.

**Discussion.** Interestingly, Gulrajani & Lopez-Paz (2020) empirically demonstrated that no domain generalization algorithm consistently outperforms Empirical Risk Minimization in *out-of-distribution* prediction. However, in this application, our goal is not to achieve high out-of-distribution accuracy but rather to identify a representation that is invariant to the effect modifiers introduced by the data labeling process. This experiment serves as a clear example of the paradigm shift of CRL via the invariance principle. While existing CRL approaches design algorithms based on specific assumptions that are often challenging to align with real-world applications, our approach begins from the application perspective. It allows for the specification of known data symmetries and desired properties of the learned representation, followed by selecting an appropriate implementation for the distance function (potentially from existing methods). Ultimately, identifiability hinges on the guarantee of asymptotic consistency in the estimates.

| Model/Hyper-parameters | Value(s) |
|---|---|
| Encoder | DINOv2 (Oquab et al., 2023) |
| Encoder (token) | class |
| MLP (head): hidden layers | 1 |
| MLP (head): hidden nodes | 256 |
| MLP (head): activation function | ReLU + Sigmoid output |
| Tass | or |
| Dropout | No |
| Regularization | No |
| Loss | BCELoss (with positive weighting) |
| Loss: Positive Weight | $\frac{\sum_{i=1}^{n_s} 1 - Y_i}{\sum_{i=1}^{n_s} Y_i}$ |
| Learning Rate | 0.0005 |
| Optimizer | Adam ($\beta_1 = 0.9, \beta_2 = 0.9, \epsilon = 10^{-8}$) |
| Batch Size | 128 |
| Epochs | 15 |
| Seeds | `range(20)` |

Table 2: Model and training details for the case study on ISTAnt (§ 5.1). Table adapted from (Cadei et al., 2024, Tab. 4)

### F.2 SYNTHETIC ABLATION WITH "NINTERVENTIONS"

The numerical data is generated using a linear Gaussian additive noise model as follows:

$$
\begin{aligned}
p(\mathbf{z}_1) &= \mathcal{N}(\mu_1, \sigma_1^2) \\
p(\mathbf{z}_2 \mid \mathbf{z}_1) &= \mathcal{N}(\alpha_1 \cdot \mathbf{z}_1 + \beta_1, \sigma_2^2) \\
p(\mathbf{z}_3 \mid \mathbf{z}_2) &= \mathcal{N}(\alpha_2 \cdot \mathbf{z}_2 + \beta_2, \sigma_3^2) \\
\tilde{p}(\mathbf{z}_2) &= \mathcal{N}(\tilde{\mu}_2, \tilde{\sigma}_2^2)
\end{aligned}
\tag{F.1}
$$

We choose $\mu_1 = 10.5, \sigma_1 = 0.8, \alpha_1 = 0.02, \beta_1 = 0, \sigma_2 = 0.5, \alpha_2 = 1, \beta_2 = 3, \sigma_3 = 1, \tilde{\sigma}_2 = 0.02$. We sample three independent $\tilde{\mu}_2$ according to a uniform distribution $\text{Unif}[2, 5]$ to validate the consistency of the identification results.

For the training, we employ a simple auto-encoder architecture implementing both encoder and decoder as 3-Layer MLP. We enforce the marginal invariance using the Max Mean Discrepancy loss (MMD) on the first and last component $\hat{\mathbf{z}}_1, \hat{\mathbf{z}}_3$. Formally, the objective function writes

$$
\mathcal{L}(g, \hat{f}) = \mathbb{E}_{\mathbf{x}, \tilde{\mathbf{x}}} \left[ \left\| \hat{f}(g(\mathbf{x})) - \mathbf{x} \right\|_2^2 + \left\| \hat{f}(g(\tilde{\mathbf{x}})) - \mathbf{x} \right\|_2^2 \right] + \text{MMD}(g(\mathbf{x})_{[1,3]}, g(\tilde{\mathbf{x}})_{[1,3]}),
$$

where $\mathbf{x}, \tilde{\mathbf{x}}$ denote the observational and ninterventional data, respectively.

Further training details are summarized in Tab. 3

## G FURTHER DISCUSSIONS AND CONNECTIONS TO OTHER FIELDS

In this paper, we take a closer look at the wide range of CRL methods. Interestingly, we find that the differences between them may often be more related to "semantics" than to fundamental method-ological distinctions. We identified two components involved in identifiability results: preserving information of the data and a set of known invariances. Our results have two immediate implications. First, they provide new insights into the "CRL problem," particularly clarifying the role of causal assumptions. We have shown that while learning the graph requires traditional causal assumptions such as additive noise models or access to interventions, identifying the causal variables may not. This is an important result, as access to causal variables is standalone useful for downstream tasks, e.g., for training robust downstream predictors or even extracting pre-treatment covariates for treat-ment effect estimation (Yao et al., 2024), even without knowledge of the full causal graph. Second, we have exemplified how causal representation can lead to successful applications in practice. We

Table 3: Training setup for synthetic ablations in § 5.2.

| Parameter | Value |
|---|---|
| Mixing function | 3-layer MLP |
| Encoder | 3-layer MLP |
| Decoder | 3-layer MLP |
| Hidden dim | 128 |
| Activation | Leaky-ReLU |
| Optimizer | Adam |
| Adam: learning rate | 1e-4 |
| Adam: beta1 | 0.9 |
| Adam: beta2 | 0.999 |
| Adam: epsilon | 1e-8 |
| Batch size | 4000 |
| Sample size | 200,000 |
| # Epochs | 500 |

moved the goal post from a characterization of specific assumptions that lead to identifiability, which often do not align with real-world data, to a general recipe that allow practitioners to specify known invariances in their problem and learn representations that align with them. In the domain generalization literature, it has been widely observed that invariant training methods often do not consistently outperform empirical risk minimization (ERM). In our experiments, instead, we have demonstrated that the specific invariance enforced by V-REx (Krueger et al., 2021) entails good performance in our causal downstream task (§ 5.1). Our paper leaves out certain settings concerning identifiability that may be interesting for future work, such as discrete variables and finite samples guarantees.

One question the reader may ask, then, is "*so what is exactly _causal_ in CRL?*". We have shown that the identifiability results in typical CRL are primarily based on invariance assumptions, which do not necessarily pertain to causality. We hope this insight will broaden the applicability of these methods. At the same time, we used causality as a language describing the "parameterization" of the system in terms of latent causal variables with associated known symmetries. Defining the symmetries at the level of these causal variables gives the identified representation a causal meaning, important when incorporating a graph discovery step or some other causal downstream task like treatment effect estimation. Ultimately, our representations and latent causal models can be "true" in the sense of (Peters et al., 2014) when they allow us to predict "causal effects that one observes in practice". Overall, our view also aligns with "phenomenological" accounts of causality (Janzing & Mejia, 2024), that define causal variables from a set of elementary interventions. In our setting too, the identified latent variables or blocks thereof are directly defined by the invariances at hand. From the methodological perspective, all is needed to learn causal variables is for the symmetries defined over the causal latent variables to entail some statistical footprint across pockets of data. If variables are available, learning the graph has a rich literature (Peters et al., 2017), with assumptions that are often compatible with learning the variables themselves. Our general characterization of the variable learning problem opens new frontiers for research in representation learning:

### G.1 REPRESENTATIONAL ALIGNMENT

Several works (Li et al. (2015); Moschella et al. (2022); Kornblith et al. (2019); Huh et al. (2024)) have highlighted the emergence of similar representations in neural models trained independently. In Huh et al. (2024) is hypothesized that neural networks, trained with different objectives on various data and modalities, are converging toward a *shared* statistical model of reality within their representation spaces. To support this hypothesis, they measure the alignment of representations proposing to use a mutual nearest-neighbor metric, which measures the mean intersection of the k-nearest neighbor sets induced by two kernels defined on the two spaces, normalized by k. This metric can be an instance to the distance function in our formulation in Thm. 3.1. Despite not being optimized directly, several models in multiple settings (different objectives, data and modalities) seem to be aligned, hinting at the fact that their individual training objectives may be respecting some unknwon symmetries. A precise formalization of the latent causal model and identifiability in the context of foundational models remains open and will be objective for future research.

### G.2 ENVIRONMENT DISCOVERY

Domain generalization methods generalize to distributions potentially far away from the training, distribution, via learning representations invariant across distinct environments. However this can be costly as it requires to have label information informing on the partition of the data into environments. Automatic environment discovery (Creager et al. (2021); Arefin et al. (2024); Pezeshki et al. (2024)) attempts to solve this problem by learning to recover the environment partition. This is an interesting new frontier for CRL, discovering data symmetries as opposed to only enforcing them. For example, this would correspond to having access to multiple interventional distributions but without knowing which samples belong to the same interventional or observational distribution. Discovering that a data set is a mixture of distributions, each being a different intervention on the same causal model, could help increase applicability of causal representations to large obeservational data sets. We expect this to be particularly relevant to downstream tasks were biases to certain experimental settings are undesirable, as in our case study on treatment effect estimation from high-dimensional recordings of a randomized controlled trial.

### G.3 GEOMETRIC DEEP LEARNING

Geometric deep learning (GDL) (Bronstein et al. (2017; 2021)) is a well estabilished learning paradigm which involves encoding a geometric understanding of data as an inductive bias in deep learning models, in order to obtain more robust models and improve performance. One fundamental direction for these priors is to encode symmetries and invariances to different types of transformations of the input data, e.g. rotations or group actions (Cohen & Welling (2016); Cohen et al. (2018)), in representational space. Our work can be fundamentally related with this direction, with the difference that we don't aim to model *explicitly* the transformations of the input space, but the invariances defined at the latent level. While an initial connection has been developed for disentanglement Fumero et al. (2021); Higgins et al. (2018), a precise connection between GDL and CRL remains a open direction. We expect this to benefit the two communities in both directions: (i) by injecting geometric priors in order to craft better CRL algorithms and (ii) by incorporating causality into successful GDL frameworks, which have been fundamentally advancing challenging real-world problems, such as protein folding (Jumper et al. (2021)).

Table 4: **A non-exhaustive summary of existing identifiability results for Causal Representation Learning.** All of the listed works assume injectivity of the mixing function and causal sufficiency (Markovianity) for the causal latent variables. Many listed papers depend on further technical assumptions and could yield additional results which we omitted for clarity; see references for details. In the table, "not assigned" means that the practical method did not directly enforce the invariance principle but considered other algorithmic designs that still implicitly preserve the data symmetries.

| Work | Causal Model | Mixing Function | Invariance | Source of invariance, Inv. subset $A$ | Invariance reg. | Sufficiency reg. | Identifiability | Expl. |
|---|---|---|---|---|---|---|---|---|
| | | | | | INTERVENTIONAL/MULTI-ENVIRONMENT CRL | | | | |
| Squires et al. (2023, Thms. 1 & 2) | linear | linear | distributional | perfect intervention per node | $\mathrm{rank}(H^\top \Delta_k H)\overset{!}{=}1$ for source nodes; linear encoder $g(\mathbf{x})=H\mathbf{x}$, where $\Delta_k := B_k^\top B_k - B_0^\top B_0,\ \mathbf{z}=B_k^{-1}\epsilon$ | $g$ invertible by assumption | affine-id. and partial order preserving graph-id. | (a) |
| Ahuja et al. (2024, Thm. 2) | nonparam. | finite-deg. poly. | marginal | single-node imperfect interventions on variant latents | $\sum_{k,k'}\sum_{j\in A}$ MMD$(p^k_{[g(\mathbf{x})]_j}, p^{k'}_{[g(\mathbf{x})]_j})$ | $\sum_k \mathbb{E}_{\mathbf{x}^k}\|\hat{f}(g(\mathbf{x}^k))-\mathbf{x}^k\|_2^2$ | block affine-id. | - |
| Ahuja et al. (2024, Thm. 3) | nonparam. | finite-deg. poly. | marginal | multi-node imperfect interventions on variant latents | $\sum_{k,k'}\sum_{j\in A}$ MMD$(p^k_{[g(\mathbf{x})]_j}, p^{k'}_{[g(\mathbf{x})]_j})$ | $\sum_k \mathbb{E}_{\mathbf{x}^k}\|\hat{f}(g(\mathbf{x}^k))-\mathbf{x}^k\|_2^2$ | block affine-id. | - |
| Ahuja et al. (2024, Thm. 4) | nonparam. | finite-deg. poly. | marginal support | imperfect interventions on variant latents | $\sum_{k,k'}\sum_{j\in A}\left\|\mathrm{bnd}(\hat{z}^k_j)-\mathrm{bnd}(\hat{z}^{k'}_j)\right\|_2^2$ | $\sum_k \mathbb{E}_{\mathbf{x}^k}\|\hat{f}(g(\mathbf{x}^k))-\mathbf{x}^k\|_2^2$ | block affine-id. | - |
| Buchholz et al. (2024) | linear Gaussian | nonparam. | marginal | perfect intervention per node | $-\mathbb{E}_{l\sim\mathcal{U}(\{0,k\})}\mathbb{E}_{\mathbf{x}^l}\ln\left(e^{\mathbf{1}_{l=k}g_k(\mathbf{x}^l)}\right)$ | $\mathbb{E}_{l\sim\mathcal{U}(\{0,k\})}\mathbb{E}_{\mathbf{x}^l}\ln\left(e^{g_k(\mathbf{x}^l)}+1\right)$ | affine id. + graph id. | (a) |

| Work | Causal Model | Mixing Function | Invariance | Source of invariance, Inv. subset $A$ | Invariance reg. | Sufficiency reg. | Identifiability | Expl. |
|---|---|---|---|---|---|---|---|---|
| Varici et al. (2023, Thm. 16) | nonparam. | linear | distributional | perfect intervention per node | $\|\Delta_{\mathbf{x}}^s(U^\top)\|_0$. For all $j,k\in[N]$, its element $[\Delta_{\mathbf{x}}^s(U^\top)]_{j,k}=$ $\mathbf{1}([U^\top S(\mathbf{x}^0)]_j \overset{P_{\mathbf{x}}^{0,k}}{\neq} [U^\top S(\mathbf{x}^k)]_j)$, $g(\mathbf{x}):=U^\top\mathbf{x}$ | $g$ invertible by assumption | affine-id. + graph-id. | (a) |
| Varici et al. (2023, Thm. 13) | nonparam. | linear | distributional | imperfect intervention per node | $\|\Delta_{\mathbf{x}}^s(U^\top)\|_0$. For all $j,k\in[N]$, its element $[\Delta_{\mathbf{x}}^s(U^\top)]_{j,k}=$ $\mathbf{1}([U^\top S(\mathbf{x}^0)]_j \overset{P_{\mathbf{x}}^{0,k}}{\neq} [U^\top S(\mathbf{x}^k)]_j)$, $g(\mathbf{x}):=U^\top\mathbf{x}$ | $g$ invertible by assumption | block affine-id. + graph-id. | (a) |
| Varici et al. (2024a, Thm. 3) | nonparam. | nonparam. | interventional target | paired perfect intervention per node | $\min\|\Delta^s(g)\|_0$ s.t. it is diagonal. $\Delta^s(g)_{j,k}=$ $\mathbb{E}\left[\left|[S(g(\mathbf{x}^k))-S(g(\mathbf{x}^{k'}))]_j\right|\right]$ | $g$ invertible by assumption | element-id. + graph-id. | (c) |
| Varici et al. (2024b, Thm. 1) | nonparam. | linear | distributional | linearly independent multi-node perfect intervention | Linear encoder $g(\mathbf{x})=H\mathbf{x}$, $H_i^*\in\mathrm{im}(\Delta s_{\mathbf{x}}\mathbf{w}_i)\backslash\mathrm{span}(H_{[i-1]}^*)$ such that the $\dim$ of $\mathrm{proj}_{\mathrm{null}\left(H_{[i-1]}^*\right)}\mathrm{im}(\Delta S_{\mathbf{x}}\mathbf{w}_i)$ equals one. | $g$ invertible by assumption | affine id. + graph id. | (b) |

| Work | Causal Model | Mixing Function | Invariance | Source of invariance, Inv. subset $A$ | Invariance reg. | Sufficiency reg. | Identifiability | Expl. |
|---|---|---|---|---|---|---|---|---|
| Varici et al. (2024b, Thm. 2) | nonparam. | linear | distributional | linearly independent multinode imperfect intervention | Linear encoder $g(\mathbf{x})=H\mathbf{x}$, $H_i^* \in \mathrm{im}(\Delta s_{\mathbf{x}}\mathbf{w}_i)\setminus\mathrm{span}(H_{[i-1]}^*)$ such that the $\dim$ of $\mathrm{proj}_{\mathrm{null}\left(H_{[i-1]}^*\right)}\mathrm{im}(\Delta S_{\mathbf{x}}\mathbf{w}_i)$ equals one. | $g$ invertible by assumption | block affine-id. + graph id. | (b) |
| Zhang et al. (2024a) | nonparam. | finite-deg. poly. | distributional | imperfect intervention per node | $-\sum_k \mathrm{MMD}(q_{\tilde{\mathbf{x}}^k}, p_{\mathbf{x}^k})$ where $\tilde{\mathbf{x}}^k$ the generated "counterfactual" pair through VAE | $-\sum_k \mathbb{E}_{\mathbf{x}^k}\log p(\mathbf{x}^k|g(\mathbf{x}^k))$ | affine-id. + graph id. | (a) |
| Wendong et al. (2024, Thm. 4.5) | nonparam. | nonparam. | marginal | marginal invariance from multiple fat-hand interventions on the same set of interventional targets $I$, invariant partition $A:=[N]\setminus I$ | model selection | $-\sum_k \log p_{\boldsymbol{\theta}}^k(\mathbf{x}^k)$ | block-id. (known graph) | (d) |
| von Kügelgen et al. (2024, Thm. 4.1) | nonparam. | nonparam. | interventional target | paired perfect intervention per node | model selection | $-\sum_k \log p_{\boldsymbol{\theta}}^k(\mathbf{x}^k)$ | element-id. + graph-id | (c) |
| | | | | MULTIVIEW CRL | | | | |
| von Kügelgen et al. (2021) | nonparam. | nonparam. | sample level on all realizations of $z_A^k$ | one imperfect fat-hand intervention | $\left\|g(\mathbf{x}^1)_A - g(\mathbf{x}^2)_{\hat{A}}\right\|_2$ | $-\sum_k H(g(\mathbf{x}^k)_{\hat{A}}), k\in\{1,2\}$ | block-id. | - |

| Work | Causal Model | Mixing Function | Invariance | Source of invariance, Inv. subset $A$ | Invariance reg. | Sufficiency reg. | Identifiability | Expl. |
|---|---|---|---|---|---|---|---|---|
| Daunhawer et al. (2023) | nonparam. | nonparam. | sample level on all realizations of $z_A^k$ | one imperfect fat-hand intervention, | $\left\| g_1(\mathbf{x}^1)_{\hat{A}} - g_2(\mathbf{x}^2)_{\hat{A}} \right\|_2$ | $-\sum_k H(g_k(\mathbf{x}^k)_{\hat{A}})$, $k \in \{1,2\}$ | block-id. | - |
| Ahuja et al. (2022b) | nonparam. | nonparam. | sample level on all realizations of $z_A^k$ | one imperfect fat-hand intervention | $\left\| g(\mathbf{x}^1)_{\hat{A}} - g(\mathbf{x}^2)_{\hat{A}} + \delta \right\|_2$ | $-\sum_k \mathbb{E}_{\mathbf{x}^k} \log p(\mathbf{x}^k | g(\mathbf{x}^k))$, $k \in \{1,2\}$ | block-id. | - |
| Locatello et al. (2020) | nonparam. | nonparam. | sample level | one imperfect fat-hand intervention | avg. encoding | $-\sum_k \mathbb{E}_{\mathbf{x}^k} \log p(\mathbf{x}^k | g(\mathbf{x}^k))$ , $k \in \{1,2\}$ | block-id. | - |
| Yao et al. (2023, Thm. 3.2) | nonparam. | nonparam. | sample level on all realizations of $z_A^k$ | partial observability | $\sum_{k,k' \in [K]} \left\| g_k(\mathbf{x})_{\hat{A}} - g_{k'}(\tilde{\mathbf{x}})_{\hat{A}} \right\|_2$ | $-\sum_{k \in [K]} H(g_k(\mathbf{x})_{\hat{A}})$ | block-id. | - |
| Yao et al. (2023, Thm. 3.8) | nonparam. | nonparam. | sample level on all realizations of $z_{A_i}^k$ | partial observability, $k \in V_i$ | $\sum_{k,k' \in V_i} \left\| g_k(\mathbf{x})_{\hat{A}(i,k)} - g_{k'}(\tilde{\mathbf{x}})_{\hat{A}(i,k')} \right\|_2$ | $-\sum_{k \in [K]} H(t_k \circ g_k(\mathbf{x}))$ | block-id | - |
| Brehmer et al. (2022) | nonparam. | nonparam. | sample level | perfect intervention per node | $D_{\mathrm{KL}}\big(q(\mathcal{I},\hat{\mathbf{z}}^{1,2} \mid \mathbf{x}^{1,2}) \| p(\mathcal{I},\hat{\mathbf{z}}^{1,2})\big)$, where $\hat{\mathbf{z}}^k := g(\mathbf{x}^k), k \in \{1,2\}$ | $-\sum_k \mathbb{E}_{\mathbf{x}^k} \log p(\mathbf{x}^k | g(\mathbf{x}^k))$, $k \in \{1,2\}$ | element-id. | - |
| TEMPORAL CRL | | | | | | | | |
| Lippe et al. (2022b) | nonparam. | nonparam. | transitional invariance on a distributional level | known-target interventions $\mathcal{I}_t$, invariant partition $A := [N] \backslash \mathcal{I}_t$ | $-H(\hat{\mathbf{z}}_{A^t}^t \mid \hat{\mathbf{z}}^{t-1})$ where $\hat{\mathbf{z}}^t := g(\mathbf{x}^t)$ | $-p(\mathbf{x}^t | \mathbf{x}^{t-1}, \mathcal{I}_t)$ | block-id. | - |
| Lippe et al. (2022a) | nonparam. | nonparam. | transitional invariance on a distributional level | known-target, partially perfect interventions $\mathcal{I}_t$, invariant partition $A := [N] \backslash \mathcal{I}_t$ | $-H(\hat{\mathbf{z}}_{A^t}^t \mid \hat{\mathbf{z}}^{t-1})$ where $\hat{\mathbf{z}}^t := g(\mathbf{x}^t)$ | $-p(\mathbf{x}^t | \mathbf{x}^{t-1}, \mathcal{I}_t)$ | block-id. | - |

| Work | Causal Model | Mixing Function | Invariance | Source of invariance, Inv. subset $A$ | Invariance reg. | Sufficiency reg. | Identifiability | Expl. |
|---|---|---|---|---|---|---|---|---|
| Lippe et al. (2023) | nonparam. | nonparam. | transitional invariance on a distributional level | binary interventions (interventional target unknown) | $D_{\mathrm{KL}}(q(\hat{z}^t \mid x^t) \| p(\hat{z}^t \mid \hat{z}^{t-1}, r^t))$, $r^t$ observed regime variable | $-\log p(x^t \mid \hat{z}^t)$ | block-id. | - |
| **MULTITASK CRL** | | | | | | | | |
| Lachapelle et al. (2023) | nonparam. | nonparam. | task support | task distribution, overlapping task supports, number of causal variables known | $\sum_t \|\hat{w}^{(t)}\|_{2,1}$ | $\sum_t \mathcal{R}(\hat{w}^{(t)} \circ g)$ | affine-id. | (e) |
| Fumero et al. (2024) | nonparam. | nonparam. | task support | task distribution, overlapping task supports | $H(\hat{w}) + \sum_t \|\hat{w}^{(t)}\|_1$ | $\sum_t \mathcal{R}(\hat{w}^{(t)} \circ g)$ | element-id. | (e) |
| **DOMAIN GENERALIZATION** | | | | | | | | |
| Sagawa et al. (2019) | nonparam. | nonparam. | risk | invariant relationship between label and invariant features, preserved under covariate shift | $\max_{k \in [K]} \mathcal{R}^k(w \circ g)$ | $\max_{k \in [K]} \mathcal{R}^k(w \circ g)$ | NA | (f) |
| Arjovsky et al. (2020) | nonparam. | nonparam. | risk | invariant relationship between label and invariant features, preserved under covariate shift | $\|\nabla_{w, w=1} \mathcal{R}^k(w \circ g)\|^2$ | $\sum_{k \in [K]} \mathcal{R}^k(w \circ g)$ | NA | - |

| Work | Causal Model | Mixing Function | Invariance | Source of invariance, Inv. subset $A$ | Invariance reg. | Sufficiency reg. | Identifiability | Expl. |
|---|---|---|---|---|---|---|---|---|
| Krueger et al. (2021) | nonparam. | nonparam. | risk | invariant relationship between label and invariant features, preserved under covariate shift | $\mathrm{Var}(\{\mathcal{R}^k(\mathbf{w} \circ g)\}_{k \in [K]})$ | $\sum_{k \in [K]} \mathcal{R}^k(\mathbf{w} \circ g)$ | NA | (f) |
| Ahuja et al. (2022a) | nonparam. | nonparam. | risk | invariant relationship between label and invariant features, preserved under covariate shift | $\|\nabla_{\mathbf{w}, \mathbf{w}=1} \mathcal{R}^k(\mathbf{w} \circ g)\|^2$ | $\sum_{k \in [K]} \mathcal{R}^k(\mathbf{w} \circ g) + \mathrm{Var}(\mathcal{R})$ | NA | - |

