# OpenReview forum: "Unifying Causal Representation Learning with the Invariance Principle"
_ICLR.cc/2025/Conference — ICLR 2025 Poster_

### Official Review · Reviewer_fBXY · 2024-11-04

**Soundness:** 3
**Presentation:** 3
**Contribution:** 3
**Rating:** 8
**Confidence:** 2

**Summary:**

This paper presents a unified framework for causal representation learning by using invariance principles to identify latent variables across various settings. The proposed framework unifies various existing CRL approaches, through the invariance principles. It introduces theoretical guarantees for latent variable identifiability, and demonstrates improved treatment effect estimation using synthetic experiments and real-world causal benchmarks.

**Strengths:**

- The paper is very well-written, with intuition provided for the definitions and main results.

- The review of related works is thorough.

- This work provides a valuable perspective by unifying multiple CRL approaches under a general framework and provides a novel perspective for the causal representation learning.

- The formalization of identifiability definitions and theorems is clear and sound.

**Weaknesses:**

While the paper presents a strong theoretical foundation, the framework is tested on limited data types, which may not reflect its applicability across various CRL settings.

**Questions:**

In practice, how do you choose the hyperparameters such as $\lambda$?

---

> ### Author Response · Authors · 2024-11-20
> **Response to Reviewer fBXY**
>
> Thank you for the positive feedback, and we will answer your questions as follows.
>
> &nbsp;
>
> ### Reply to weaknesses
>
> *Apply the invariance framework to diverse data types*
>
> * The **primary motivation** of our work is indeed **theoretical**; however, prior works (shown to be special cases of our framework) have been applied to diverse data types, such as video \[`7, 12, 13, 14`\], image \[`3, 4, 5, 6, 8`\] and text \[`4, 5`\]. We did not replicate these results because our work is a strict generalization of individual special cases.
> * **Instead, our experiments focus on new capabilities enabled by our framework**. Most notably, we have shown the *first application of causal representation learning on a real-world ecological problem* (and data – `Section 5.1`), which was posed as an open challenge for CRL in \[`15`\] as existing approaches are not directly applicable. The Nintervention experiment on numerical data (`Section 5.2`)  successfully removed the causal ingredient in existing interventional approaches and demonstrated that the data symmetry employed by these methods does not have to be causal (e.g., the result of an intervention).
> * In general, the proposed invariance framework is not limited to any certain data types, and we agree with the reviewer that applying the invariance approach to various data types would be exciting for future research. We believe that the flexibility of our framework will be broadly beneficial for applications in CRL, which are a known major challenge in the field, and we have already demonstrated this with our case study in ecology (`Section 5.1`).
>
> &nbsp;
>
> ### Reply to questions
>
> *How to choose the invariance regularizer $\\lambda_{\text{INV}}$?*
>
> * We suggest selecting the invariance regularizer $\\lambda_{\text{INV}}$ by **minimizing the treatment effect bias** on a small (ideally heterogeneous) validation set, as elaborated in `Lines 476-177`.
> * In `Figure 2`, we highlighted three models among all trained models (varying seed and $\\lambda\_{\\text{INV}}$): the one minimizing the **Empirical Risk** (yellow star), the **Invariance Loss** (violet star), and the **Treatment Effect** (green star) on the validation set.
> * Among these criteria, minimizing the **Treatment Effect** on the validation set **better** **approximates** the true Treatment Effect for the entire population (i.e., the minimum of the blue function).
> * Investigating **more principled selection criteria** and incorporating **uncertainty** **quantification** could be interesting directions for future work.

---

> > ### Author Response · Authors · 2024-11-29
> > **We kindly ask if the rebuttal has addressed the reviewer's concerns**
> >
> > Dear reviewer `fBXY`:
> >
> > We greatly appreciate your constructive feedback and thoughtful consideration. As the discussion period concludes soon, we kindly ask if the rebuttal has addressed your concerns.
> >
> > Should any further questions or concerns arise, we are happy to provide clarification during the remaining discussion time.

---

### Official Review · Reviewer_oQZt · 2024-11-05

**Soundness:** 2
**Presentation:** 2
**Contribution:** 4
**Rating:** 6
**Confidence:** 4

**Summary:**

Many approaches have emerged in recent years that tackle different variants of the causal representation learning (CRL) problem, under different assumptions and using different types of algorithms, and proving different levels of identifiability. This paper presents a unifying framework that allows the commonalities and differences between these different approaches to be more easily recognized.

**Strengths:**

In the quickly growing body of literature on CRL, it is becoming hard to see the forest for the trees. In this paper, the different main types are clearly delineated and discussed. I think this is an extremely valuable contribution to the field of CRL.

**Weaknesses:**

- Section 2 was very hard to read, and the definitions introduced here turn out to be misformulated. I only started to understand the intended meaning of definition 2.1 after reading parts of section D in the appendix: In multiview CRL scenarios (D.1), indeed the invariance property says that the two random variables take identical values. This type of invariance is captured by definition 2.1. But in multi-environment CRL (D.2), the invariance properties are not a function of the random variables (or their realizations), but of their density functions. The definition of $\iota$ is inappropriate in the latter case (specifically, one can't write $\iota(\mathbf{a})$ or $\iota:\mathbb{R}^{\lvert A \rvert} \to \mathcal{M}$ if $\iota$ is meant to be a function of the *density* of $\mathbf{a}$). Since the core contribution of this paper is to unify these different notions of invariance under a single theory (with a single notation), I believe it is of paramount importance that this notation is clear and correct.
- The following is not a weakness of the paper, but a weakness of the paper-venue pairing: A proper exposition of the theory in section 2 would require some examples to be presented, that demonstrate how the new definitions play out in known scenarios. At present, this technical section is very difficult to get through because there is no room for such examples, or for any redundancy to help the reader confirm they are on the right track. If the examples in appendix D could be integrated into the main text, the result would be a much better paper. Another sign this paper wants more space is that this paper has more to say than fits in 10 pages, is that of the four main contributions listed on page 2, only the fourth and part of the first are in the main text. Indeed I think the most valuable sections of this paper are currently in the appendix. Presenting this work as-is might be doing it a disservice in the long run.
- There were some more flaws in the theory on page 6, but I believe these are easily fixed. Proposition 3.3 says "only if", but what the proof shows is "if" (i.e., if (i) and (ii) are met, then the identifiability is established). Now the statements of propositions 3.2 and 3.3 are contradictory. The statement of proposition 3.2 should make clear that it only shows that identification is not possible *in general* here.

**Questions:**

- What is the content of assumption 2.1? It looks like this just defines $\mathbf{x}_{V_i}$, but oddly this definition doesn't involve $V_i$ anywhere, so it remains unclear what it means to say that $V_i$ is "unique known". I assume there is an implicit role for $V_i$ here? (As an aside, wouldn't the notation $\mathbf{x}^{V_i}$ be more consistent with the use of superscripts on the bottom of page 3?) Additionally, the "in particular" in the last sentence suggests that this is implied by the previous. If that's indeed the intended meaning, could you explain how this follows?
#### Small comments
- Please, sort the references alphabetically!
- $\iota$ is interchangeably called "invariance" and "invariant" property.
- line 215: "A subset of latent variable $\mathbf{z}_A$" - "variable" should be "variables". But then it sounds like a subset of $\mathbf{z}_A$ is meant, while $\mathbf{z}_A$ is the subset. I suggest to move "of the latent variables" to after the math.
- Definition 3.3: Replace "s.t." by "where" ("s.t." is used for conditions).
- page 22: I think on line 1145, "interior" should just be "subset". "equity" should be "equality", and on line 1152, in the big intersection the $S_k$ is missing.
#### Language-level comments
- l126: "smooth function**s**"
- l133: "violates **the** invariance principle"
- l146: "present**s**"
- l157: "interio**r**"
- the image of a function is sometimes denoted Im (l162), sometimes $\mathcal{I}$ (l951)
- l180: "as a": remove "as"
- l363: "Mult**i**-task"
- l987: remove "is a"
- l1109: "remain**s**"

---

> ### Author Response · Authors · 2024-11-20
> **Response to Reviewer oQZt**
>
> Thank you for your thoughtful comments and constructive feedback. We hope that our responses below adequately address your concerns.
>
> &nbsp;
>
> ### Reply to weaknesses
>
> *Explicit form of the invariance property $\\iota$ for special cases of our framework*
>
> * Thank you for your suggestion. The new `Table 1` has provided a list of concrete examples and explicit forms of the underlying invariance principle of individual special cases.
> * As shown by Table 1, multiple kinds of invariance are exhibited in **multi-environment CRL**. For example, when performing single-node intervention (row 3 in Table 1), the marginal distribution of non-descendants remains invariant.
> * Here, the invariance property \\(\\iota\\) is a function that **maps** the invariant partition \\(\\mathbf{z}\_A\\) **to its probability density function** \\(p\_{\\mathbf{z}\_A}\\), i.e., \\(\\mathbf{z}\_A \\mapsto p\_{\\mathbf{z}\_A}\\).
> * Correspondingly, the codomain $\\mathcal{M}$ is the set of valid probability density functions, formalized as: \\\[\\{ p\_{\\mathbf{z}\_A} : \\mathcal{Z}\_A \\to \\mathbb{R}\_{\>0} \\mid p\_{\\mathbf{z}\_A} \\text{ is a valid pdf} \\}.\\\]
> * In most cases, the codomain \\(\mathcal{M}\\) of \\(\iota\\) is implicitly determined by the choice of the equivalence relation $\sim_{\iota}$, as it is inherently defined as the quotient space corresponding to this relation (see `Definition 2.1`), and therefore does not require explicit specification.
>
>
> *Space issue and a better main-appendix split*
>
> * We indeed faced challenges in deciding what content to include in the main text due to the extensive volume of the paper.
> * The main reasons for submitting to ICLR are: (1) this paper focuses on **learning representations**, making it a strong fit for ICLR, and (2) we hope that the research idea will **reach a broader audience** highlighted through the conference, as many works we unified have been published at ICLR \[`4, 5, 12, 13, 14`\]. (3) While our contributions are primarily theoretical, they have immediate practical implications, which we have demonstrated with our case study in experimental ecology.
> * We have revised the split between the main text and the appendix, adding more concrete mathematical explanations to the main text (`Table 1`) to make the paper more convincing. Please refer to “**Modifications in the manuscript**”(under the [general response](https://openreview.net/forum?id=lk2Qk5xjeu&noteId=pVCS9DGbnM)) and the updated manuscript for further details. If you have other feedback on improving the presentation, we will happily take them into account.

---

> > ### Author Response · Authors · 2024-11-20
> > **Continued from previous part**
> >
> > *Additional counterexamples for Proposition 3.3*
> >
> > Thank you very much for carefully reviewing our paper\! We apologize for not explicitly providing arguments for the necessity of conditions (i) and (ii). To address this, we present two counterexamples that illustrate this aspect, which are now integrated in `Appendix E.3`. These are simple counterexamples, but we fully agree that they should be included in the paper for completeness.
> >
> > * **For (i)**. We consider the following scenario where condition (ii) holds, but condition (i) is violated. Given a non-invertible encoder $g \\in G^\*$ where $g(\\mathbf{x})\_A \= h(\\mathbf{z}\_A)$ for some diffeomorphism $h$ (implied by Thm. 3.1 ) and $g(\\mathbf{x})\_{A^\\mathrm{c}} \= \\vec{0} \\in \\{0\\}^{|A^\\mathrm{c}|}$. The encoder $g$ is non-invertible because the $A^\\mathrm{c}$ partition does not contain any information about $\\mathbf{z}\_{A^\\mathrm{c}}$ but only constant zeros. This can be formally verified by the mutual information between the observable $\\mathbf{x}$ and its encoding $g(\\mathbf{x})$:
> >   \\[
> >   I(\\mathbf{x}, g(\\mathbf{x})) \= I(f(\\mathbf{z}\_A, \\mathbf{z}\_{A^\\mathrm{c}}), \[h(\\mathbf{z}\_A), \\vec{0}\]) \= H(\\mathbf{z}\_A)
> >   \\]
> >   which is smaller than $H(\\mathbf{x})$ because the ground truth partition $\\mathbf{z}\_{A^\\mathrm{c}}$ contains independent information of $\\mathbf{z}\_{A}$.
> >   Formally, the mapping between the ground truth $\\mathbf{z}\_{A^\\mathrm{c}}$ and the representations $g(\\mathbf{x})\_{A^\\mathrm{c}}$ becomes a constant function
> >  \\[
> >   h\_0: \\mathcal{Z}\_{A^\\mathrm{c}} \\to \\{0\\}^{|A^\\mathrm{c}|}, \\, h\_0(\\mathbf{z}\_{A^\\mathrm{c}}) \= \\vec{0}
> > \\]
> >   which is clearly not a diffeomorphism, indicating $g(\\mathbf{x})\_{A^\\mathrm{c}}$ does not block-identify $\\mathbf{z}\_{A^\\mathrm{c}}$. Thus, condition (i) is shown to be necessary for the statement in Proposition 3.3.
> >
> > * **For (ii)**. Now, assume the complementary partition $\\mathbf{z}\_{A^c}$ depends on the identified partition $\\mathbf{z}\_A$, thereby violating condition (ii).  For example, let  $\\mathbf{z}\_A \= \\mathbf{z}\_{A^\\mathrm{c}}$ for some observable $\\mathbf{x} \= f(\\mathbf{z}\_A, \\mathbf{z}\_{A^\\mathrm{c}}) \= f(\\mathbf{z}\_A)$ (Note that here we slightly abuse the notation $f$ for simplification). Consider the same encoder $g \\in G^\*$ as described in the counterexample for (i), i.e., $g(\\mathbf{x})\_A \= h(\\mathbf{z}\_A)$ for some diffeomorphism $h$ (implied by Thm. 3.1) and $g(\\mathbf{x})\_{A^\\mathrm{c}} \= \\vec{0} \\in \\{0\\}^{|A^\\mathrm{c}|}$. However, note that this encoder $g$ is invertible because
> >  \\[
> >   I(\\mathbf{x}, g(\\mathbf{x})) \= I(f(\\mathbf{z}\_A), \[h(\\mathbf{z}\_A), \\vec{0}\]) \= H(\\mathbf{z}\_A) \= H(\\mathbf{x}).
> >   \\]
> >   Nevertheless, the mapping between the encoding $g(\\mathbf{x})\_{A^\\mathrm{c}}$ and the ground truth latents $\\mathbf{z}\_{A^\\mathrm{c}}$ remains the constant zero mapping $h\_0$ (see definition above), which fails to block-identify $\\mathbf{z}\_{A^c}$.
> >
> > &nbsp;
> >
> > ### Reply to questions
> >
> > Thank you for your valuable question, and we apologize for any confusion in the current version.
> >
> > *Further explanation on Assumption 2.1*
> >
> > * Assumption 2.1 ensures that for each invariance property \\(\\iota\_i\\) from the set \\(\\mathfrak{I}\\), there are **at least two observables** generated from latents that share this invariance property. In other words, each equivalence class \\(\[\\mathbf{z}\]\_{\\iota\_i}\\) consists of at least two elements. The **indexing subset \\(V\_i\\)** is used to represent any **subset of observables** that **share the underlying data symmetry** (originating from the equivalence relation \\(\\sim\_{\\iota}\\)).
> > * In summary, this assumption is designed to ensure that invariance properties are present within the available dataset; **otherwise, no identification results can be derived.**
> > * For example, let \\(\\mathfrak{I} \= \\{\\iota\_1, \\iota\_2\\}\\) represent the set of invariance properties. Suppose we have three observables \\(\\{\\mathbf{x}^1, \\mathbf{x}^2, \\mathbf{x}^3\\}\\) generated by the latent vectors \\(\\mathbf{z}^1, \\mathbf{z}^2, \\mathbf{z}^3\\), respectively. A scenario that violates Assumption 2.1 would be \\(\\mathbf{z}^1 \\sim\_{\\iota\_1} \\mathbf{z}^2\\), but no latents are equivalent with respect to \\(\\iota\_2\\). Consequently, there would be no shared partition \\(A\_2\\), making the identification statement in Theorem 3.1 meaningless.
> > * Also note that the same observable $\\mathbf{x}$ could be included in different subsets to present different equivalence relations, e.g., $V\_1 \= \\{1, 2\\}$ for $\\iota\_1$ and $V\_2 \= \\{1, 3\\}$ for $\\iota\_2$ in the previous example.
> > * We have removed the last sentence and updated the remark correspondingly for clarity.
> >
> > We are very grateful for your additional comments on the wording and the formatting. We have incorporated your suggestions in the revision.

---

> > > ### Author Response · Authors · 2024-11-29
> > > **We kindly ask if the rebuttal has addressed the reviewer's concerns**
> > >
> > > Dear reviewer `oQZt`:
> > >
> > > We greatly appreciate your constructive feedback and thoughtful consideration. As the discussion period concludes soon, we kindly ask if the rebuttal has addressed your concerns.
> > >
> > > Should any further questions or concerns arise, we are happy to provide clarification during the remaining discussion time.

---

> > > > ### Comment · Reviewer_oQZt · 2024-11-29
> > > >
> > > > Dear authors,
> > > >
> > > > Thank you for the effort you put into improving the presentation. I believe the changes you have made improve the clarity of the paper.
> > > >
> > > > I think we have a miscommunication about Proposition 3.3. Looking at its proof, I see that it uses conditions (i) and (ii) as assumptions, and concludes that $z_{A^c}$ can be identified. Hence, I wrote that in the statement of the proposition, the "only if" should be an "if" (i.e. the implication doesn't go left-to-right, but right-to-left). I appreciate how the two examples you provide are a worthwhile addition to the paper, but note in this context that they do not justify the "only if" in the proposition statement.
> > > >
> > > > About Assumption 2.1: Thank you for the elaborate explanation of the intuition of this assumption. However, my concern remains with the notation used, which I can't match to this intuition. Let me ask some specific questions that I hope will clarify the matter:
> > > > - Is $x_{V_i} = F([z]_{\sim_{\iota_i}})$ a definition or an equality?
> > > > - If it is an equality, then what is the definition of $x_{V_i}$? (I suggested in my initial review that maybe the $V_i$ should be a superscript...?)
> > > > - If it is a definition, then why doesn't $V_i$ appear in the right-hand side?
> > > >
> > > > (Re-reading the paragraph just before assumption 2.1, I think this should have said: "each inducing **as** a projection onto its quotient an~~d~~ invariance property $\iota_i$".)
> > > >
> > > > I also couldn't tell if you addressed my comment about $\iota(\mathbf{a})$ not being appropriate notation for a function of the density of $\mathbf{a}$.

---

> > > > > ### Author Response · Authors · 2024-12-01
> > > > > **Response to Reviewer oQZt [1/2]**
> > > > >
> > > > > Thank you very much for the further clarification.
> > > > >
> > > > > &nbsp;
> > > > >
> > > > > *Proposition 3.3*
> > > > >
> > > > > We sincerely apologize for the misunderstanding and show in the following that **“block identifiability of $\mathbf{z}_{A^c}$ by $(1-\phi) \oslash g$” implies conditions (i) and (ii)** (i.e., the left-to-right implication for “only if”). For convenience, we restate conditions (i) and (ii) as follows:
> > > > >
> > > > > (i) $g$ is invertible in the sense that $I(\mathbf{x}, g(\mathbf{x})) = H(\mathbf{x})$;
> > > > >
> > > > > (ii) $\mathbf{z}\_{A^{\mathrm{c}}}$ is independent on $\mathbf{z}\_{A}$.
> > > > >
> > > > > **For condition (i)**:  To show $g$ is invertible in the sense that $I(\\mathbf{x}, g(\\mathbf{x})) \= H(\\mathbf{x})$, it is equivalent to show that $H(\\mathbf{x} \\mid g(\\mathbf{x})) \= 0$  because
> > > > > $$
> > > > > I(\\mathbf{x}, g(\\mathbf{x})) \= H(\\mathbf{x}) \- H(\\mathbf{x} \\mid g(\\mathbf{x}))
> > > > > $$
> > > > > Writing $\\mathbf{x} \= f(\\mathbf{z}\_A, \\mathbf{z}\_{A^\\mathrm{c}})$, we have
> > > > > \\begin{equation\*}
> > > > > 	\\begin{aligned}
> > > > >     	H(\\mathbf{x} \~|\~ g(\\mathbf{x})) \= H(f(\\mathbf{z}\_A, \\mathbf{z}\_{A^\\mathrm{c}}) \~|\~ g(\\mathbf{x})) \= H(\\mathbf{z}\_A, \\mathbf{z}\_{A^\\mathrm{c}} \~|\~ g(\\mathbf{x})),
> > > > > 	\\end{aligned}
> > > > > \\end{equation\*}
> > > > > because the mixing function $f$ is a diffeomorphism as given by Definition B.2.
> > > > >
> > > > > Given that $\\phi \\oslash g(\\mathbf{x}) \= h(\\mathbf{z}\_A)$ (indicated by Theorem 3.1) and $(1-\\phi) \\oslash g(\\mathbf{x}) \= h^{c}(\\mathbf{z}\_{A^{c}})$ (block-identifiability) for some diffeomorphisms $h: \\mathcal{Z}\_{A} \\to \\mathcal{Z}\_A$ and $h^{c}: \\mathcal{Z}\_{A^\\mathrm{c}} \\to \\mathcal{Z}\_{A^\\mathrm{c}}$, we can further decompose the conditional entropy $ H(\\mathbf{x} \~|\~ g(\\mathbf{x}))$ into a sum of two separate terms
> > > > > \\begin{equation\*}
> > > > > 	\\begin{aligned}
> > > > >     	H(\\mathbf{x} \~|\~ g(\\mathbf{x}))
> > > > >     	& \= H(\\mathbf{z}\_A, \\mathbf{z}\_{A^\\mathrm{c}} \~|\~ \\phi \\oslash g(\\mathbf{x}), (1-\\phi) \\oslash g(\\mathbf{x}))\\\\
> > > > >     	& \= H(\\mathbf{z}\_{A^\\mathrm{c}} \~|\~ \\mathbf{z}\_{A}, \\phi \\oslash g(\\mathbf{x}), (1-\\phi) \\oslash g(\\mathbf{x})) \+ H(\\mathbf{z}\_{A} \~|\~  \\phi \\oslash g(\\mathbf{x}), (1-\\phi) \\oslash g(\\mathbf{x}))\\\\
> > > > >     	& \= H(\\mathbf{z}\_{A^\\mathrm{c}} \~|\~ \\mathbf{z}\_{A}, h(\\mathbf{z}\_A), h^\\mathrm{c}(\\mathbf{z}\_{A^{c}})) \+ H(\\mathbf{z}\_{A} \~|\~  h(\\mathbf{z}\_A), h^\\mathrm{c}(\\mathbf{z}\_{A^{c}})))
> > > > > 	\\end{aligned}
> > > > > \\end{equation\*}
> > > > > Since both $h, h^\\mathrm{c}$ are diffeomorphisms (meaning $h(\\mathbf{z}\_A)$, $h^\\mathrm{c}(\\mathbf{z}\_{A^\\mathrm{c}})$ fully determines $\\mathbf{z}\_A, \\mathbf{z}\_{A^\\mathrm{c}}$, respectively), both conditional entropy terms are zero; thus $H(\\mathbf{x} \~|\~ g(\\mathbf{x})) \= 0$. Hence, we have shown that $g$ is invertible in the sense that $I(\\mathbf{x}, g(\\mathbf{x})) \= H(\\mathbf{x})$.
> > > > >
> > > > > **For condition (ii)**:  Given that $\\phi \\oslash g(\\mathbf{x})$ block identifies $\\mathbf{z}\_A$, $(1-\\phi) \\oslash g(\\mathbf{x})$ cannot contain any information about $\\mathbf{z}\_A$ (Defn. 3.1), i.e.,
> > > > > $$
> > > > > I(\\mathbf{z}\_A , (1-\\phi) \\oslash g(\\mathbf{x})) \= 0
> > > > > $$
> > > > >
> > > > > Writing $(1-\\phi) \\oslash g(\\mathbf{x}) \= h^\\mathrm{c}(\\mathbf{z}\_{A^\\mathrm{c}})$, we have
> > > > > $$
> > > > > I(\\mathbf{z}\_A , h^\\mathrm{c}(\\mathbf{z}\_{A^\\mathrm{c}})) \= I(\\mathbf{z}\_A , \\mathbf{z}\_{A^\\mathrm{c}}) \= 0,
> > > > > $$
> > > > > as mutual information is invariant under invertible transformations and $h^\\mathrm{c}$ is invertible.
> > > > > Therefore, we have shown that $\\mathbf{z}\_{A^\\mathrm{c}}$ is independent on $\\mathbf{z}\_A$.

---

> > > > > > ### Author Response · Authors · 2024-12-01
> > > > > > **Response to Reviewer oQZt [2/2]**
> > > > > >
> > > > > > *Assumption 2.1*
> > > > > >
> > > > > > + We agree that the $V\_i$ is an indexing subset for the observables and should thus be superscript for consistency. We apologize for not having changed it in the current version and commit to improving this notation in future revisions.
> > > > > >
> > > > > > + $  \\mathbf{x}^{V\_i} \= F(\[\\mathbf{z}\]\_{\\sim \\iota\_i}) $  is **an equality following the definition of the function $F$ (Definition 2.2)**, which maps a set of equivalent latent vectors  $\[\\mathbf{z}\]\_{\\sim \\iota\_i} := \\{ \\tilde{\\mathbf{z}} \\in \\mathbb{R}^N : \\mathbf{z}\_{A\_i} \\sim\_{\\iota\_i} \\tilde{\\mathbf{z}}\_{A\_i} \\}$  to a set of observables $ \\mathbf{x}^{V\_i} := \\{ \mathbf{x}^{k}: k \\in V\_i \\subseteq \[K\]\\} $. Here, the observables are indexed by the subset $ V\_i$. This subset-indexing notation is similar to that used for $ \\mathbf{z}\_A $, as introduced in the "Notation" paragraph on Page 2\.
> > > > > >
> > > > > > **To improve the clairity of the notation of $x^{V\_i}$, we propose the following reformulation of Assumption 2.1:**
> > > > > >
> > > > > > **Assumption 2.1.** For each $ \\iota\_i \\in \\mathfrak{I} $, there exists a unique, known index subset $ V\_i \\subseteq \[K\] $ with at least two elements (i.e., $ |V\_i| \\geq 2 $) such that the corresponding observables $ \\mathbf{x}^{V\_i} := \\{\\mathbf{x}^k : k \\in V\_i\\} $ are generated from an equivalence class of latent vectors:
> > > > > > $$
> > > > > > \[\\mathbf{z}\]\_{\\sim\_{\\iota\_i}} := \\{ \\tilde{\\mathbf{z}} \\in \\mathbb{R}^N : \\mathbf{z}\_{A\_i} \\sim\_{\\iota\_i} \\tilde{\\mathbf{z}}\_{A\_i} \\}.
> > > > > > $$
> > > > > > Following Definition 2.2, the generating process is formally written as:
> > > > > > $$
> > > > > > \\mathbf{x}^{V\_i} \= \\{ f^k(\\mathbf{z}^k) : k \\in V\_i , \mathbf{z}^k  \in \[\\mathbf{z}\]\_{\\sim\_{\\iota\_i}} \\} \= F(\[\\mathbf{z}\]\_{\\sim\_{\\iota\_i}}).
> > > > > > $$
> > > > > >
> > > > > >
> > > > > > &nbsp;
> > > > > >
> > > > > > *Further explanation on $\\iota(\\mathbf{a})$*
> > > > > >
> > > > > > We would like to clarify that **$\\iota(\\mathbf{a})$ is not defined as a function of the density of $\\mathbf{a}$. Instead,** **it is an operator (i.e., function-valued function) that maps $\\mathbf{a}$ to its density $p\_{\\mathbf{a}}$**. Correspondingly, the codomain $\\mathcal{M}$ is the set of valid probability density functions, formalized as: \\[\\{ p\_{\\mathbf{a}} : \\mathbb{R}^{\\text{dim}(\\mathbf{a})} \to \mathbb{R}\_{\\geq 0} \\mid p\_{\\mathbf{a}} \\text{ is a valid pdf} \\}.\\]  In the context of multi-environment CRL, the marginal density $p\_{\\mathbf{z}\_A}$ is specified by the ground truth generating process and thus is unique, indicating there is only one image $p\_{\\mathbf{z}\_A}$ for each invariant latent vector $\\mathbf{z}\_A$ thus ensuring the $\\iota$ is a valid operator. If you have further questions, we are happy to address them.
> > > > > >
> > > > > > &nbsp;
> > > > > >
> > > > > > We kindly apologize for being unable to include the updates in the manuscript as the deadline for paper revision has passed, but we commit to implementing these changes if the reviewer agrees that they improve the clarity of the paper. We are also grateful for your additional suggestion on the paragraph before Assumption 2.1 and will integrate it correspondingly. If you have any additional actionable suggestions on improving the clarity, we will gladly consider incorporating them.

---

> > > > > > > ### Author Response · Authors · 2024-12-02
> > > > > > >
> > > > > > > Dear Reviewer `oQZt`,
> > > > > > >
> > > > > > > Thank you for your time reviewing and discussing our work. We kindly ask if you have any additional feedback on our latest response. Your insights have been invaluable throughout this process, and we hope to address any remaining concerns before the discussion deadline.

---

### Official Review · Reviewer_Egrh · 2024-11-05

**Soundness:** 3
**Presentation:** 3
**Contribution:** 4
**Rating:** 8
**Confidence:** 3

**Summary:**

The authors provide a general framework for identifying a group of $N$ random variables that encode invariances between a set of $K$ joint probability distributions, each over $D$ variables. Their framework describes many existing causal representation learning (CRL) algorithms and their corresponding (partial) identifiability results. The authors present their insights as methodologically connecting CRL across different settings where invariances may correspond to, for instance, the effect of specific interventions over a shared latent (natural) structural causal model, or various data missingness regimes.

**Strengths:**

The authors motivate their approach well by describing the breadth of assumptions made across different CRL settings. Their framework is laid out formally, and the boxed plain-English commentary is helpful for first-time reading.

**Weaknesses:**

(Section 4) As the authors aim to unify many existing problem settings, discussion on the application of the authors' framework is broad. In the main text this discussion is often informal. The authors could explicitly compare the invariance objectives for between common CRL approaches/settings in the main text. Does this lead to any surprising insights, for instance by showing methodological similarities between settings for which no comparison had previously been discussed?

(Future work) To my understanding, the variables preserving symmetries across different data pockets are not necessarily causal, but there are many existing problem settings for which certain corresponding symmetries and their interpretations can lead to causal variables being (block) identified. Do the authors see potential for their framework to be used to discover now CRL settings/approaches.

**Questions:**

(Future work) The authors' sufficiency constraint (Constraint 3.2) in general requires a supervised approach to enforce since the ground truth invariant representation $\mathbf{z}_{A_i}$ is unknown. Does this provide insights into the impossibility of unsupervised/semi supervised CRL in certain settings?

(Line 79) Is there a reason that the authors' framework is limited to describing nonparametric CRL? Are there known cases in which parametric assumptions cannot be represented through data invariances? Perhaps some parametric assumptions allow for CRL in a setting with only one dataset?

---

> ### Author Response · Authors · 2024-11-20
> **Response to Reviewer Egrh**
>
> We thank the reviewer for the positive feedback and will address the concerns individually.
>
> &nbsp;
> ### Reply to weaknesses
>
> *More mathematical explanations in Section 4:*
>
> Thank you very much for suggesting that we include the explicit mathematical explanations from Appendix D in the main text.
>
> * In the current manuscript, we opted for more plain text primarily due to space constraints. Additionally, we believed that a simpler, plain-text style might be easier for readers to digest, potentially reaching a broader audience.
> * However, we fully agree that presenting the related works in a more mathematical manner would enhance the comprehensiveness and rigor of the paper. Accordingly, we have added Table 1 to provide more mathematical details and to serve as concrete examples for the related works in Section 4.
>
>
> *What new insights does the invariance framework lead to?*
>
> Yes, there are many new insights that this framework enables, some of which have already been explored in the paper, and we can further highlight them if the reviewer suggests we do.
>
> * *Theoretical derivatives explored in the paper*
>   * Directly following the invariance framework, we have proven **identifiability with single-node imperfect intervention per node under the nonparametric setting** (`Corollary D.1`), which was an open conjecture from [`2`].
>
> * *Methodological similarities across areas*
>   * The proposed invariance framework sheds light on connecting causal representation learning with various settings, including more surprising ones like domain generalization and multi-task learning. For example, for the ISTAnt experiment (Section 5.1), we have derived the V-REx[`1`] loss from our framework. See `Appendix D.6, (f)`. The lack of applicability of CRL methods on ISTAnt was called out explicitly in [`15`].
>   * From the causality perspective, CRL methods are often arranged in Pearl’s causal hierarchy(e.g., “counterfactual”\[3, 4, 5\] or “interventional”[`2, 10, 11`]). Finding a methodological equivalence goes against the common folklore, and we think is very surprising.
>
> * *Other possible links we did not explicitly explore:*
>   * Our theory does not assume that the type of equivalence relation is the same for all latents. While we believe this may be useful in applications, we did not have a real-world problem in mind, and so we did not explore this empirically.
>
> * *Potential for new CRL approaches and future work*
>   * We strongly believe that this paper will inspire new work in causal representation learning. However, we hope that it will also shift the focus away from proving custom identification results based on assumption classes (e.g., one or more interventions per node or N views with overlapping latents), which implicitly describe equivalence relations. Instead, we hope **our work enables us to** **work backward from concrete invariances that can be assumed in applications**, as we did in our ISTAnt case study (`Section 5.1`). Overall, we hope our paper will enable a flurry of new results that are specific to and inspired by real-world problems, which current CRL research largely lacks.

---

> > ### Author Response · Authors · 2024-11-20
> > **Continued from previous part**
> >
> > ### Reply to questions
> >
> > *Can the sufficiency constraint be implemented without supervision?*
> >
> > Yes, the sufficiency constraint can be implemented using unsupervised or self-supervised approaches.
> >
> > * For example, minimizing the **reconstruction loss** is a common method to satisfy the sufficiency constraint, as it aims to preserve information about the original observations \[`6, 7, 8`\]. This approach does not require external supervision, as it aligns the reconstructed output with the input itself.
> > * In multiview CRL, the sufficiency constraint is enforced by maximizing the **differential entropy** of the learned representations \[`3, 4, 5`\]. In practice, this is often implemented using a **contrastive** loss, which operates in a self-supervised fashion.
> >
> > *How are parametric assumptions incorporated in our invariance framework?*
> >
> > **Parametric assumptions** in CRL are often included to achieve more **fine-grained identifiability** results.
> >
> > * For instance, in the multiview line of works \[`3, 4, 5`\], the mixing function is often assumed to be a general non-parametric diffeomorphism, and thus the theory proves block-identifiability on the shared latents.
> > * However, by assuming a parametric form of the mixing function, such as a finite degree polynomial, one immediately refines the block-identifiability \[`Definition 3.1`\] results to block affine-identifiability \[`Definition C.1`\], indicating the estimated latents are a block of affine transformation of the ground truth latents. This connection is formalized in `Proposition C.2` (Proof in Appendix E.4).
> > * From this perspective, parametric settings are not excluded from our framework but are rather specified by additional assumptions that are **orthogonal** **to** the underlying **invariance** properties.
> > * In fact, `Table 4` (at the end of the Appendix) *has included several works that assume a parametric form of the latent causal model or the mixing function* \[`9, 10, 11`\].

---

> > > ### Comment · Reviewer_Egrh · 2024-11-24
> > >
> > > Many thanks to the authors for clarifying my concerns. I am satisfied by their proposed changes to presentation and will retain my score.

---

### Official Review · Reviewer_khbJ · 2024-11-06

**Soundness:** 2
**Presentation:** 2
**Contribution:** 4
**Rating:** 6
**Confidence:** 4

**Summary:**

This work provides a unifying theory for multiple causal representation learning works by using the fact that all these works are learning some form of invariance. It also acts as a review work that neatly characterises the different problem settings in causal representation learning.

**Strengths:**

- The paper is comprehensive and the unifying theory seems like a very good way to consolidate the findings in the causal representation learning literature.

**Weaknesses:**

- I think some of the contributions are over stated, contribution 2 and 3 in L86 and L90 are already known. While it's great that these are included, I would not put them down as contributions of the work. It may mislead the reader to think that there are new findings here. The other contributions are enough for a good paper.
- This work can be *greatly* improved if the grey boxes after the definitions, assumptions etc. in sections 2 and 3 included explicit examples relating to literature. This is not present in section 4 and only partially in section D. With this I mean something like, what is the invariance property in multi environment learning and temporal learning? What is the space $\mathcal{M}$ in all cases? The intuition in L135 to L143 is ok, but I would like to see *explicit* definitions in each related work case to be thoroughly convinced that each case is covered by the theory presented. The same is true of things like definition 2.2. Explicit examples of why the definition can cover multiview, multienvironment etc. will help greatly. I'm not sure saying the data is "paired" in L170 is enough. In short, I would like to see how each related work fits in mathematically with your theory. I'm also not convinced the tables at the end do this.
- I'm very unsure how the theory lead to a method that can work on the ISTAnt dataset as mentioned in L54. I can't see the reason why previous assumptions were insufficient written anywhere, or why the theory directly leads to the application of the domain generalisation method? As this is stated as a main contribution, this needs to be a lot clearer than it is.

I think this work will be of great interest to the community, but more careful exposition of why most CRL works can be seen as a special case, and more careful statement of the contributions, would greatly help the clarity of this work.

**Questions:**

- L258: Where is H defined? If H is used in constraint 3.2, it should be defined, or the definition pointed to.
- L63: What is a data pocket? This is used a few times in the work but has not been defined.

---

> ### Author Response · Authors · 2024-11-20
> **Response to Reviewer khbJ**
>
> Thank you for your valuable feedback; we hope our responses provide the necessary clarification.
>
> &nbsp;
>
> ### Reply to weaknesses
>
> *Clarifications on contributions 2 and 3:*
>
> * It is possible that these two contributions may have been overstated, and we agree that this is not one of the most important points of the paper in any case. We are not aware of existing literature that extensively examines the **transition** between different levels of identification granularity (`contribution 2`), or the **distinction** between **assumptions** for **variable** identifiability and **graph** **identifiability** (`contribution 3`).
> * We would be very grateful if you could **suggest** **relevant theoretical results or similar works** that we could cite to provide a more complete context in our paper. What we want to say is that *any CRL algorithm is capable of achieving any of the identifiability definitions depending on the assumptions we highlight*. For example, single fat-hand intervention (which was shown to provide block-identifiability under non-parametric mixing \[`3, 4, 5`\]) and polynomial mixing lead to block-affine identifiability (implied by `Proposition C.2`); or paired perfect intervention per node, leading to element-wise identifiability under non-parametric mixing \[`2`\], can be refined to affine-identifiability by employing polynomial mixing (again, implied by `Proposition C.2`). **All these individual cases do not need to be proven one-by-one and to the best of our knowledge were not included in neither the original nor follow-up papers, with our result we can tell exactly which identifiability result can be achieved.**
>
> *Explicit examples of CRL categories:*
>
> * The new `Table 1` provides concrete examples of special cases of our theory and summarizes their underlying invariance (see **“Concrete examples of our theory”** under the [general response](https://openreview.net/forum?id=lk2Qk5xjeu&noteId=pVCS9DGbnM) for more details). We have included this table in the revised version, and hopefully, this will help with understanding both Definition 2.1 and 2.2. If you have further suggestions regarding this aspect, we will be happy to incorporate them as well.
>
> *Why are previous CRL approaches not directly applicable to ISTAnt?*
>
> * We would like to clarify that previous assumptions in CRL lack the flexibility needed for ISTAnt. In this setting, we consider a concrete task to identify the factual outcome $Y$, e.g., whether ants are grooming or not.
> * Since **interventions** on the behavior of live ants **cannot be performed**, multi-environment approaches (interventional CRL) are not applicable.  Additionally, contrastive **multi-view** approaches **lack negative samples** (we need pairs of samples with the same or different $Y$, while we can assume that nearby frames will have similar $Y$, we cannot assume that far away frames will have different $Y$); we **cannot** **take** **actions** **over time** as required by **temporal** CRL; and we **only have one task** which is misaligned with the assumption in **multi-task** CRL.
>
> *Why does V-REx\[`1`\] (the method used for the ISTAnt case study) fit under the invariance framework?*
>
> * For ISTAnt, we exploit an **invariance** we can assume **on the causal mechanism** among different experiment settings. Although the training set is biased, called out as a challenge in \[`15`\], we can naturally incorporate this constraint to improve the estimate. The proof in the appendix (see `Appendix D.6, (f)`) derives how the domain generalization approach used in the ISTAnt experiment (i.e., V-REx \[`1`\]) falls under our invariance framework. This proof is also referenced in the main text (`Page 8, Line 463`).
>
> &nbsp;
>
> ### Reply to questions
>
> *Notations in Constraint 3.2*
>
> * Thank you for your question. Here, \\( H(\\cdot) \\) denotes the **differential entropy** of the ground truth latent distribution \\( p\_{\\mathbf{z}\_{A\_i}} \\), where \\( A\_i \\subset \[N\] \\) is the index subset for which the invariance property \\( \\iota\_i \\) holds. We have updated the manuscript accordingly to provide additional clarity.
>
> *Explanations for “data pockets”*
>
> * We apologize for the confusion. **By "data pocket," we refer to a set of observables generated from latent variables that share certain invariance properties**, such as the observables $\\mathbf{x}^1, \\mathbf{x}^2$ in the examples from Table 1\. This can be **formally** defined using $\\mathbf{x}\_{V\_i}$ as introduced in Assumption 2.1. We have incorporated this clarification in the revised manuscript; see `Assumption 2.1`.

---

> > ### Comment · Reviewer_khbJ · 2024-11-27
> > **Response to authors**
> >
> > Thank you for your response. I will increase my score and I like the work, however, I still feel the presentation of the material can be vastly improved. I think it would help the reader the most is if the related work in Appendix D was shown to fit neatly in with the theory. The discussion in Appendix D is very high level and it's hard to parse the fact that it fits in with the theory introduced in the paper. As this is the main claim in the paper, I feel this could be much clearer.
> >
> > >All these individual cases do not need to be proven one-by-one and to the best of our knowledge were not included in neither the original nor follow-up papers, with our result we can tell exactly which identifiability result can be achieved.
> >
> > The above point is interesting, but I did not get this from reading the work again.
> >
> > > contribution 3
> >
> > It's nice to have these results in one place, but they are simply a restatement of known results. There are also missing results such as those for location scale noise models.  Further the claim "we draw a distinction between the causal assumptions necessary for graph discovery and those that may not be required for variable discovery" is not really justified anywhere in Appendix C.2.

---

> > > ### Author Response · Authors · 2024-11-28
> > > **Response to Reviewer khbJ**
> > >
> > > Thank you for reviewing our rebuttal and providing further feedback.
> > >
> > > &nbsp;
> > >
> > > *Regarding contributions 2 and 3*
> > >
> > > We are happy that our rebuttal has clarified contributions 2 and 3 in our paper. We have updated our manuscript correspondingly for added clarity (`Page 2`: List of contributions, `Page 21`: Appendix C.1 and C.2).
> > >
> > >
> > > *Presentation of Appendix D*
> > >
> > > * Appendix D focuses on explaining the ground invariance principle exploited in special cases of our framework, which is now also summarized in `Table 1` together with concrete examples. Our theory is abstract and holds for any equivalence class satisfying our Definition 2.1 and Assumption 2.1. In practice, what we called “invariance constraint” (`Constraint 3.1`) is implemented with a discriminant, i.e., a function that disambiguates samples that are not equivalent; also see Section 2 in \[`c`\] for another example in machine learning.  Note that the discriminant of a specific equivalence class does not need to be unique. For example, when considering paired interventions per node \[`a, b`\], there are two possible discriminates:
> > >     1) The interventional targets are the same (see `Table 1`, Row 2), which may be difficult to estimate directly in practice;
> > >     2) Only the scores of the intervened variables differ (see `Table 4`, which reports the specific implementation choice taken in \[`a, b`\]).
> > > * **It does not matter which invariance relation is enforced. Any choice of a discriminant function implicitly defines an invariance property, which leads to identification as long as the invariance aligns with the ground truth variables**. In the paper, we originally preferred the terminology of “invariance constraint” because the discriminant is used to constrain the representation. The discriminant choice is irrelevant to the theory, which only uses the existence of an arbitrary invariance relation that is entailed by a discriminant and aligns with a subset of ground-truth latents.
> > >
> > >
> > >
> > > **We propose the following improvements to the paper’s presentation:**
> > >
> > > 1. We add an explanation using the “discriminant” terminology of the fact that each discriminant function defines an equivalence relation, but they are not unique.
> > > 2. **Table 1 (in the main text):** explains which type of invariance relation pertains to different CRL settings (i.e., not the concrete implementation taken by specific papers). Additionally, we will cite which papers fall within each category we unified.
> > > 3. **Table 4 at the end of the paper:** We split the table according to the categories as in Table 1, and list the concrete implementation.
> > >
> > > We kindly apologize for not being able to show the result as the deadline to update the manuscript has passed, but we commit to implementing this change if the reviewer agrees that this will make the paper clearer. If you have any additional concrete suggestions, we would be glad to consider incorporating them.
> > >
> > > &nbsp;
> > >
> > > ### Reference for this response
> > >
> > > \[a\] Burak Varici, Emre Acartürk, Karthikeyan Shanmugam, and Ali Tajer. General identifiability and achievability for causal representation learning. In International Conference on Artificial Intelligence and Statistics, pp. 2314–2322. PMLR, 2024a.
> > >
> > > \[b\] Julius von Kügelgen, Michel Besserve, Liang Wendong, Luigi Gresele, Armin Keki´c, Elias Barein- boim, David Blei, and Bernhard Schölkopf. Nonparametric identifiability of causal represen- tations from unknown interventions. Advances in Neural Information Processing Systems, 36, 2024.
> > >
> > > \[c\] Soatto, Stefano, Paulo Tabuada, Pratik Chaudhari, and Tian Yu Liu. Taming ai bots: Controllability of neural states in large language models. arXiv preprint arXiv:2305.18449 (2023).

---

> > > > ### Author Response · Authors · 2024-12-02
> > > >
> > > > Dear Reviewer `khbJ`,
> > > >
> > > > Thank you for your time reviewing and discussing our work. We kindly ask if you have any additional feedback on our latest response. Your insights have been invaluable throughout this process, and we hope to address any remaining concerns before the discussion deadline.

---

### Author Response · Authors · 2024-11-20
**General response to all reviewers and AC**

We thank all reviewers for their time and valuable feedback. We very much appreciate that they consider this paper’s contribution to be “excellent” (`khbj`) and “extremely valuable to the field of CRL”(`oQZt`). We are also happy to see our framework is received as “formal”(`Egrh`), “clear and sound”(`fBXY`). Also, thank you for your positive feedback on the writing style of the paper, such as “very well written”(`fBXY`) and “plain-English comments are helpful”(`Egrh`).

&nbsp;

### Concrete examples of our theory (`khbj`, `Egrh`, `oQZt` )

This seems to be the only common criticism by the reviewers. We thank you for your constructive feedback and apologize for not clearly summarizing this aspect in the main text and relegating the discussion only in Appendix D, with definitions and the explicit implementations of the invariance constraints.

* To address this, we have included a new table (`Table 1, Page 3`\) that provides **concrete examples** of individual CRL sub-categories and domain generalization that our framework unifies. Each example is accompanied by an **illustration** **of the corresponding** **invariance.**
* Intuitively, we remark that the invariance property  \\(\\iota\\) is introduced as a mathematical tool to formalize the equivalence relation $\\sim\_{\\iota}$, distinguishing which latent vectors (and their corresponding observables) belong to the same equivalence class and which do not. The codomain $\\mathcal{M}$ of $\\iota$ is also directly implied by the choice of the equivalence relation but does not need to be made explicit (see `Definition 2.1`). In practice, what must be specified is the implementation of the constraint (which can be thought of as a discriminant for an equivalence class) and which observables fall into the same equivalence class.


We think it would be more accessible to keep the related works in the main text at a high level, referring to the appendix for the specific implementation of the constraints (which is different in different papers). Instead, we give more precise descriptions of the invariances in the new `Table 1`, which is presented in `Section 2` in the updated paper, where we describe the problem setting with concrete examples. We are open to suggestions from the reviewers, but we think it is better to keep the setting and the previous implementations separate.

&nbsp;

### Modifications in the manuscript

*Note: All modifications in the manuscript are highlighted in green.

1. Table 1, summarizing concrete examples of different categories, is included in Section 2 of the updated manuscript (`Page 3`).
2. An intuitive explanation is added for Assumption 2.1 (`Page 5, Lines 224-226`).
3. Further explanation is provided for notations in Constraint 3.2 (`Page 6, Lines 298-299`).
4. Additional counterexamples are added for Proposition 3.3  in Appendix E.3 (`Page 32-33`).

---

> ### Author Response · Authors · 2024-11-20
> **References for all responses**
>
> [1] David Krueger, Ethan Caballero, Joern-Henrik Jacobsen, Amy Zhang, Jonathan Binas, Dinghuai
> Zhang, Remi Le Priol, and Aaron Courville. Out-of-distribution generalization via risk extrapola-
> tion (rex). In International conference on machine learning, pages 5815–5826. PMLR, 2021
>
> [2] Julius von Kügelgen, Michel Besserve, Liang Wendong, Luigi Gresele, Armin Keki´c, Elias Barein-
> boim, David Blei, and Bernhard Schölkopf. Nonparametric identifiability of causal represen-
> tations from unknown interventions. Advances in Neural Information Processing Systems, 36,
> 2024.
>
> [3] Julius von Kügelgen, Yash Sharma, Luigi Gresele, Wieland Brendel, Bernhard Schölkopf, Michel
> Besserve, and Francesco Locatello. Self-supervised learning with data augmentations provably
> isolates content from style. Advances in neural information processing systems, 34:16451–16467,
> 2021.
>
> [4] Dingling Yao, Danru Xu, Sebastien Lachapelle, Sara Magliacane, Perouz Taslakian, Georg Martius,
> Julius von Kügelgen, and Francesco Locatello. Multi-view causal representation learning with
> partial observability. In The Twelfth International Conference on Learning Representations, 2023.
>
> [5] Imant Daunhawer, Alice Bizeul, Emanuele Palumbo, Alexander Marx, and Julia E Vogt. Identifia-
> bility results for multimodal contrastive learning. In The Eleventh International Conference on
> Learning Representations, 2023.
>
> [6] Kartik Ahuja, Jason S Hartford, and Yoshua Bengio. Weakly supervised representation learning
> with sparse perturbations. Advances in Neural Information Processing Systems, 35:15516–15528,
> 2022b.
>
> [7] Sébastien Lachapelle, Pau Rodríguez López, Yash Sharma, Katie Everett, Rémi Le Priol, Alexan-
> dre Lacoste, and Simon Lacoste-Julien. Nonparametric partial disentanglement via mecha-
> nism sparsity: Sparse actions, interventions and sparse temporal dependencies.
>
> [8] Francesco Locatello, Ben Poole, Gunnar Raetsch, Bernhard Schölkopf, Olivier Bachem, and
> Michael Tschannen. Weakly-supervised disentanglement without compromises. In Hal Daumé
> III and Aarti Singh, editors, Proceedings of the 37th International Conference on Machine Learn-
> ing, volume 119 of Proceedings of Machine Learning Research, pages 6348–6359. PMLR, 13–18
> Jul 2020.
>
> [9] Kartik Ahuja, Amin Mansouri, and Yixin Wang. Multi-domain causal representation learning via
> weak distributional invariances. In International Conference on Artificial Intelligence and Statis-
> tics, pages 865–873. PMLR, 2024.
>
> [10] Jiaqi Zhang, Kristjan Greenewald, Chandler Squires, Akash Srivastava, Karthikeyan Shanmugam,
> and Caroline Uhler. Identifiability guarantees for causal disentanglement from soft interventions.
> Advances in Neural Information Processing Systems, 36, 2024a.
>
> [11] Chandler Squires, Anna Seigal, Salil S. Bhate, and Caroline Uhler. Linear causal disentanglement
> via interventions. In International Conference on Machine Learning, volume 202, pages 32540–
> 32560. PMLR, 2023.
>
> [12] Phillip Lippe, Sara Magliacane, Sindy Löwe, Yuki M Asano, Taco Cohen, and Efstratios Gavves. Causal representation learning for instantaneous and temporal effects in interactive systems. In The Eleventh International Conference on Learning Representations, 2022a.
>
> [13] Weiran Yao, Yuewen Sun, Alex Ho, Changyin Sun, and Kun Zhang. Learning temporally causal latent processes from general temporal data. International Conference on Learning Representations, 2022a.
>
> [14] David A Klindt, Lukas Schott, Yash Sharma, Ivan Ustyuzhaninov, Wieland Brendel, Matthias Bethge, and Dylan Paiton. Towards nonlinear disentanglement in natural data with temporal sparse coding. In International Conference on Learning Representations, 2021.
>
> [15] Riccardo Cadei, Lukas Lindorfer, Sylvia Cremer, Cordelia Schmid, and Francesco Locatello. Smoke and mirrors in causal downstream tasks. Advances in Neural Information Processing Systems, 37, 2024.

---

### Meta-Review · Area_Chair_1vX3 · 2024-12-22

**Metareview:**

Invariances are at the core of causal modeling, but also at a large variety of approaches for robust modeling that incorporate theoretical rather than purely data-driven approaches for learning. Connections about the different ways of invariant representations from causal and non-causal points of view are presented in a coherent and comprehensive manner. All reviewers agreed this is a valuable contribution that will reach a considerable audience, and very well-aligned with the goals of ICLR.

**Additional Comments On Reviewer Discussion:**

Several suggestions were provided on how to improve the presentation of the paper, which seemed to be the more contentious point Authors were receptive and there was reasonable engagement on all parts.

---

### Decision · Program_Chairs · 2025-01-22

Accept (Poster)